# Recent evolution of a TET-controlled and DPPA3/STELLA-driven pathway of passive DNA demethylation in mammals

Christopher B. Mulholland [1], Atsuya Nishiyama [2,9], Joel Ryan [1,9], Ryohei Nakamura[3], Merve Yiğit[1], Ivo M. Glück[4], Carina Trummer[1], Weihua Qin[1], Michael D. Bartoschek [1], Franziska R. Traube [5], Edris Parsa[5], Enes Ugur[1,6], Miha Modic[7], Aishwarya Acharya [1], Paul Stolz [1], Christoph Ziegenhain[8], Michael Wierer [6], Wolfgang Enard[8], Thomas Carell [5], Don C. Lamb [4], Hiroyuki Takeda [3], Makoto Nakanashi [2], Sebastian Bultmann [1,10 ✉] & Heinrich Leonhardt [1,10 ✉]

Genome-wide DNA demethylation is a unique feature of mammalian development and naïve pluripotent stem cells. Here, we describe a recently evolved pathway in which global hypomethylation is achieved by the coupling of active and passive demethylation. TET activity is required, albeit indirectly, for global demethylation, which mostly occurs at sites devoid of TET binding. Instead, TET-mediated active demethylation is locus-specific and necessary for activating a subset of genes, including the naïve pluripotency and germline marker *Dppa3* (*Stella, Pgc7*). DPPA3 in turn drives large-scale passive demethylation by directly binding and displacing UHRF1 from chromatin, thereby inhibiting maintenance DNA methylation. Although unique to mammals, we show that DPPA3 alone is capable of inducing global DNA demethylation in non-mammalian species (Xenopus and medaka) despite their evolutionary divergence from mammals more than 300 million years ago. Our findings suggest that the evolution of *Dppa3* facilitated the emergence of global DNA demethylation in mammals.

[1] Department of Biology II and Center for Integrated Protein Science Munich (CIPSM), Human Biology and BioImaging, Ludwig-Maximilians-Universität München, Planegg-Martinsried, Germany. [2] Division of Cancer Cell Biology, The Institute of Medical Science, The University of Tokyo, 4-6-1 Shirokanedai, Minato-ku, Tokyo 108-8639, Japan. [3] Department of Biological Sciences, Graduate School of Science, The University of Tokyo, 7-3-1 Hongo, Bunkyo-ku, Tokyo 113-0033, Japan. [4] Physical Chemistry, Department of Chemistry, Center for Nanoscience, Nanosystems Initiative Munich and Center for Integrated Protein Science Munich, Ludwig-Maximilians-Universität München, Munich, Germany. [5] Center for Integrated Protein Science (CIPSM) at the Department of Chemistry, Ludwig-Maximilians-Universität München, Munich, Germany. [6] Department of Proteomics and Signal Transduction, Max Planck Institute for Biochemistry, Am Klopferspitz 18, 82152 Martinsried, Germany. [7] The Francis Crick Institute and UCL Queen Square Institute of Neurology, London, UK. [8] Department of Biology II, Anthropology and Human Genomics, Ludwig-Maximilians-Universität München, Planegg-Martinsried, Germany. [9] These authors contributed equally: Atsuya Nishiyama, Joel Ryan. [10] These authors jointly supervised: Sebastian Bultmann, Heinrich Leonhardt. ✉email: bultmann@bio.lmu.de; h.leonhardt@lmu.de

During early embryonic development the epigenome undergoes massive changes. Upon fertilization, the genomes of highly specialized cell types—sperm and oocyte—need to be reprogrammed in order to obtain totipotency. This process entails decompaction of the highly condensed gametic genomes and global resetting of chromatin states to confer the necessary epigenetic plasticity required for the development of a new organism[1]. At the same time, the genome needs to be protected from the activation of transposable elements (TEs) abundantly present in vertebrate genomes[2]. Activation and subsequent transposition of TEs result in mutations that can have deleterious effects and are passed onto offspring if they occur in the germline during early development[2,3]. The defense against these genomic parasites has shaped genomes substantially[4,5].

Cytosine DNA methylation (5-methylcytosine (5mC)) is a reversible epigenetic mark essential for cellular differentiation, genome stability, and embryonic development in vertebrates[6]. Predominantly associated with transcriptional repression, DNA methylation has important roles in gene silencing, genomic imprinting, and X inactivation[7]. However, the most basic, conserved function of DNA methylation is the stable repression of TEs and other repetitive sequences[8]. Accordingly, the majority of genomic 5mC is located within these highly abundant repetitive elements. Global DNA methylation loss triggers the derepression of transposable and repetitive elements, which leads to genomic instability and cell death, highlighting the crucial function of vertebrate DNA methylation[9–14]. Hence, to ensure continuous protection against TE reactivation, global DNA methylation levels remain constant throughout the lifetime of non-mammalian vertebrates[15–18]. Paradoxically, mammals specifically erase DNA methylation during preimplantation development[19,20], a process that would seemingly expose the developing organism to the risk of genomic instability through the activation of TEs. DNA methylation also acts as an epigenetic barrier to restrict and stabilize cell fate decisions and thus constitutes a form of epigenetic memory. The establishment of pluripotency in mammals requires the erasure of epigenetic memory and as such, global hypomethylation is a defining characteristic of pluripotent cell types including naïve embryonic stem cells (ESCs), primordial germ cells (PGCs), and induced pluripotent stem cells (iPSCs)[21].

In animals, DNA methylation can be reversed to unmodified cytosine by two mechanisms; either actively by Ten-eleven translocation (TET) dioxygenase-mediated oxidation of 5mC in concert with the base excision repair machinery[22–25] or passively by a lack of functional DNA methylation maintenance during the DNA replication cycle[26,27]. Both active and passive demethylation pathways have been implicated in the genome-wide erasure of 5mC accompanying mammalian preimplantation development[28–34]. Despite the extensive conservation of the TET enzymes and DNA methylation machinery throughout metazoa[35], developmental DNA demethylation appears to be unique to placental mammals[19,36–43]. In contrast, 5mC patterns have been found to remain constant throughout early development in all non-mammalian vertebrates examined to date[15,44–48]. This discrepancy implies the existence of yet-to-be-discovered mammalian-specific pathways that orchestrate the establishment and maintenance of global hypomethylation.

Here, we use mouse embryonic stem cells (ESCs) cultured in conditions promoting naïve pluripotency[49–51] as a model to study global DNA demethylation in mammals. By dissecting the contribution of the catalytic activity of TET1 and TET2 to global hypomethylation, we find that TET-mediated active demethylation drives the expression of the Developmental pluripotency-associated protein 3 (DPPA3/PGC7/STELLA). We show that DPPA3 directly binds UHRF1 and triggers its release from chromatin, thereby inhibiting maintenance methylation and

causing global passive demethylation. Although DPPA3 is only found in mammals, we found that DPPA3 can also potently induce global demethylation when introduced into non-mammalian vertebrates. In summary, our study uncovers a novel TET-controlled and DPPA3-driven pathway for passive demethylation in naïve pluripotency in mammals.

## Results

### TET1 and TET2 indirectly protect the naïve genome from hypermethylation.
Mammalian TET proteins, TET1, TET2, and TET3, share a conserved catalytic domain and the ability to oxidize 5mC but exhibit distinct expression profiles during development[52]. Naïve ESCs and the inner cell mass (ICM) of the blastocyst from which they are derived feature high expression of Tet1 and Tet2 but not Tet3[29,53–55]. To dissect the precise contribution of TET-mediated active DNA demethylation to global DNA hypomethylation in naïve pluripotency we generated isogenic Tet1 (T1CM) and Tet2 (T2CM) single as well as Tet1/Tet2 (T12CM) double catalytic mutant mouse ESC lines using CRISPR/Cas-assisted gene editing (Supplementary Fig. 1). We derived two independent clones for each mutant cell line and confirmed the inactivation of TET1 and TET2 activity by measuring the levels of 5-hydroxymethylcytosine (5hmC), the product of TET-mediated oxidation of 5mC[22] (Supplementary Fig. 1i). While the loss of either Tet1 or Tet2 catalytic activity significantly reduced 5hmC levels, inactivation of both TET1 and TET2 resulted in the near total loss of 5hmC in naïve ESCs (Supplementary Fig. 1i) indicating that TET1 and TET2 account for the overwhelming majority of cytosine oxidation in naïve ESCs. We then used reduced representation bisulfite sequencing (RRBS) to determine the DNA methylation state of T1CM, T2CM, and T12CM ESCs as well as wild-type (wt) ESCs. All Tet catalytic mutant (T1CM, T2CM, and T12CM) cell lines exhibited severe DNA hypermethylation throughout the genome including promoters, gene bodies, and repetitive elements (Fig. 1a, b and Supplementary Fig. 2a). The increase in DNA methylation was particularly pronounced at LINE-1 (L1) elements of which 97%, 98%, and 99% were significantly hypermethylated in T1CM, T2CM, and T12CM ESCs, respectively (Supplementary Fig. 2b). This widespread DNA hypermethylation was reminiscent of the global increase in DNA methylation accompanying the transition of naïve ESCs to primed epiblast-like cells (EpiLCs)[54,56,57], which prompted us to investigate whether the DNA methylation signature in T1CM, T2CM, and T12CM ESCs resembles that of more differentiated cells. In line with this hypothesis, Tet catalytic mutant ESCs displayed DNA methylation levels similar to or higher than those of wt EpiLCs (Supplementary Fig. 2c). Moreover, hierarchical clustering and principal component analyses (PCA) of the RRBS data revealed that ESCs from Tet catalytic mutants clustered closer to wt EpiLCs than wt ESCs (Fig. 1c and Supplementary Fig. 2d). In fact, the vast majority of significantly hypermethylated CpGs in Tet catalytic mutant ESCs overlapped with those normally gaining DNA methylation during the exit from naïve pluripotency (Fig. 1d). In contrast, T1CM, T2CM, and T12CM transcriptomes are clearly clustered by differentiation stage, indicating that the acquisition of an EpiLC-like methylome was not due to premature differentiation (Supplementary Fig. 2e). When comparing our data to that of TET knockout ESCs[58], we found that the catalytic inactivation of the TET proteins caused a far more severe hypermethylation phenotype than the complete removal of the TET proteins (Supplementary Fig. 2f). Intriguingly, whereas TET1 and TET2 prominently associate with sites of active demethylation (Supplementary Fig. 2g), we found that the majority of sites hypermethylated in Tet catalytic mutant ESCs

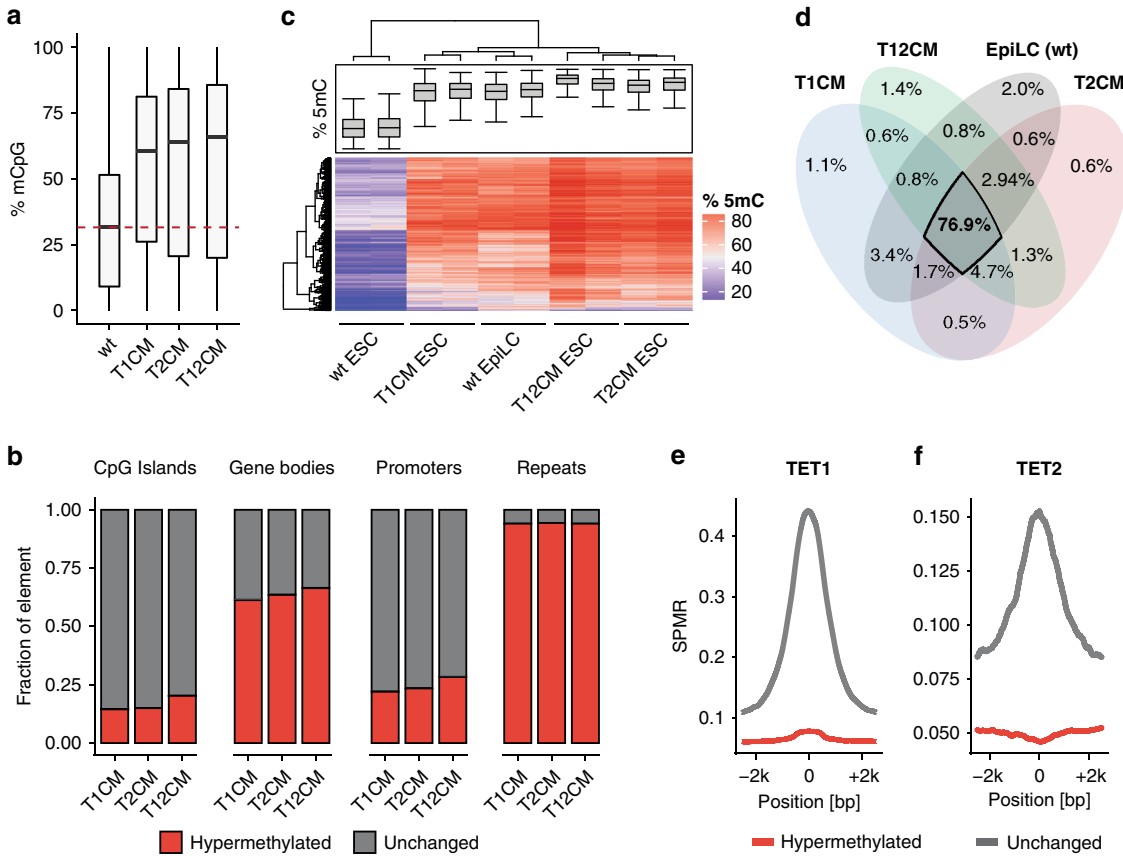

**Fig. 1 TET1 and TET2 prevent hypermethylation of the naïve genome. a** Loss of TET catalytic activity leads to global DNA hypermethylation. Percentage of total 5mC as measured by RRBS. For each genotype, $n = 2$ biologically independent samples per condition. **b** Loss of TET catalytic activity leads to widespread DNA hypermethylation especially at repetitive elements. Relative proportion of DNA hypermethylation ($q$ value < 0.05; absolute methylation difference >20%) at each genomic element in T1CM, T2CM, and T12CM ESCs compared to wt ESCs. **c** Heat map of the hierarchical clustering of the RRBS data depicting the top 2000 most variable 1 kb tiles during differentiation of wt ESCs to EpiLCs with $n = 2$ biologically independent samples per genotype and condition. **d** Venn diagram depicting the overlap of hypermethylated (compared to wt ESCs; $q$ value < 0.05; absolute methylation difference >20%) sites among T1CM, T2CM, and T12CM ESCs and wt EpiLCs. **e, f** TET binding is not associated with DNA hypermethylation in TET mutant ESCs. Occupancy of (**e**) TET1[66] and (**f**) TET2[67] over 1 kb tiles hypermethylated (dark red) or unchanged (dark gray) in T1CM and T2CM ESCs, respectively (SPMR: Signal per million reads). In the boxplots in (**a**) and (**c**), horizontal black lines within boxes represent median values, boxes indicate the upper and lower quartiles, and whiskers extend to the most extreme value within 1.5 x the interquartile range from each hinge. In (**b**) and (**d**), the $q$-values were calculated with a two-sided Wald test followed by $p$-value adjustment using SLIM[209].

are not bound by either enzyme (Fig. 1e, f) suggesting that TET1 and TET2 maintain the hypomethylated state of the naïve methylome by indirect means.

**TET1 and TET2 control *Dppa3* expression in a catalytically dependent manner.** To explore how TET1 and TET2 might indirectly promote demethylation of the naïve genome, we first examined the expression of the enzymes involved in DNA methylation. Loss of TET catalytic activity was not associated with changes in the expression of *Dnmt1*, *Uhrf1*, *Dnmt3a*, and *Dnmt3b* nor differences in UHRF1 protein abundance, indicating the hypermethylation in *Tet* catalytic mutant ESCs is not caused by aberrant upregulation of DNA methylation machinery components (Fig. 2a, Supplementary Fig. 2h). To identify candidate factors involved in promoting global hypomethylation, we compared the transcriptome of hypomethylated wild-type ESCs with those of hypermethylated cells, which included wt EpiLCs as well as T1CM, T2CM, and T12CM ESCs (Fig. 2b). Among the 14 genes differentially expressed in hypermethylated cell lines, the naïve pluripotency factor, *Dppa3* (also known as *Stella* and *Pgc7*), stood out as an interesting candidate due to its reported

involvement in the regulation of global DNA methylation in germ cell development and oocyte maturation[59–62]. In contrast to the core components of the DNA (de)methylation machinery (DNMTs, UHRF1, TETs), which are conserved throughout metazoa, *Dppa3* is only present in mammals, suggesting it might also contribute to the mammal-specific hypomethylation in naïve pluripotency (Fig. 2c).

While normally highly expressed in naïve ESCs and only downregulated upon differentiation[63,64], *Dppa3* was prematurely repressed in T1CM, T2CM, and T12CM ESCs (Fig. 2d). The strongly reduced expression of *Dppa3* in TET mutant ESCs was accompanied by significant hypermethylation of the *Dppa3* promoter (Fig. 2e), consistent with reports demonstrating *Dppa3* to be one of the few pluripotency factors downregulated by promoter methylation upon differentiation in vitro and in vivo[51,63–65]. In contrast to the majority of genomic sites gaining methylation in TET mutant ESCs (Fig. 1e, f), hypermethylation at the *Dppa3* locus occurred at sites bound by both TET1 and TET2 (Fig. 2e)[66,67]. This hypermethylation overlapped with regions at which the TET oxidation product 5-carboxylcytosine (5caC) accumulates in Thymine DNA glycosylase (TDG)-knockdown ESCs (Fig. 2e)[68], indicating that the

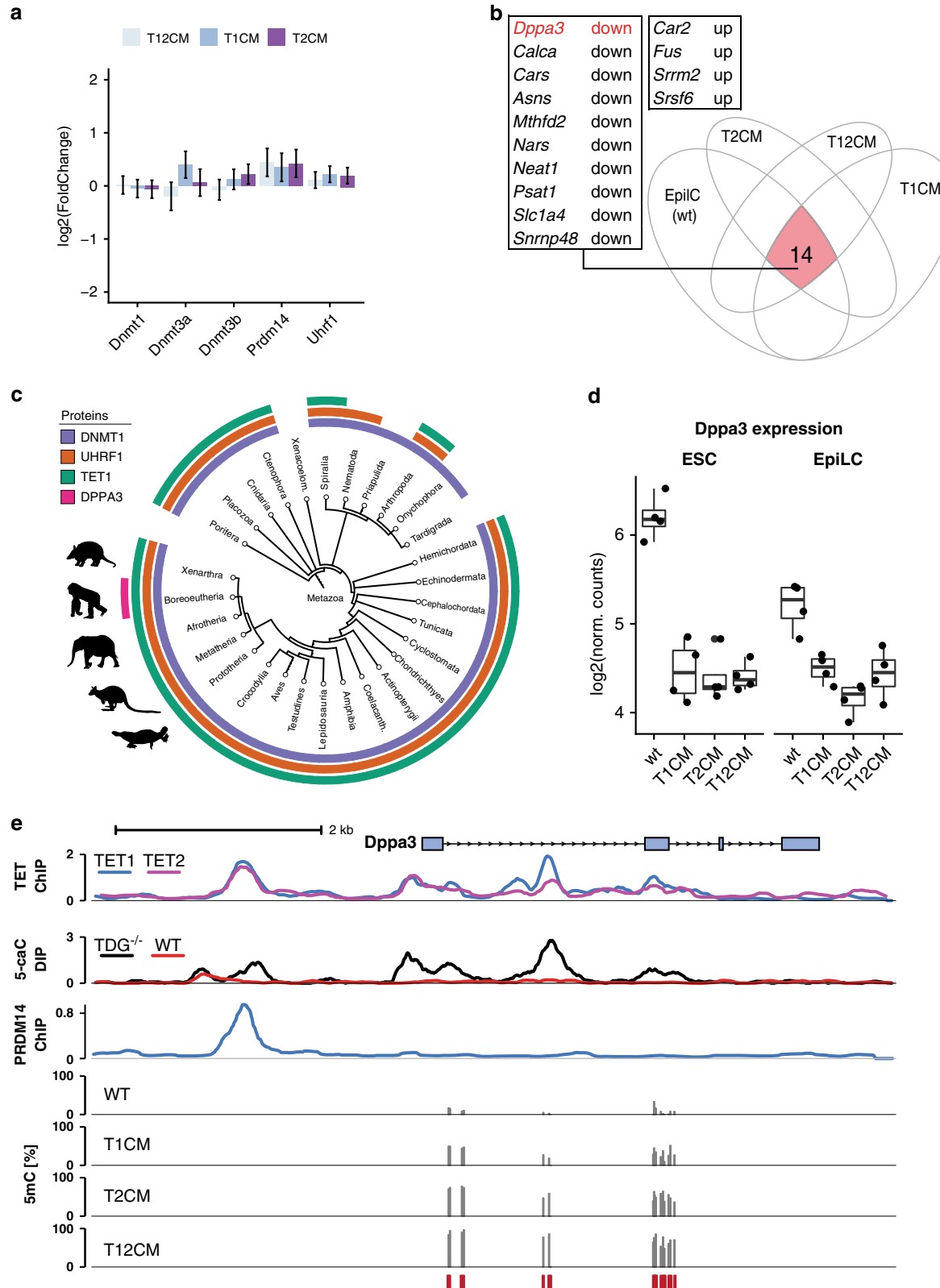

*Dppa3* locus is a direct target of TET/TDG-mediated active DNA demethylation in ESCs. To test whether Dppa3 transcription can be induced by DNA demethylation, we analyzed RNA-seq data from conditional Dnmt1 KO ESCs[69]. In the absence of genome-wide DNA methylation, *Dppa3* levels more than doubled, thus confirming our results that the *Dppa3* promoter is sensitive to DNA methylation (Supplementary Fig. 2i).

In addition, *Dppa3* is also a direct target of PRDM14, a PR domain-containing transcriptional regulator known to promote the DNA hypomethylation associated with naïve pluripotency[50,70–72] (Fig. 2e). PRDM14 has been shown to recruit TET1 and TET2 to sites of active demethylation and establish global hypomethylation in naïve pluripotency[50,54,71–73]. As the expression of *Prdm14* was not altered in *Tet* catalytic mutant ESCs (Fig. 2a), we analyzed PRDM14

**Fig. 2 TET1 and TET2 catalytic activity is necessary for *Dppa3* expression. a** Expression of genes involved in regulating DNA methylation levels in T1CM, T2CM, and T12CM ESCs as assessed by RNA-seq. Expression is given as the $\log_2$ fold-change compared to wt ESCs. Error bars indicate mean ± SD, $n = 4$ biological replicates. No significant changes observable (Likelihood ratio test). **b** *Dppa3* is downregulated upon loss of TET activity and during differentiation. Venn diagram depicting the overlap (red) of genes differentially expressed (compared to wt ESCs; adjusted p < 0.05) in T1CM, T2CM, T12CM ESCs, and wt EpiLCs. **c** Phylogenetic tree of TET1, DNMT1, UHRF1, and DPPA3 in metazoa. **d** *Dppa3* expression levels as determined by RNA-seq in the indicated ESC and EpiLC lines ($n = 4$ biological replicates). **e** TET proteins bind and actively demethylate the *Dppa3* locus. Genome browser view of the *Dppa3* locus with tracks of the occupancy (Signal pileup per million reads; (SPMR)) of TET1[66], TET2[67], and PRDM14[71] in wt ESCs, 5caC enrichment in wt vs. TDG$^{-/-}$ ESCs[68], and 5mC (%) levels in wt, T1CM, T2CM, and T12CM ESCs (RRBS). Red bars indicate CpGs covered by RRBS. In the boxplots in (**d**), horizontal black lines within boxes represent median values, boxes indicate the upper and lower quartiles, and whiskers extend to the most extreme value within 1.5 x the interquartile range from each hinge.

occupancy at the *Dppa3* locus using publicly available ChIP-seq data[71]. This analysis revealed that PRDM14 binds the same upstream region of *Dppa3* occupied by TET1 and TET2 (Fig. 2e). Taken together, these data suggest that TET1 and TET2 are recruited by PRDM14 to maintain the expression of *Dppa3* by active DNA demethylation.

**DPPA3 acts downstream of TET1 and TET2 and is required to safeguard the naïve methylome.** DPPA3 has been reported to both prevent and promote DNA demethylation depending on the cellular and developmental context[59,61,62,74–78]. However, the function of DPPA3 in naïve pluripotency, for which it is a well-established marker gene[63], remains unclear. To investigate the relationship between *Dppa3* expression and DNA hypomethylation in naïve pluripotency, we established *Dppa3* knockout (Dppa3KO) mouse ESCs (Supplementary Fig. 3a–c) and profiled their methylome by RRBS. Deletion of *Dppa3* led to severe global hypermethylation (Fig. 3a), with substantial increases in DNA methylation observed across all analyzed genomic features, including promoters, repetitive sequences, and imprinting control regions (ICRs) (Supplementary Fig. 3d–f). In particular, transposable elements experienced the most extensive gains in DNA methylation, with >90% of detected LINE and ERVs found hypermethylated in Dppa3KO ESCs (Supplementary Fig. 3e).

A principal component analysis of the RRBS data revealed that Dppa3KO ESCs clustered closer to wt EpiLCs and *Tet* catalytic mutant ESCs rather than wt ESCs (Fig. 3b). Furthermore, we observed a striking overlap of hypermethylated CpGs between *Tet* catalytic mutant and Dppa3KO ESCs (Fig. 3c), suggesting that DPPA3 and TETs promote demethylation at largely the same targets. A closer examination of the genomic distribution of overlapping hypermethylation in *Tet* catalytic mutant and Dppa3KO ESCs revealed that the majority (~90%) of hypermethylated events within repetitive elements are common to both cell lines (Fig. 3d and Supplementary Fig. 3g–j) and are globally correlated with heterochromatic histone modifications (Supplementary Fig. 3k). In contrast, only half of the observed promoter hypermethylation among all cell lines was dependent on DPPA3 (classified as "common", Fig. 3d and Supplementary Fig. 3h–j). This allowed us to identify a set of strictly TET-dependent promoters (N = 1573) (Fig. 3d, Supplementary Fig. 3i and Supplementary Data 1), which were enriched for developmental genes (Fig. 3e and Supplementary Data 2). Intriguingly, these TET-specific promoters contained genes (such as *Pax6*, *Foxa1*, and *Otx2*) that were recently shown to be conserved targets of TET-mediated demethylation during *Xenopus*, zebrafish, and mouse development[79].

DPPA3 appeared to act downstream of TETs as the global increase in DNA methylation in Dppa3KO ESCs was not associated with a reduction in 5hmC levels nor with a downregulation of TET family members (Fig. 3f and Supplementary Fig. 3l). In support of this notion, inducible overexpression of *Dppa3* (Supplementary Fig. 3m–o) completely rescued the observed hypermethylation phenotype at LINE-1

elements in T1CM as well as T2CM ESCs and resulted in a significant reduction of hypermethylation in T12CM cells (Fig. 3g). Strikingly, prolonged induction of *Dppa3* resulted in hypomethylation in wild-type as well as T1CM ESCs (Fig. 3g). Collectively, these results show that TET1 and TET2 activity contributes to genomic hypomethylation in naïve pluripotency by both direct and indirect pathways. Whereas direct and active demethylation protects a limited but key set of promoters, global DNA demethylation occurs as an indirect effect of *Dppa3* activation.

**TET-dependent expression of DPPA3 regulates UHRF1 subcellular distribution and controls DNA methylation maintenance in embryonic stem cells.** To investigate the mechanism underlying the regulation of global DNA methylation patterns by DPPA3, we first generated an endogenous DPPA3-HALO fusion ESC line to monitor the localization of DPPA3 throughout the cell cycle (Supplementary Fig. 4a, c). Previous studies have shown that DPPA3 binds H3K9me2[77] and that in oocytes its nuclear localization is critical to inhibit the activity of UHRF1[62], a key factor for maintaining methylation. Expecting a related mechanism to be present in ESCs, we were surprised to find that DPPA3 primarily localized to the cytoplasm of ESCs (Fig. 4a). Although present in the nucleus, DPPA3 was far more abundant in the cytoplasmic fraction (Supplementary Fig. 4e). Furthermore, DPPA3 did not bind to mitotic chromosomes indicating a low or absent chromatin association of DPPA3 in ESCs (Fig. 4a). To further understand the mechanistic basis of DPPA3-dependent DNA demethylation in ESCs, we performed FLAG-DPPA3 pulldowns followed by liquid chromatography tandem mass spectrometry (LC-MS/MS) to profile the DPPA3 interactome in naïve ESCs. Strikingly, among the 303 significantly enriched DPPA3 interaction partners identified by mass spectrometry, we found both UHRF1 and DNMT1 (Fig. 4b and Supplementary Data 3), the core components of the DNA maintenance methylation machinery[80,81]. A reciprocal immunoprecipitation of UHRF1 confirmed its interaction with DPPA3 in ESCs (Supplementary Fig. 4g). Moreover, GO analysis of the top 131 interactors of DPPA3 in ESCs showed the two most enriched GO terms to be related to DNA methylation (Supplementary Data 4). These findings are consistent with previous studies implicating DPPA3 in the regulation of maintenance methylation in other cellular contexts[60,62]. We also detected multiple members of the nuclear transport machinery in our DPPA3 interactome (highlighted in purple, Fig. 4b and Supplementary Data 3), which prompted us to investigate whether DPPA3 influences the subcellular localization of UHRF1. Surprisingly, biochemical fractionation experiments revealed UHRF1 to be present in both the nucleus and cytoplasm of naïve wt ESCs (Supplementary Fig. 4f). Despite comparable total UHRF1 protein levels in wt and Dppa3KO ESCs (Supplementary Fig. 4h), loss of DPPA3 completely abolished the cytoplasmic fraction of UHRF1 (Supplementary Fig. 4f).

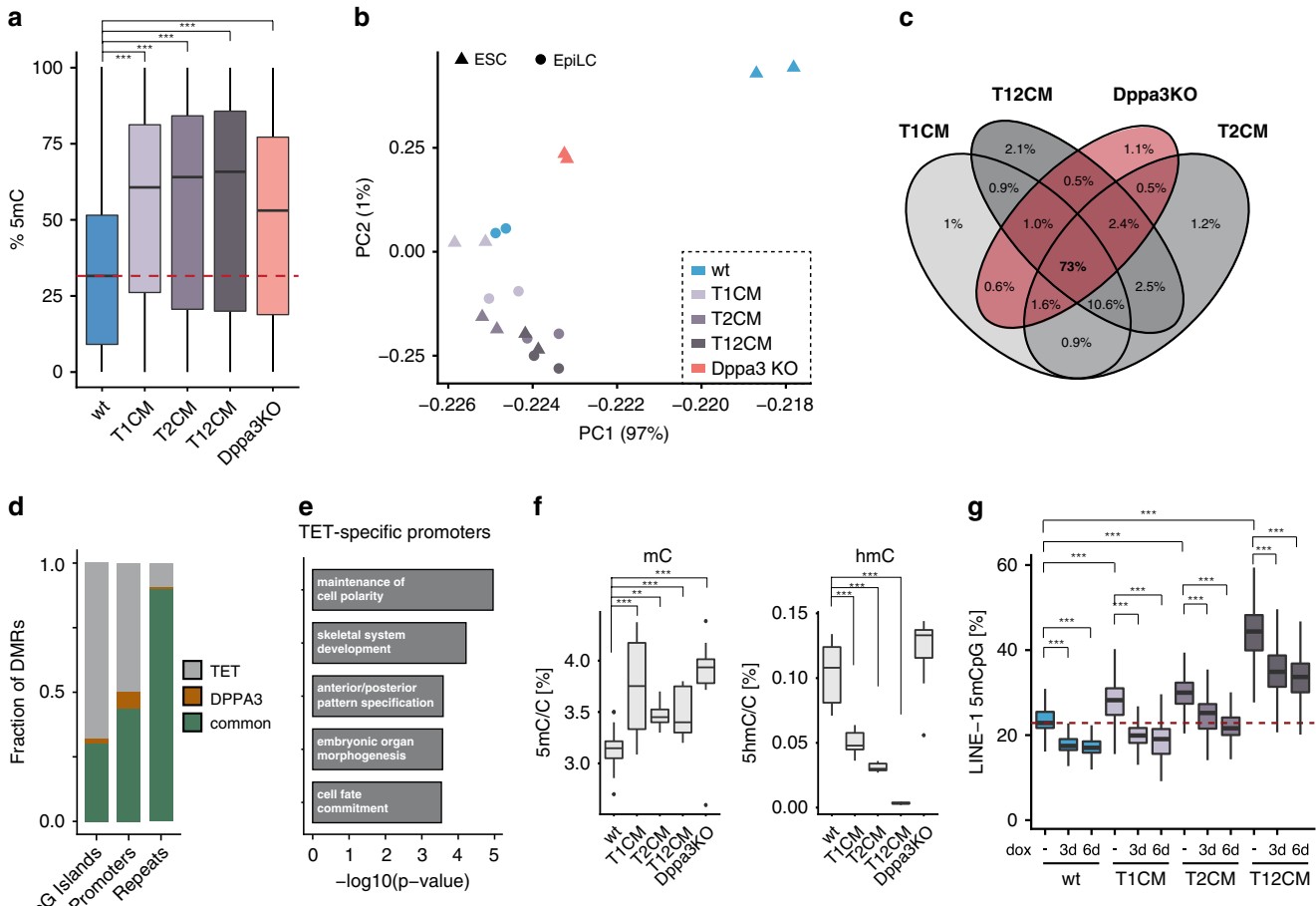

**Fig. 3 DPPA3 acts downstream of TET1 and TET2 to establish and preserve global hypomethylation. a** *Dppa3* loss results in global hypermethylation. Percentage of total 5mC as measured by RRBS using *n* = 2 biologically independent samples per condition. **b** *Dppa3* prevents the premature acquisition of a primed methylome. Principal component (PC) analysis of RRBS data from wt, T1CM, T2CM, and T12CM ESCs and EpiLCs and Dppa3KO ESCs. **c** DPPA3 and TET proteins promote demethylation of largely similar targets. Venn Diagram depicting the overlap of hypermethylated sites among T1CM, T2CM, T12CM, and Dppa3KO ESCs. **d** *Dppa3* protects mostly repeats from hypermethylation. Fraction of hypermethylated genomic elements classified as TET-specific (only hypermethylated in TET mutant ESCs), DPPA3-specific (only hypermethylated in Dppa3KO ESCs), or common (hypermethylated in TET mutant and Dppa3KO ESCs). **e** Gene ontology (GO) terms associated with promoters specifically dependent on TET activity; adjusted p-values calculated using a two-sided Fisher's exact test followed by Benjamini-Hochberg correction for multiple testing. **f** TET activity remains unaffected in Dppa3KO ESCs. DNA modification levels for 5-methylcytosine (5mC) and 5-hydroxymethylcytosine (5hmC) as measured by mass spectrometry (LC-MS/MS) in wt (*n* = 24), T1CM (*n* = 8), T2CM (*n* = 12), T12CM (*n* = 11), Dppa3KO (*n* = 12) mESC biological replicates. **g** *Dppa3* expression can rescue the hypermethylation in TET mutant ESCs. DNA methylation levels at LINE-1 elements (%) as measured by bisulfite sequencing 0, 3, or 6 days after doxycycline (dox) induction of *Dppa3* expression using *n* = 2 replicates per condition. The dashed red line indicates the median methylation level of wt ESCs. In the boxplots in (**a**, **f** and **g**), horizontal black lines within boxes represent median values, boxes indicate the upper and lower quartiles, and whiskers extend to the most extreme value within 1.5 x the interquartile range from each hinge. In (**a**, **f**, and **g**), p-values were calculated using Welch's two-sided t-test: ***p < 2e−16. Source data are provided as a Source Data file.

As maintenance DNA methylation critically depends on the correct targeting and localization of UHRF1 within the nucleus[82–85], we asked whether TET-dependent regulation of DPPA3 might affect the subnuclear distribution of UHRF1. To this end, we tagged endogenous UHRF1 with GFP in wild-type (U1G/wt) as well as Dppa3KO and T12CM ESCs (U1G/Dppa3KO and U1G/T12CM, respectively) enabling us to monitor UHRF1 localization dynamics in living cells (Supplementary Fig. 4b, d). Whereas UHRF1-GFP localized to both the nucleus and cytoplasm of wt ESCs, UHRF1-GFP localization was solely nuclear in Dppa3KO and T12CM ESCs (Supplementary Fig. 4i, j). In addition, UHRF1 appeared to display a more diffuse localization in wt ESCs compared to Dppa3KO and T12CM ESCs, in which we observed more focal patterning of UHRF1 particularly at heterochromatic foci

(Supplementary Fig. 4i). To quantify this observation, we calculated the coefficient of variation (CV) of nuclear UHRF1-GFP among wt, Dppa3KO, and T12CM ESCs. The CV of a fluorescent signal correlates with its distribution, with low CV values reflecting more homogenous distributions and high CV values corresponding to more heterogeneous distributions[86,87]. Indeed, the pronounced focal accumulation of UHRF1-GFP observed in Dppa3KO and T12CM ESCs corresponded with a highly significant increase in the CV values of nuclear UHRF1-GFP compared with wt ESCs (Supplementary Fig. 4i, j).

To assess whether these differences in nuclear UHRF1 distribution reflected altered chromatin binding, we used fluorescence recovery after photobleaching (FRAP) to study the dynamics of nuclear UHRF1-GFP in wt, Dppa3KO, and T12CM ESCs. Our FRAP analysis revealed markedly increased UHRF1

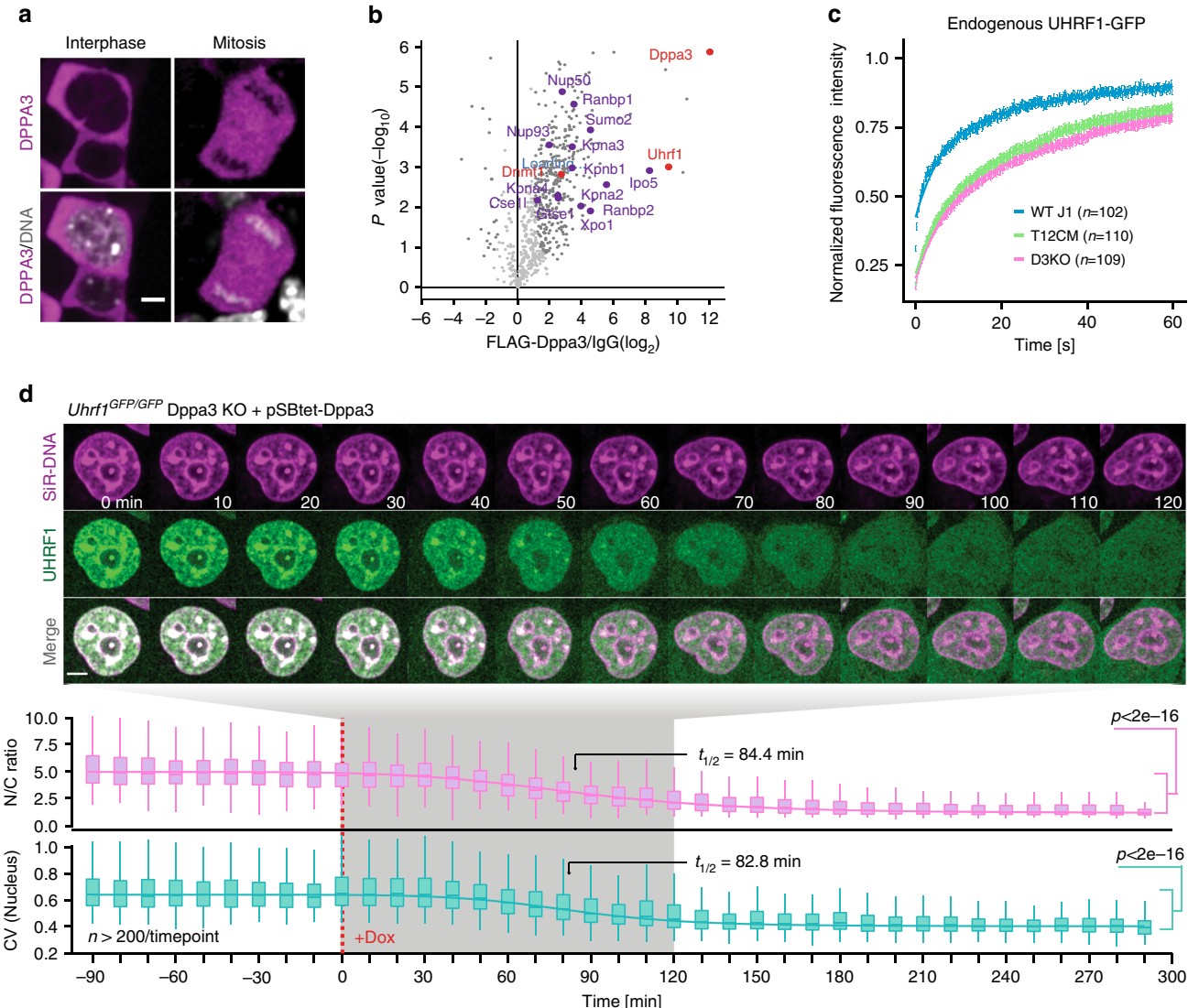

**Fig. 4 TET-dependent expression of DPPA3 alters UHRF1 localization and chromatin binding in naïve ESCs. a** Localization of endogenous DPPA3-HALO in live ESCs counterstained with SiR-Hoechst (DNA). Representative result, $n \geq 4$. Scale bar: 5 µm. **b** Volcano plot from DPPA3-FLAG pulldowns in ESCs. Dark gray dots: significantly enriched proteins. Red dots: proteins involved in DNA methylation regulation. Purple dots: proteins involved in nuclear transport. anti-FLAG antibody: $n = 3$ biological replicates, IgG control antibody: $n = 3$ biological replicates. Statistical significance determined by performing a Student's $t$ test with a permutation-based FDR of 0.05 and an additional constant S0 = 1. **c** FRAP analysis of endogenous UHRF1-GFP. Each genotype comprises the combined single-cell data from two independent clones acquired in two independent experiments. **d** Localization dynamics of endogenous UHRF1-GFP in response to *Dppa3* induction in U1G/D3KO + pSBtet-D3 ESCs with confocal timelapse imaging over 8 h (10 min intervals). $t = 0$ corresponds to start of *Dppa3* induction with doxycycline (+Dox). (*top panel*) Representative images of UHRF1-GFP and DNA (SiR-Hoechst stain) throughout confocal timelapse imaging. Scale bar: 5 µm. (middle panel) Nucleus to cytoplasm ratio (N/C ratio) of endogenous UHRF1-GFP signal. (bottom panel) Coefficient of variance (CV) of endogenous UHRF1-GFP intensity in the nucleus. (*middle and bottom panel*) N/C ratio and CV values: measurements in $n > 200$ single cells per time point (precise values can be found in the Source Data file), acquired at $n = 16$ separate positions. Curves represent fits of four parameter logistic (4PL) functions to the N/C ratio (pink line) and CV (green line) data. Live-cell imaging was repeated three times with similar results. In (**c**), the mean fluorescence intensity of $n$ cells (indicated in the plots) at each timepoint are depicted as shaded dots. Error bars indicate mean ± SEM. Curves (solid lines) indicate double-exponential functions fitted to the FRAP data. In the boxplots in (**d**), darker horizontal lines within boxes represent median values. The limits of the boxes indicate upper and lower quartiles, and whiskers extend to the most extreme value within 1.5 x the interquartile range from each hinge. *P*-values based on Welch's two-sided $t$ test. Source data are provided as a Source Data file.

chromatin binding in both Dppa3KO and T12CM ESCs as demonstrated by the significantly slower recovery of UHRF1-GFP in these cell lines compared to wt ESCs (Fig. 4c and Supplementary Fig. 4k, l). These data confirmed the notion that the more pronounced focal patterning of nuclear UHRF1 observed in Dppa3KO and T12CM ESCs (Supplementary Fig. 4i, j) was indeed a consequence of increased UHRF1 chromatin binding. Interestingly, although strongly reduced compared to wt

ESCs, UHRF1 mobility was slightly higher in T12CM ESCs than Dppa3KO ESCs, consistent with a severe but not total loss of *Dppa3* in the absence of TET activity (Supplementary Fig. 4m). Induction of ectopic *Dppa3* rescued the cytoplasmic fraction of UHRF1 (N/C ratio: Fig. 4d) as well as the diffuse localization of nuclear UHRF1 in Dppa3KO ESCs (CV: Fig. 4d), which reflected a striking increase in the mobility of residual nuclear UHRF1-GFP as assessed by FRAP (Supplementary Figs. 4n and 5a, b).

Our analysis also revealed that the nuclear export of UHRF1 and the inhibition of UHRF1 chromatin binding caused by *Dppa3* induction occur with almost identical kinetics (N/C $t_{1/2} = 84.4$ min; CV $t_{1/2} = 82.8$) (Fig. 4d). UHRF1 is required for the proper targeting of DNMT1 to DNA replication sites and therefore essential for DNA methylation maintenance[80,81]. We observed a marked reduction of both UHRF1 and DNMT1 at replication foci upon induction of *Dppa3*, indicating that DPPA3 promotes hypomethylation in naïve ESCs by impairing DNA methylation maintenance (Supplementary Fig. 5c, d). Ectopic expression of DPPA3 not only altered the subcellular distribution of endogenous UHRF1 in mouse ESCs (Fig. 4d and Supplementary Fig. 5e) but also in human ESCs suggesting evolutionary conservation of this mechanism among mammals (Supplementary Fig. 5f, g). Collectively, our results demonstrate that TET proteins control both the subcellular localization and chromatin binding of UHRF1 in naïve ESCs via the regulation of DPPA3 levels. Furthermore, these data show that DPPA3 is both necessary and sufficient for ensuring the nucleocytoplasmic translocation, diffuse nuclear localization, and attenuated chromatin binding of UHRF1 in ESCs.

**DPPA3-mediated demethylation is achieved via inhibition of UHRF1 chromatin binding and attenuated by nuclear export**. Our results demonstrated that *Dppa3* induction causes UHRF1 to be released from chromatin and exported to the cytoplasm near simultaneously (Fig. 4d, Supplementary Figs. 4n and 5a, b). In principle, either a reduction in the nuclear concentration of UHRF1 or the impairment of UHRF1 chromatin binding alone would suffice to compromise effective maintenance DNA methylation[84,88]. To dissect the contribution of these distinct modes of disrupting UHRF1 activity to DPPA3-mediated DNA demethylation in naïve ESCs, we generated inducible *Dppa3-mScarlet* expression cassettes (Supplementary Fig. 6a) harboring mutations to residues described to be critical for its nuclear export (ΔNES)[61] and the interaction with UHRF1 (KRR and R107E)[62], as well as truncated forms of DPPA3 found in zygotes, 1-60 and 61-150[78] (Fig. 5a). After introducing these *Dppa3* expression cassettes into U1GFP/Dppa3KO ESCs, we used live-cell imaging to track each DPPA3 mutant's localization and ability to rescue the Dppa3KO phenotype (Fig. 5b). DPPA3-ΔNES and DPPA3 61-150, which both lacked a functional nuclear export signal, were retained in the nucleus (Fig. 5b). In contrast DPPA3-WT as well as the DPPA3-KRR, DPPA3-R107E, and DPPA3 1-60 mutants localized primarily to the cytoplasm (Fig. 5b), closely mirroring the localization of endogenous DPPA3 in naïve ESCs (Fig. 4a). However, all tested DPPA3 mutants failed to efficiently reestablish nucleocytoplasmic translocation of UHRF1 (Fig. 5b and Supplementary Fig. 6b), indicating that the DPPA3-UHRF1 interaction and nuclear export of DPPA3 are both required for the shuttling of UHRF1 from the nucleus to the cytoplasm in naïve ESCs.

Nevertheless, DPPA3-ΔNES and DPPA3 61-150 managed to significantly disrupt the focal pattern and heterochromatin association of UHRF1 within the nucleus, with DPPA3-ΔNES causing a more diffuse localization of nuclear UHRF1 than DPPA3-WT (Fig. 5b and Supplementary Fig. 6c). In contrast, the loss or mutation of residues critical for its interaction with UHRF1 compromised DPPA3's ability to effectively restore the diffuse localization of nuclear UHRF1 (Fig. 5b and Supplementary Fig. 6c). FRAP analysis revealed that the disruption or deletion of the UHRF1 interaction interface (DPPA3-KRR, DPPA3-R107E, DPPA3 1-60) severely diminished the ability of DPPA3 to release UHRF1 from chromatin (Fig. 5c and Supplementary Fig. 6f–k). On the other hand, the C-terminal half of DPPA3, lacking a nuclear export signal but retaining

UHRF1 interaction, came close to fully restoring the mobility of UHRF1 (Fig. 5c and Supplementary Fig. 6i–k). DPPA3-ΔNES mobilized UHRF1 to a greater extent than DPPA3-WT (Fig. 5c and Supplementary Fig. 6d, e, j, k), suggesting that active nuclear export might antagonize DPPA3-mediated inhibition of UHRF1 chromatin binding. Supporting this notion, chemical inhibition of nuclear export using leptomycin-B (LMB) significantly enhanced the inhibition of UHRF1 chromatin binding in U1G/D3KO ESCs expressing DPPA3-WT (Supplementary Fig. 5h–k). Taken together, our data show that the efficiency of DPPA3-dependent release of UHRF1 from chromatin requires its interaction with UHRF1 but not its nuclear export.

To further address the question whether the nucleocytoplasmic translocation of UHRF1 and impaired UHRF1 chromatin binding both contribute to DPPA3-mediated inhibition of DNA methylation maintenance, we assessed the ability of each DPPA3 mutant to rescue the hypermethylation of LINE-1 elements in Dppa3KO ESCs (Fig. 5d). Strikingly, DPPA3-ΔNES fully rescued the hypermethylation and achieved a greater loss of DNA methylation than DPPA3-WT, whereas DPPA3 mutants lacking the residues important for UHRF1 binding failed to restore low methylation levels (Fig. 5d). Overall, the ability of each DPPA3 mutant to reduce DNA methylation levels closely mirrored the extent to which each mutant impaired UHRF1 chromatin binding (Fig. 5c and Supplementary Fig. 6d–k). In line with the high mobility of UHRF1 achieved by the DPPA3-ΔNES, (Fig. 5c, Supplementary Figs. 5h–k and 6d, e, j, k), nuclear export is not only dispensable for DPPA3-mediated demethylation, but attenuates the ability of DPPA3 to inhibit maintenance methylation (Fig. 5d). Collectively, our findings demonstrate the inhibition of UHRF1 chromatin binding, as opposed to nucleocytoplasmic translocation of UHRF1, to be the primary mechanism by which DPPA3 drives hypomethylation in naïve ESCs.

**DPPA3 binds nuclear UHRF1 with high affinity prompting its release from chromatin in ESCs**. Next, we set out to investigate the mechanistic basis of DPPA3's ability to inhibit UHRF1 chromatin binding in naïve ESCs. DPPA3 has been reported to specifically bind H3K9me2[77], a histone modification critical for UHRF1 targeting[84,89,90]. These prior findings led us to consider two possible mechanistic explanations for DPPA3-mediated UHRF1 inhibition in naïve ESCs: (1) DPPA3 blocks access of UHRF1 to chromatin by competing in binding to H3K9me2, (2) DPPA3 directly or indirectly binds to UHRF1 and thereby prevents it from accessing chromatin.

To simultaneously assess the dynamics of both UHRF1 and DPPA3 under physiological conditions in live ES cells, we employed raster image correlation spectroscopy with pulsed interleaved excitation (PIE-RICS) (Fig. 6a). RICS is a confocal imaging method that measures the diffusive properties of fluorescently labeled molecules, and thereby also their binding, in living cells. Using images acquired on a laser scanning confocal microscope, spatiotemporal information of fluorescently labeled proteins can be extracted from the shape of the spatial autocorrelation function (SACF). A diffusive model is fitted to the SACF which yields the average diffusion coefficient, the concentration, and the fraction of quickly diffusing and slowly diffusing (in this case, bound) molecules[91]. If two proteins are labeled with distinct fluorophores and imaged simultaneously with separate detectors, the extent of their interaction can be extracted from the cross-correlation of their fluctuations using cross-correlation RICS (ccRICS) (Fig. 6a)[92].

We first measured the mobility of DPPA3-mScarlet variants expressed in U1GFP/D3KOs (Supplementary Fig. 7a, b). The RICS analysis revealed that, over the timescale of the

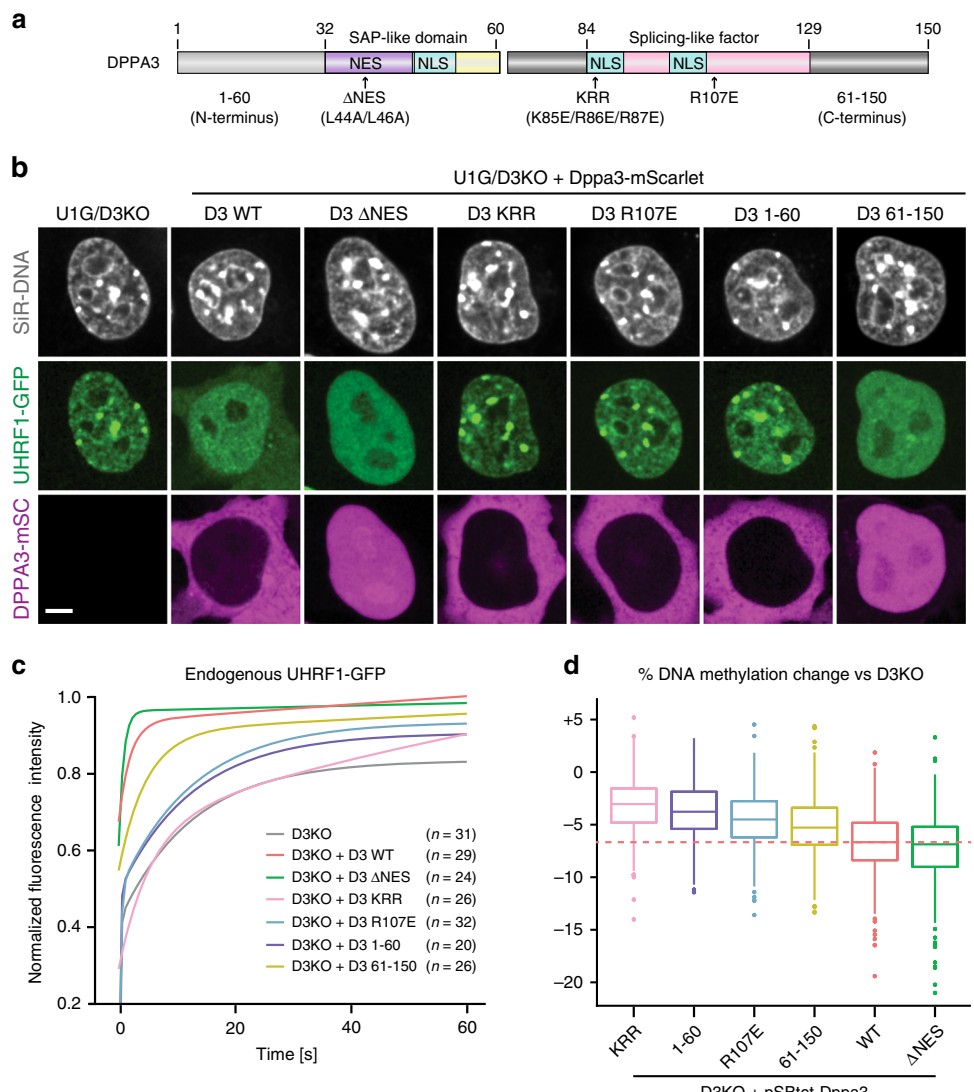

**Fig. 5 DPPA3-mediated demethylation is achieved via inhibition of UHRF1 chromatin binding and attenuated by nuclear export. a** Schematic illustration of murine DPPA3 with the nuclear localization signals (NLS), nuclear export signal (NES), and predicted domains (SAP-like and splicing factor-like[210]) annotated. For the DPPA3 mutant forms used in this study, point mutations are indicated with arrows (ΔNES, KRR, R107E) and the two truncations are denoted by the middle break (1–60, left half; 61–150, right half). **b, c** Nuclear export and the N-terminus of DPPA3 are dispensable for disrupting focal UHRF1 patterning and chromatin binding in ESCs. **b** Representative confocal images illustrating the localization of endogenous UHRF1-GFP and the indicated mDPPA3-mScarlet fusions in live U1G/D3KO + pSB-D3-mSC ESCs after doxycycline induction. DNA counterstain: SiR-Hoechst. Scale bar: 5 μm. **c** FRAP analysis of endogenous UHRF1-GFP in U1G/D3KO ESCs expressing the indicated mutant forms of DPPA3. FRAP Curves (solid lines) indicate double-exponential functions fitted to the FRAP data acquired from n cells (shown in the plots). For single-cell FRAP data and additional quantification, see Supplementary Fig. 6d–k. **d** DPPA3-mediated inhibition of UHRF1 chromatin binding is necessary and sufficient to promote DNA demethylation. Percentage of DNA methylation change at LINE-1 elements (%) in D3KO ESCs after induction of the indicated mutant forms of *Dppa3* as measured by bisulfite sequencing of n = 4 biological replicates. In the boxplot in (**d**), horizontal lines within boxes represent median values, boxes indicate the upper and lower quartiles, whiskers extend to the most extreme value within 1.5 x the interquartile range from each hinge, and dots indicate outliers. Source data are provided as a Source Data file.

measurements, nuclear DPPA3-WT was predominantly unbound from chromatin and freely diffusing through the nucleus at a rate of $7.18 \pm 1.87$ μm$^2$/s (Supplementary Fig. 7f). The fraction of mobile DPPA3-mScarlet molecules was measured to be $88.4 \pm 5.2\%$ (Fig. 6f), validating the globally weak binding inferred from ChIP-Seq profiles[76]. These mobility parameters were largely unaffected by disruption of the UHRF1 interaction, with the DPPA3-KRR mutant behaving similarly to wild-type DPPA3 (Fig. 6f and Supplementary Fig. 7f). To rule out a potential competition between UHRF1 and DPPA3 for H3K9me2 binding, we next used RICS to determine if DPPA3 dynamics are altered

in the absence of UHRF1. For this purpose, we introduced the DPPA3-WT-mScarlet cassette into Uhrf1KO (U1KO) ESCs[93], in which free eGFP is expressed from the endogenous *Uhrf1* promoter (Supplementary Fig. 7c). However, neither the diffusion rate nor the mobile fraction of DPPA3 were appreciably altered in cells devoid of UHRF1, suggesting the high fraction of unbound DPPA3 to be unrelated to the presence of UHRF1 (Fig. 6f and Supplementary Fig. 7f). Overall, our RICS data demonstrate that, in contrast to zygotes[77], DPPA3 in ESCs lacks a strong capacity for chromatin binding, and, as such, is not engaged in competition with UHRF1 for chromatin binding.

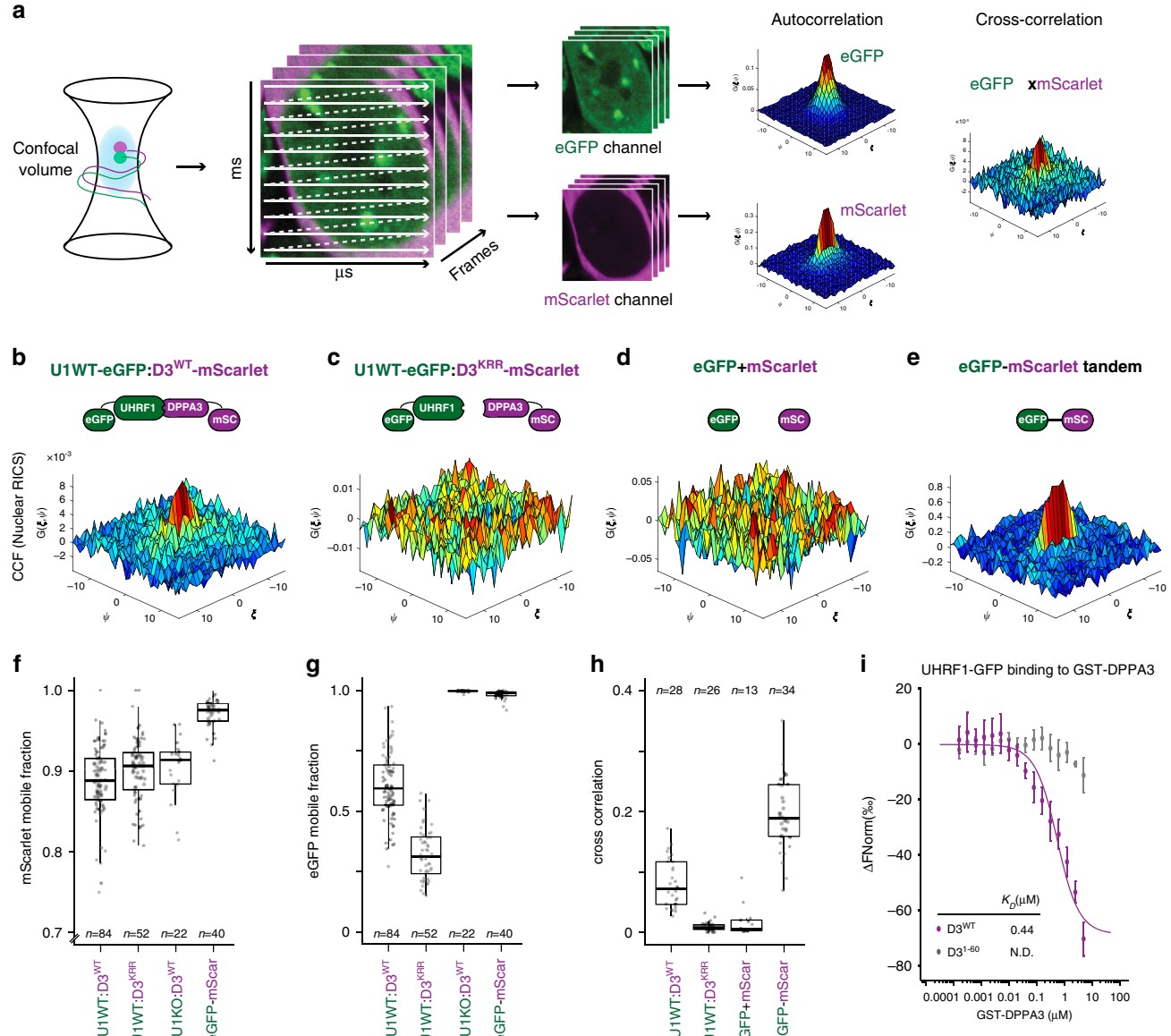

**Fig. 6 DPPA3 binds nuclear UHRF1 with high affinity prompting its release from chromatin in naïve ESCs. a** Overview of RICS and ccRICS. Confocal image series are acquired on a laser scanning confocal microscope, containing spatiotemporal fluorescence information on the microsecond and millisecond timescales. A spatial autocorrelation function (SACF) is calculated from the fluorescence image and fit to a diffusive model. The cross-correlation of intensity between two channels is used to estimate the co-occurrence of two fluorescent molecules in live cells. The mean cross-correlation of the fluctuations is calculated and shown in the 3D plot color-coded according to the correlation value. **b–e** Representative plots of the spatial cross-correlation function (SCCF) between the depicted fluorescent molecules in cells from each cell line measured: (**b**) wild-type (U1WT:D3$^{WT}$) and (**c**) K85E/R85E/R87E DPPA3 mutant (U1WT:D3$^{KRR}$), and control ESCs expressing (**d**) free eGFP, free mScarlet (eGFP + mScarlet) and (**e**) an eGFP-mScarlet tandem fusion (eGFP-mScarlet). **f, g** Mobile fraction of (**f**) mScarlet and (**g**) eGFP species in the cell lines depicted in (**b, c,** and **e**) and in Uhrf1KO ESCs expressing free eGFP and wild-type DPPA3-mScarlet (U1KO:D3$^{WT}$). The mobile fraction was derived from a two-component model fit of the autocorrelation function. Data are pooled from three (U1WT:D3$^{WT}$, U1WT:D3$^{KRR}$) or two (U1KO:D3$^{WT}$, eGFP-mScar) independent experiments. **h** Mean cross-correlation values of mobile eGFP and mScarlet measured in the cell lines depicted in (**b–e**). The spatial lag in the x-dimension (sensitive to fast fluctuations) is indicated by $\xi$, and the spatial lag in the y-dimension (sensitive to slower fluctuations) is indicated by $\psi$. Data are pooled from two independent experiments. **i** Microscale thermophoresis measurements of UHRF1-eGFP binding to GST-DPPA3 WT (D3$^{WT}$) or GST-DPPA3 1–60 (D3$^{1-60}$). Error bars indicate the mean ± SEM of $n = 2$ technical replicates from $n = 4$ independent experiments. In (**f–h**), each data point represents the measured and fit values from a single cell where $n =$ number of cells measured (indicated in the plots). In the boxplots, darker horizontal lines within boxes represent median values. The limits of the boxes indicate the upper and lower quartiles; the whiskers extend to the most extreme value within 1.5 x the interquartile range from each hinge. Source data are provided as a Source Data file.

We next used RICS to analyze the dynamics of UHRF1-GFP in response to DPPA3 induction (Fig. 6a). In cells expressing DPPA3-KRR, RICS measurements revealed that only 32.4 ± 10% of UHRF1 is mobile, indicating that the majority of UHRF1 is chromatin-bound (Fig. 6g). In contrast, expression of wild-type DPPA3 leads to a dramatic increase in the mobile fraction of UHRF1 (60.6 ± 13.7% mobile fraction for UHRF1) (Fig. 6g and Supplementary Fig. 7g, h). Furthermore, the mobile fraction of

UHRF1 increased as a function of the relative abundance of nuclear DPPA3 to UHRF1 (Supplementary Fig. 7i), thereby indicating a stoichiometric effect of DPPA3 on UHRF1 chromatin binding, consistent with physical interaction. Thus, these results demonstrate that DPPA3 potently disrupts UHRF1 chromatin binding in live ESCs and suggest its interaction with UHRF1 to be critical to do so.

To determine whether such an interaction is indeed present in the nuclei of live ESCs, we performed cross-correlation RICS (ccRICS) (Fig. 6a). We first validated ccRICS in ESCs by analyzing live cells expressing a tandem eGFP-mScarlet fusion (Fig. 6e and Supplementary Fig. 7d), or expressing both freely diffusing eGFP and mScarlet (Fig. 6d and Supplementary Fig. 7e). For the tandem eGFP-mScarlet fusion, we observed a clear positive cross-correlation indicative of eGFP and mScarlet existing in the same complex (Fig. 6e, h), as would be expected for an eGFP-mScarlet fusion. On the other hand, freely diffusing eGFP and mScarlet yielded no visible cross-correlation (Fig. 6d, h), consistent with two independent proteins that do not interact. Upon applying ccRICS to nuclear UHRF1-GFP and DPPA3-mScarlet, we observed a prominent cross-correlation between wild-type DPPA3 and the primarily unbound fraction of UHRF1 (Fig. 6b, h), indicating that mobilized UHRF1 exists in a high affinity complex with DPPA3 in live ESCs. In marked contrast, DPPA3-KRR and UHRF1-GFP failed to exhibit detectable cross-correlation (Fig. 6c, h), consistent with the DPPA3-KRR mutant's diminished capacity to bind[62] and mobilize UHRF1 (Fig. 5c and Supplementary Fig. 6f, j, k). Overall, these findings demonstrate that nuclear DPPA3 interacts with UHRF1 to form a highly mobile complex in naïve ESCs which precludes UHRF1 chromatin binding.

To determine whether the DPPA3-UHRF1 complex identified in vivo (Fig. 6h) corresponds to a high affinity direct interaction, we performed microscale thermophoresis (MST) measurements using recombinant UHRF1-GFP and DPPA3 proteins. MST analysis revealed a direct and high affinity ($K_D$: 0.44 μM) interaction between the DPPA3 WT and UHRF1 (Fig. 6i). No binding was observed for DPPA3 1-60, lacking the residues essential for interaction with UHRF1 (Fig. 6i). In line with the results obtained by ccRICS, these data support the notion that DPPA3 directly binds UHRF1 in vivo. Interestingly, the affinity of the UHRF1-DPPA3 interaction was comparable or even greater than that reported for the binding of UHRF1 to H3K9me3 or unmodified H3 peptides, respectively[94,95].

To better understand how UHRF1 chromatin loading is impaired by its direct interaction with DPPA3, we applied a fluorescent-three-hybrid (F3H) assay to identify the UHRF1 domain bound by DPPA3 in vivo (Supplementary Fig. 7j, k). In short, this method relies on a cell line harboring an array of *lac* operator binding sites in the nucleus at which a GFP-tagged "bait" protein can be immobilized and visualized as a spot. Thus, the extent of recruitment of an mScarlet-tagged "prey" protein to the nuclear GFP spot offers a quantifiable measure of the interaction propensity of the "bait" and "prey" proteins in vivo (Supplementary Fig. 7k)[96]. Using UHRF1-GFP domain deletions as the immobilized bait (Supplementary Fig. 7j, k), we assessed how the loss of each domain affected the recruitment of DPPA3-mScarlet to the GFP spot. In contrast to the other UHRF1 domain deletions, removal of the PHD domain essentially abolished recruitment of DPPA3 to the *lac* spot, demonstrating DPPA3 binds UHRF1 via its PHD domain in vivo (Supplementary Fig. 7l, m). The PHD of UHRF1 is essential for its recruitment to chromatin[88,95,97], ubiquitination of H3 and recruitment of DNMT1 to replication foci[82,83]. Thus, our in vivo results suggest that the high affinity interaction of DPPA3 with UHRF1's PHD domain precludes UHRF1 from binding

chromatin in ESCs, which is also supported by a recent report demonstrating that DPPA3 specifically binds the PHD domain of UHRF1 to competitively inhibit H3 tail binding in vitro[98].

**DPPA3 can inhibit UHRF1 function and drive global DNA demethylation in distantly related, non-mammalian species.** Whereas UHRF1 and TET proteins are widely conserved throughout plants and vertebrates[99,100], both early embryonic global hypomethylation[101] and the *Dppa3* gene are unique to mammals. Consistent with UHRF1's conserved role in maintenance DNA methylation, a multiple sequence alignment of UHRF1's PHD domain showed that the residues critical for the recognition of histone H3 are completely conserved from mammals to invertebrates (Fig. 7a). This prompted us to consider the possibility that DPPA3 might be capable of modulating the function of distantly related UHRF1 homologs outside of mammals. To test this hypothesis, we used amphibian (*Xenopus laevis*) egg extracts to assess the ability of mouse DPPA3 (mDPPA3) to interact with a non-mammalian form of UHRF1. Despite the 360 million years evolutionary distance between mouse and *Xenopus*[102], mDPPA3 not only bound *Xenopus* UHRF1 (xUHRF1) with high affinity (Fig. 7b, c and Supplementary Fig. 8a, b) it also interacted with xUHRF1 specifically via its PHD domain (Supplementary Fig. 8c–e). Moreover, the first 60 amino acids of DPPA3 were dispensable for its interaction with UHRF1 (Supplementary Fig. 8a, b). Interestingly, mutation to R107, reported to be critical for DPPA3's binding with mouse UHRF1[62], diminished but did not fully disrupt the interaction (Supplementary Fig. 8b, e). The R107E mutant retained the ability to bind the xUHRF1-PHD domain but exhibited decreased binding to xUHRF1-PHD-SRA under high-salt conditions (Supplementary Fig. 8e), suggesting that R107E changes the binding mode of mDPPA3 to xUHRF1, rather than inhibiting the complex formation. Considering the remarkable similarity between DPPA3's interaction with mouse and *Xenopus* UHRF1, we reasoned that the ability of DPPA3 to inhibit UHRF1 chromatin binding and maintenance DNA methylation might be transferable to *Xenopus*. To address this, we took advantage of a cell-free system derived from interphase *Xenopus* egg extracts to reconstitute DNA maintenance methylation[82]. Remarkably, recombinant mDPPA3 completely disrupted chromatin binding of both *Xenopus* UHRF1 and DNMT1 without affecting the loading of replication factors such as xCDC45, xRPA2, and xPCNA (Fig. 7d). We determined that the inhibition of xUHRF1 and xDNMT1 chromatin loading only requires DPPA3's C-terminus (61-150 a.a.) and is no longer possible upon mutation of R107 (R107E) (Supplementary Fig. 8h), in line with our results in mouse ESCs (Fig. 5d). Moreover, DPPA3-mediated inhibition of xUHRF1 chromatin loading resulted in the severe perturbation of histone H3 dual-monoubiquitylation (H3Ub2), which is necessary for the recruitment of DNMT1[82,83,103] (Supplementary Fig. 8f). To determine whether mDPPA3 can displace xUHRF1 already bound to chromatin, we first depleted *Xenopus* egg extracts of xDNMT1 to stimulate the hyper-accumulation of xUHRF1 on chromatin[82,104] and then added recombinant mDPPA3 after S-phase had commenced (Supplementary Fig. 8g). Under these conditions, both wild-type mDPPA3 and the 61-150 fragment potently displaced xUHRF1 from chromatin, leading to suppressed H3 ubiquitylation (Supplementary Fig. 8g).

We next assessed the effect of DPPA3 on *Xenopus* maintenance DNA methylation. Consistent with the severe disruption of xDNMT1 chromatin loading, both DPPA3 wild-type and 61–150 effectively abolished replication-dependent DNA methylation in *Xenopus* egg extracts (Fig. 7e). In contrast, DPPA3 1-60 and DPPA3 R107E, which both failed to suppress xUHRF1 and

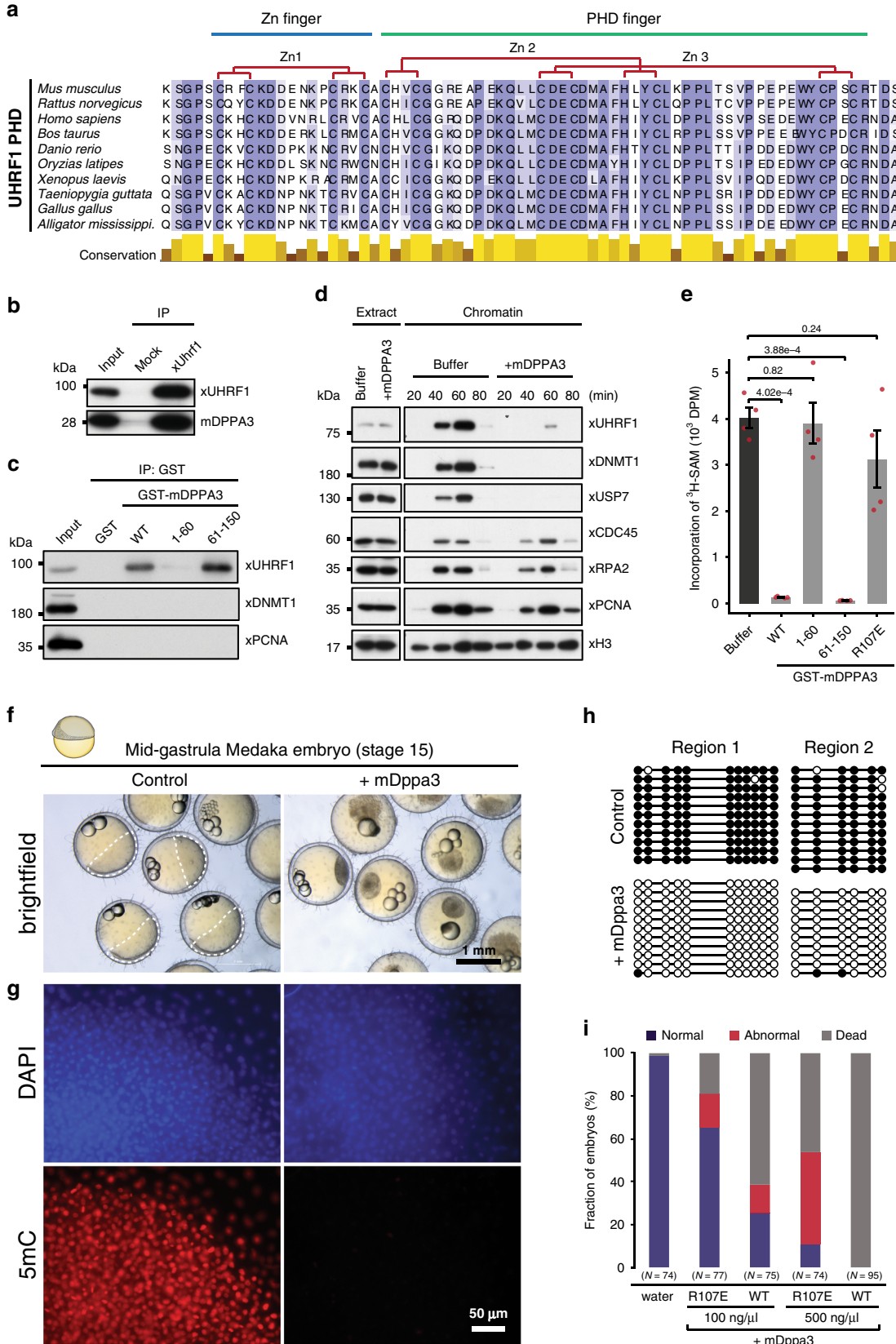

These findings raised the question whether a single protein capable of inhibiting UHRF1 function like DPPA3 could establish a mammalian-like global hypomethylation during the early embryonic development of a non-mammalian organism. To explore this possibility we turned to the biomedical model fish

xDNMT1 binding, did not significantly alter maintenance DNA methylation activity (Fig. 7e and Supplementary Fig. 8d, e). Taken together, our data demonstrate DPPA3 to be capable of potently inhibiting maintenance DNA methylation in a non-mammalian system.

**Fig. 7 DPPA3 evolved in boreoeutherian mammals but also functions in lower vertebrates. a** Protein sequence alignment of the PHD domain of the UHRF1 family. **b** Endogenous xUHRF1 binds mDPPA3. IPs were performed on *Xenopus* egg extracts incubated with FLAG-mDPPA3 using either a control (Mock) or anti-xUHRF1 antibody and then analyzed by immunoblotting using the indicated antibodies. Representative of n = 3 independent experiments. **c** GST-tagged mDPPA3 wild-type (WT), point mutant R107E, and truncations (1–60 and 61–150) were immobilized on GSH beads and incubated with *Xenopus* egg extracts. Bound proteins were analyzed using the indicated antibodies. Representative of n = 3 independent experiments. **d** Sperm chromatin was incubated with interphase *Xenopus* egg extracts supplemented with buffer (+buffer) or GST-mDPPA3 (+mDPPA3). Chromatin fractions were isolated and subjected to immunoblotting using the antibodies indicated. Representative of n = 3 independent experiments. **e** The efficiency of maintenance DNA methylation was assessed by the incorporation of radiolabelled methyl groups from S-[methyl-$^3$H]-adenosyl-L-methionine ($^3$H-SAM) into DNA purified from egg extracts. Disintegrations per minute (DPM). Error bars indicate mean ± SD calculated from n = 4 independent experiments. Depicted p-values based on Welch's two-sided t-test. **f** Representative images of developing mid-gastrula stage embryos (control injection) and arrested, blastula stage embryos injected with *mDppa3*. Injections were performed on one-cell stage embryos and images were acquired ~18 h after fertilization. **g** Immunofluorescence staining of 5mC in control and *mDppa3*-injected medaka embryos at the late blastula stage (~8 h after fertilization). Images are representative of n = 3 independent experiments. DNA counterstain: DAPI,4',6-diamidino-2-phenylindole. **h** Bisulfite sequencing of two intergenic regions (Region 1: chr20:18,605,227-18,605,449, Region 2: chr20:18,655,561-18,655,825) in control and *mDppa3*-injected medaka embryos at the late blastula stage. **i** Percentage of normal, abnormal, or dead medaka embryos. Embryos were injected with wild-type *mDppa3* (WT) or *mDppa3* R107 (R107E) at two different concentrations (100 ng/μl or 500 ng/μl) or water at the one-cell stage and analyzed ~18 h after fertilization. N = number of embryos from n = 3 independent injection experiments. Source data are provided as a Source Data file.

medaka (*Oryzias latipes*), which does not exhibit genome-wide erasure of DNA methylation[105] and diverged from mammals 450 million years ago[102]. We injected medaka embryos with *Dppa3* mRNA at the one-cell stage and then tracked their developmental progression. Remarkably, medaka embryos injected with *Dppa3* failed to develop beyond the blastula stage (Fig. 7f) and exhibited a near-complete elimination of global DNA methylation as assessed by immunofluorescence and bisulfite sequencing (Fig. 7g, h). DPPA3-mediated DNA methylation loss was both dose dependent and sensitive to the R107E mutation, which induced only partial demethylation (Supplementary Fig. 8i). Interestingly, medaka embryos injected with DPPA3 R107E showed far fewer developmental defects than those injected with wild-type DPPA3 (Fig. 7i), suggesting that the embryonic arrest resulting from DPPA3 expression is truly a consequence of the global loss of DNA methylation. Taken together, these results demonstrate that mammalian DPPA3 can inhibit UHRF1 to drive passive demethylation in distant, non-mammalian contexts.

## Discussion

In this study, we aimed to identify the mechanistic basis for the formation of genome-wide DNA hypomethylation unique to mammals. As the role of TET enzymes in active demethylation is well documented[106], we investigated their contribution to the hypomethylated state of naïve ESCs. Mutation of the catalytic core of TET enzymes caused—as expected—a genome-wide increase in DNA methylation but mostly at sites where TET proteins do not bind suggesting a rather indirect mechanism. Among the few genes depending on TET activity for expression in naïve ESCs and downregulated at the transition to EpiLCs was *Dppa3*. Demethylation at the *Dppa3* locus coincides with TET1 and TET2 binding and TDG-dependent removal of oxidized cytosine residues via base excision repair. DPPA3 in turn binds and displaces UHRF1 from chromatin and thereby prevents the recruitment of DNMT1 and the maintenance of DNA methylation in ESCs (see graphic summary in Fig. 8).

Despite long recognized as a marker of naïve ESCs resembling the inner cell mass[63,107], we provide, to our knowledge, the first evidence that DPPA3 directly promotes the genome-wide DNA hypomethylation characteristic of mammalian naïve pluripotency. This unique pathway, in which TET proteins indirectly cause passive demethylation, is based upon two uniquely mammalian innovations: the expression of TET genes in pluripotent cell types[53,79,108] and the evolution of the novel *Dppa3* gene, which is positioned within a conserved pluripotency gene cluster[109] and dependent on TET activity for expression. In support

of this novel pathway for passive demethylation, we found that TET mutant ESCs show a similar phenotype as Dppa3KO cells with respect to UHRF1 chromatin binding and hypermethylation and can be rescued by ectopic expression of *Dppa3*.

Our findings also provide the missing link to reconcile previous, apparently conflicting reports. To date, three distinct mechanisms have been proposed for the global hypomethylation accompanying naïve pluripotency: TET-mediated active demethylation[51,54,58], impaired maintenance DNA methylation[58], and PRDM14-dependent suppression of methylation[50,51,71]. As a downstream target of both TETs and PRDM14 as well as a direct inhibitor of maintenance DNA methylation, DPPA3 mechanistically connects and integrates these three proposed pathways of demethylation (see graphic summary in Fig. 8).

Our mechanistic data showing DPPA3 to displace UHRF1 and DNMT1 from chromatin provide a conclusive explanation for the previous observation that global hypomethylation in naïve ESCs was accompanied by reduced levels of UHRF1 at replication foci[58]. The hypomethylated state of naïve ESCs has also been reported to be dependent on PRDM14[50,71], which has been suggested to promote demethylation by repressing de novo DNA methyltransferases[50,54,71,73]. However, recent studies have demonstrated that the loss of de novo methylation only marginally affects DNA methylation levels in mouse and human ESCs[58,110]. Interestingly, while the loss of *Prdm14* leads to global hypermethylation, it also causes downregulation of *Dppa3*[71,73,111]. Our results suggest that the reported ability of PRDM14 to promote hypomethylation in naïve ESCs largely relies on its activation of the *Dppa3* gene ultimately leading to an inhibition of maintenance methylation.

Of note, other epigenetic pathways such as suppression of H3K9me2 by MAD2L2 as well as eRNA dependent enhancer regulation also have been shown to positively regulate the transcription of *Dppa3*[109,112], and silencing of *Dppa3* has been shown to depend on Lin28a, TBX3, and intact DNA methylation maintenance[113–115]. Taken together, these findings suggest that *Dppa3* is regulated by a complex network of pathways to ensure proper timing of its expression in order to prevent unwanted global DNA demethylation.

The comparison of TET catalytic mutants and Dppa3KO ESCs allows us to distinguish TET-dependent passive DNA demethylation mediated by DPPA3 from *bona fide* active demethylation. We show that TET activity is indispensable for the active demethylation of a subset of promoters in naïve ESCs, especially those of developmental genes. These findings uncover two evolutionary and mechanistically distinct functions of TET catalytic activity.

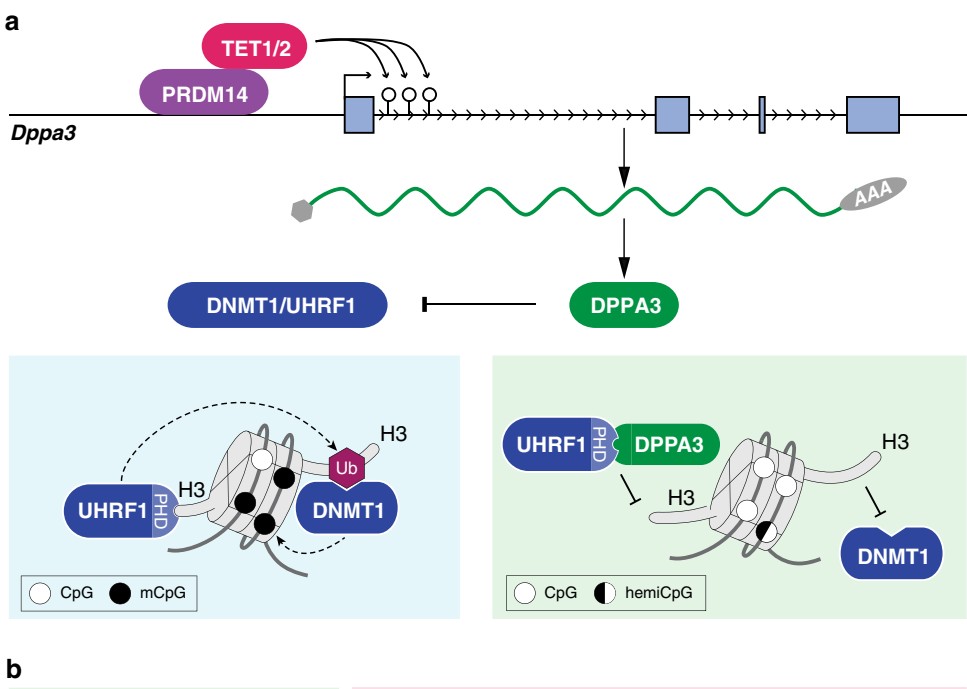

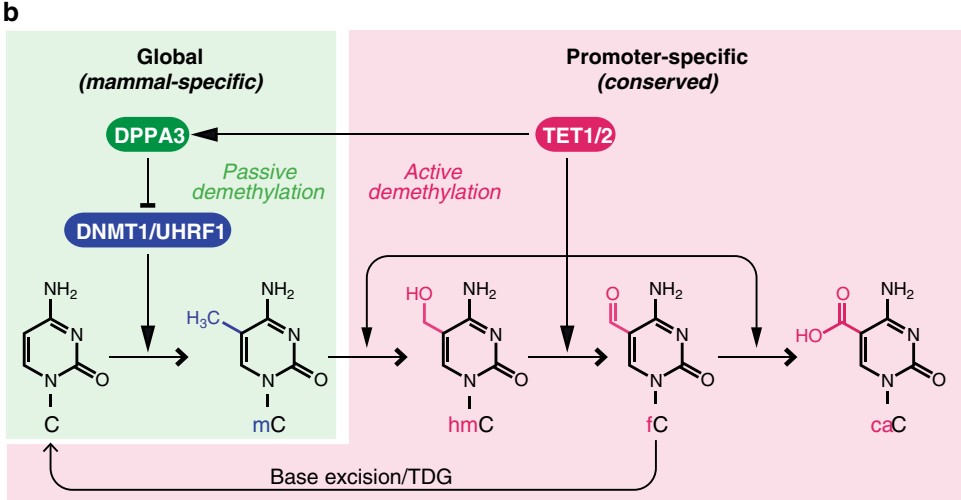

**Fig. 8 Recent evolution of a TET-controlled and DPPA3-mediated pathway of DNA demethylation in boreoeutherian mammals. a** In mammals, TET1 and TET2 are recruited by PRDM14 to the promoter of *Dppa3* where they promote active DNA demethylation and transcription of *Dppa3*. In most cellular contexts, high fidelity maintenance DNA methylation is guaranteed by the concerted activities of UHRF1 and DNMT1 at newly replicated DNA. Both the recruitment and activation of DNMT1 critically depend on the binding and ubiquitination of H3 tails by UHRF1. In naïve pluripotent cells, DPPA3 is expressed and inhibits maintenance DNA methylation by directly binding UHRF1 via its PHD domain and releasing it from chromatin. **b** TET1 and TET2 control DNA methylation levels by two evolutionary and mechanistically distinct pathways. TET-mediated active demethylation regulates focal DNA methylation states e.g. developmental genes and is evolutionarily conserved among vertebrates. The use of TET proteins to promote global demethylation appears to be specific to mammalian pluripotency and mediated by the recently evolved *Dppa3*.

Whereas TET-mediated active demethylation of developmental genes is evolutionarily conserved among vertebrates[79,116–118], the use of TET proteins to promote global demethylation appears to be specific to mammalian pluripotency[51,54,58] and mediated by the recently evolved *Dppa3* (Figs. 2c, 8).

In contrast to our findings in TET catalytic mutant ESCs, TET knockout ESCs do not appear to exhibit global hypermethylation[58]. This discrepancy might be explained by recent findings demonstrating that TET proteins influence global DNA methylation not only via their catalytic activity but also by their genomic binding[119,120]. Knockout of TET proteins results in a seemingly paradoxical loss of DNA methylation at repetitive elements like LINEs and LTRs due to a global redistribution of DNMT3A from heterochromatin to euchromatic sites previously occupied by TETs. In contrast to TET KOs, disruption of TET catalytic activity would not be expected to affect global TET occupancy, presumably leaving DNMT3A genomic occupancy intact. Thus, the extensive hypermethylation occurring upon TET inactivation, but not TET knockout, could be attributable to both the preservation of TET binding as well as the enhanced loading of the DNA methylation machinery on chromatin in TET CM ESCs.

To date, our understanding of DPPA3's function in the regulation of DNA methylation has been clouded by seemingly conflicting reports from different developmental stages and cell types. DPPA3's ability to modulate DNA methylation was first described in the context of zygotes[61], where it was demonstrated to specifically protect the maternal genome from TET3-dependent

demethylation[29,74,77]. In contrast, DPPA3 was later shown to prevent aberrant DNA hypermethylation during PGC specification[59], iPSC reprogramming[75], and oocyte maturation[62,121]. Whereas DPPA3 was shown to disrupt UHRF1 function by sequestering it to the cytoplasm in oocytes[62], we demonstrate that DPPA3-mediated nucleocytoplasmic translocation of UHRF1 is not only dispensable but actually attenuates DPPA3's promotion of hypomethylation in ESCs. Another example of development- and context-specific function of DPPA3 is its role in the regulation of imprinting. While DPPA3 has no impact on ICR methylation in oocytes[61,62], it is required to prevent the loss of both paternal and maternal imprints in zygotes[77]. In naïve ESCs, we found that the Dppa3 KO results in a gain of DNA methylation at ICRs. Although contrary to its zygotic role in protecting imprints from demethylation, our data is consistent with previous findings examining the effect of Dppa3 loss on iPSC generation, where imprints also became hypermethylated[75].

In light of our data from naïve ESCs, Xenopus, and medaka, DPPA3's capacity to directly bind UHRF1's PHD domain and thereby inhibit UHRF1 chromatin binding appears to be its most basal function. Considering that DPPA3 localization is highly dynamic during the different developmental time periods at which it is expressed[59,78,122], it stands to reason that its role in modulating DNA methylation might also be dynamically modulated by yet-to-be determined regulatory mechanisms. For example, immediately following fertilization, full-length DPPA3 is cleaved and its C-terminal domain is specifically degraded[78]. Interestingly, we identified this exact C-terminal stretch of DPPA3 to be necessary and sufficient for DPPA3's inhibition of maintenance DNA methylation. Thus, the precisely timed destruction of this crucial domain might offer an explanation for the differing roles of DPPA3 in regulating DNA methylation between oocytes and zygotes[62,74,77,121].

As the most basic and evolutionarily conserved function of DNA methylation is the repression of TEs[6], the post-fertilization wave of DNA demethylation found in mammals raises several fundamental questions. Considering the mutational risks associated with TE activity, why have mammals come to dispense with such a central genomic defense mechanism during early development? Whereas derepression of TEs leads to genomic instability and ultimately cell death in most cell types[9,10,13,14], TE activity is not only tolerated but increasingly appreciated to fulfill key roles in early mammalian development[123–129]. The activation of TEs, in particular endogenous retroviruses (ERVs), appears to be a conserved feature of early mammalian embryos[130], beginning after fertilization and continuing for the duration of gestation in the cells of the trophoblast and the placenta[131,132]. During mammalian evolution, the placenta emerged in the common ancestor of therian mammals, after the divergence from the egg-laying monotremes[133,134]. Accumulating evidence suggests ERVs facilitated the complex, network level changes necessary for the evolutionary emergence and diversification of placental viviparity[135–137]. By enabling embryos to directly regulate the allocation of maternal resources, placental viviparity creates unique evolutionary challenges absent in egg-laying species[138]. At the fetal–maternal interface, the interests of the mother and her offspring as well as those of the paternal and maternal genomes within the embryo are brought into conflict, unleashing a coevolutionary arms race for control of maternal resources and provisioning[139]. The existence of such an evolutionary struggle is perhaps best exemplified by the emergence of genomic imprinting, or parent-of-origin-specific gene expression, in therian mammals[140]. Transposons, particularly ERVs, have played an important role in the evolution of genomic imprinting as an adaption to

parental conflict; many of the cis-elements controlling imprinting status and, in some cases, even the imprinted genes themselves are derived from ERV insertions[141–143]. The retroviral origins of genomic imprinting are further illustrated by the use of conserved vertebrate host defense systems, namely DNA methylation and KRAB-ZFPs, to maintain imprint status[144,145]. In agreement with the parental conflict hypothesis, the evolution of more elaborated and invasive placentation has been accompanied by the expansion of genomic imprinting, with only 6 genes imprinted in marsupials compared with >100 in eutherians[146]. Indeed, the progressive cooption of retrotransposons over evolutionary time appears to have been a key driver in the transformation of a marsupial-like reproductive mode to the invasive and extended pregnancy of eutherians by facilitating the emergence of many of the unique, defining features of eutherian development such as the early allocation of the trophoblast cell lineage, invasive placentation, and suppression of the maternal immune response provoked by implantation[124,147–150]. Despite the importance of ERVs in eutherian development, the majority of ERV-derived regulatory elements, genes, and cis-elements controlling genomic imprinting are the result of evolutionarily recent and largely species-specific insertions[123,125,128,151–153].

How did eutherians come to rely on ERVs for so many aspects of their unique development? Such prolific ERV co-option among eutherians is proposed to have been a consequence of the evolution of precocious zygotic genome activation (ZGA) and an epigenetically permissive environment during early embryonic development[154,155]. It is tempting to speculate that post-fertilization demethylation was an important event in Eutherian evolution that contributed to the emergence and expansion of ERV/TE-based developmental regulation, including genomic imprints. Once ERV-derived genes and, in particular, regulatory networks acquired essential roles, mammalian preimplantation and placental development would have become "addicted" to the active transcription of ERVs[156]. Likewise, proper host development would require the establishment and maintenance of epigenetic states permissive for global ERV activity. In both mice and humans, the onset of ERV-dependent regulation coincides with a wave of genome-wide DNA demethylation, which commences upon fertilization and reaches its nadir in the ICM and trophectoderm of the blastocyst[19,40,157]. Whereas ERVs are silenced in the cells of the embryo proper by the wave of global de novo DNA methylation accompanying implantation, ERV activity and DNA hypomethylation persist in the trophoblast lineage throughout development[123,126,157–161]. Indeed, hypomethylation of the placenta relative to somatic cells appears to be conserved throughout Eutheria, despite dramatic differences in the embryonic and placental development among taxa[162].

As genome-wide DNA methylation is static throughout the lifecycle of most vertebrates, the evolution of novel mechanisms would have been required for the emergence of global DNA methylation erasure in the early embryonic development of eutherian mammals. DPPA3 may have arisen as a means to facilitate the early embryonic exposure of ERVs by neutralizing the host defense system of an ancestral eutherian mammal. In line with this notion, mouse embryos lacking Dppa3 exhibit extensive genome-wide hypermethylation and undergo developmental arrest before the blastocyst stage as a result of impaired ERV activation and ZGA failure[62,76]. As Dppa3 orthologs exhibit similar patterns of early embryonic expression in mice, humans, marmosets, cows, sheep, and pigs[163–167], it is plausible that function of DPPA3 during development is broadly conserved among mammals. However, our analysis identified Dppa3 orthologs to be present in only a single clade of placental mammals, namely Boreoeutheria (Fig. 2c).

This raises the question whether eutherian lineages that lack DPPA3 also erase their methylomes and if so how? Pre-implantation DNA demethylation has been documented in every boreoeutherian species tested to date (e.g. mice, humans, monkeys, pigs, cows, sheep, rabbits)[19,36–43], however early embryonic DNA methylation dynamics have not been investigated in Eutherian lineages other than Boreoeutheria, i.e Afrotheria and Xenarthra, not to mention the more distant marsupial and monotreme groups. Likewise, the functional importance of ERV activity in early developmental and placental gene expression programs has also only been demonstrated in boreoeutherian species. Thus, it is currently wholly unclear whether global DNA demethylation and ERV-dependent regulatory networks are even present, let alone important for early embryonic and trophoblast development outside of Boreoeutheria. Follow-up studies that investigate the origins of *Dppa3* and whether a similar ERV-based rewiring of early development may have occurred in other, not yet studied branches of vertebrates, are needed to understand how global DNA demethylation shaped the evolution of placental mammals.

## Methods

**Cell culture**. Naïve J1 mouse ESCs were cultured and differentiated into EpiLCs using an established protocol[168,169]. In brief, for both naïve ESCs and EpiLCs defined media was used, consisting of N2B27: 50% neurobasal medium (Life Technologies), 50% DMEM/F12 (Life Technologies), 2 mM L-glutamine (Life Technologies), 0.1 mM β-mercaptoethanol (Life Technologies), N2 supplement (Life Technologies), B27 serum-free supplement (Life Technologies), 100 U/mL penicillin, and 100 µg/mL streptomycin (Sigma). Naïve ESCs were maintained on flasks treated with 0.2% gelatin in defined media containing 2i (1 µM PD032591 and 3 µM CHIR99021 (Axon Medchem, Netherlands)), 1000 U/mL recombinant leukemia inhibitory factor (LIF, Millipore), and 0.3% BSA (Gibco) for at least three passages before commencing differentiation. To differentiate naïve ESCs into Epiblast-like cells (EpiLCs), flasks were first pre-treated with Geltrex (Life Technologies) diluted 1:100 in DMEM/F12 (Life Technologies) and incubated at 37 °C overnight. Naïve ESCs were plated on Geltrex-treated flasks in defined medium containing 10 ng/mL Fgf2 (R&D Systems), 20 ng/mL Activin A (R&D Systems) and 0.1× Knockout Serum Replacement (KSR) (Life Technologies). Media was changed after 24 h and EpiLCs were harvested for RRBS and RNA-seq experiments after 48 h of differentiation.

For CRISPR-assisted cell line generation, mouse ESCs were maintained on 0.2% gelatin-coated dishes in Dulbecco's modified Eagle's medium (Sigma) supplemented with 16% fetal bovine serum (FBS, Sigma), 0.1 mM ß-mercaptoethanol (Invitrogen), 2 mM L-glutamine (Sigma), 1× MEM Non-essential amino acids (Sigma), 100 U/mL penicillin, 100 µg/mL streptomycin (Sigma), homemade recombinant LIF tested for efficient self-renewal maintenance, and 2i (1 µM PD032591 and 3 µM CHIR99021 (Axon Medchem, Netherlands)).

Human ESCs (line H9) were maintained in mTeSR1 medium (05850, STEMCELL Technologies) on Matrigel-coated plates (356234, Corning) prepared by 1:100 dilution, and 5 ml coating of 10 cm plates for 1 h at 37 °C. Colonies were passaged using the gentle cell dissociation reagent (07174, StemCell Technologies).

All cell lines were regularly tested for Mycoplasma contamination by PCR.

**Sleeping beauty constructs**. To generate the sleeping beauty donor vector with an N-terminal 3xFLAG tag and a fluorescent readout of doxycycline induction, we first used primers with overhangs harboring SfiI sites to amplify the IRES-DsRed-Express from pIRES2-DsRed-Express (Clontech)(Supplementary Data 5). This fragment was then cloned into the NruI site in pUC57-GentR via cut-ligation to generate an intermediate cloning vector pUC57-SfiI-IRES-DsRed-Express-SfiI. A synthesized gBlock (IDT, Coralville, IA, USA) containing Kozak-BIO-3XFLAG-AsiSI-NotI-V5 was cloned into the Eco47III site of the intermediate cloning vector via cut-ligation. The luciferase insert from pSBtet-Pur[170] (Addgene plasmid #60507) was excised using SfiI. The SfiI-flanked Kozak-BIO-3XFLAG-AsiSI-NotI-V5-IRES-DsRed-Express cassette was digested with SfiI and ligated into the pSBtet-Pur vector backbone linearized by SfiI. The end result was the parental vector, pSBtet-3xFLAG-IRES-DsRed-Express-PuroR. The pSBtet-3x-FLAG-mScarlet-PuroR vector was constructed by inserting a synthesized gBlock (IDT, Coralville, IA, USA) containing the SfiI-BIO-3XFLAG-AsiSI-NotI-mScarlet sequence into the SfiI-linearized pSBtet-Pur vector backbone using Gibson assembly[171]. For *Dppa3* expression constructs, the coding sequence of wild-type and mutant forms of *Dppa3* were synthesized as gBlocks (IDT, Coralville, IA, USA) and inserted into the pSBtet-3xFLAG-IRES-DsRed-Express-PuroR vector (linearized by AsiSI and NotI) using Gibson assembly. To produce the *Dppa3*-mScarlet fusion expression constructs, wild-type and mutant forms of *Dppa3* were amplified from pSBtet-3xFLAG-Dppa3-IRES-DsRed-Express-PuroR constructs using primers with overhangs homologous to the AsiSI and NotI

restriction sites of the pSBtet-3x-FLAG-mScarlet-PuroR vector (Supplementary Data 5). Wild-type and mutant *Dppa3* amplicons were subcloned into the pSBtet-3x-FLAG-mScarlet-PuroR vector (linearized with AsiSI and NotI) using Gibson assembly.

For experiments involving the SBtet-3xFLAG-Dppa3 cassette, all inductions were performed using 1 µg/mL doxycycline (Sigma-Aldrich). The DPPA3-WT construct was able to rescue the cytoplasmic localization and chromatin association of UHRF1 indicating that C-terminally tagged DPPA3 remains functional (Fig. 5b–d).

**CRISPR/Cas9 genome engineering**. For the generation of *Tet1*, *Tet2*, and *Tet1/Tet2* catalytic mutants, specific gRNAs targeting the catalytic center of *Tet1* and *Tet2* (Supplementary Data 5) were cloned into a modified version of the SpCas9-T2A-GFP/gRNA plasmid (px458[172], Addgene plasmid #48138), where we fused a truncated form of human Geminin (hGem) to SpCas9 in order to increase homology-directed repair efficiency[173] generating SpCas9-hGem-T2A-GFP/gRNA.

To generate *Tet1* and *Tet2* catalytic mutant targeting donors, 200 bp single-stranded DNA oligonucleotides carrying the desired HxD mutations (*Tet1*: H1652Y and D1654A, *Tet2*: H1304Y and D1306A) and ~100 bp homology arms were synthesized (IDT, Coralville, IA, USA) (Supplementary Data 5). For targetings in wild-type J1 ESCs, cells were transfected with a 4:1 ratio of donor oligo and SpCas9-hGem-T2A-GFP/gRNA construct. Positively transfected cells were isolated based on GFP expression using fluorescence-activated cell sorting (FACS) and plated at clonal density in ESC media 2 days after transfection. After 5–6 days, single colonies were picked and plated on 96-well plates. These plates were then duplicated 2 days later and individual clones were screened for the desired mutation by PCR followed by restriction fragment length polymorphism (RFLP) analysis. Cell lysis in 96-well plates, PCR on lysates, and restriction digests were performed as previously described[169]. The presence of the desired *Tet1* and/or *Tet2* catalytic mutations in putative clones was confirmed by Sanger sequencing.

As C-terminally tagged GFP labeled UHRF1 transgenes were shown to be able to rescue U1KO[83], the tagging of endogenous *Uhrf1* was also performed at the C-terminus. For insertion of the HALO or eGFP coding sequence into the endogenous *Dppa3* and *Uhrf1* loci, respectively, *Dppa3* and *Uhrf1* specific gRNAs were cloned into SpCas9-hGem-T2A-Puromycin/gRNA vector, which is a modified version of SpCas9-T2A-Puromycin/gRNA vector (px459;[172], Addgene plasmid #62988) similar to that described above. To construct the homology donors plasmids, gBlocks (IDT, Coralville, IA, USA) were synthesized containing either the HALO or eGFP coding sequence flanked by homology arms with ~200-400 bp homology upstream and downstream of the gRNA target sequence at the *Dppa3* or *Uhrf1* locus, respectively, and then cloned into the NruI site of pUC57-GentR via cut-ligation. ESCs were transfected with equimolar amounts of gRNA and homology donor vectors. Two days after transfection, cells were plated at clonal density and subjected to a transient puromycin selection (1 µg/mL) for 40 h. After 5-6 days, ESCs positive for HALO or eGFP integration were isolated via fluorescence-activated cell sorting (FACS) and plated again at clonal density in ESC media. After 4–5 days, colonies were picked and plated on Optical bottom µClear 96-well plates and re-screened for the correct expression and localization of eGFP or HALO using live-cell spinning-disk confocal imaging. Clones were subsequently genotyped using the aforementioned cell lysis strategy and further validated by Sanger sequencing[169].

To generate *Dppa3* knockout cells, the targeting strategy entailed the use of two gRNAs with target sites flanking the *Dppa3* locus to excise the entire locus on both alleles. gRNA oligos were cloned into the SpCas9-T2A-PuroR/gRNA vector (px459) via cut-ligation (Supplementary Data 5). ESCs were transfected with an equimolar amount of each gRNA vector. Two days after transfection, cells were plated at clonal density and subjected to a transient puromycin selection (1 µg/mL) for 40 h. Colonies were picked 6 days after transfection. The triple PCR strategy used for screening is depicted in Supplementary Fig. 3a. Briefly, PCR primers 1F and 4R were used to identify clones in which the *Dppa3* locus had been removed, resulting in the appearance of a ~350 bp amplicon. To identify whether the *Dppa3* locus had been removed from both alleles, PCRs were performed with primers 1F and 2R or 3F and 4R (Supplementary Data 5) to amplify upstream or downstream ends of the *Dppa3* locus, which would only be left intact in the event of mono-allelic locus excision. Removal of the *Dppa3* locus was confirmed with Sanger sequencing and loss of *Dppa3* expression was assessed by qRT-PCR.

For CRISPR/Cas gene editing, all transfections were performed using Lipofectamine 3000 (Thermo Fisher Scientific) according to the manufacturer's instructions. All DNA oligos used for gene editing and screening are listed in Supplementary Data 5.

**Bxb1-mediated recombination and Sleeping Beauty transposition**. To generate stable mESC lines carrying doxycycline-inducible forms of *Dppa3* or *Dppa3-mScarlet*, mES cells were first transfected with equimolar amounts of the pSBtet-3xFLAG-Dppa3-IRES-DsRed-PuroR or pSBtet-3xFLAG-Dppa3-mScarlet-PuroR and the Sleeping Beauty transposase, pCMV(CAT)T7-SB100[174] (Addgene plasmid #34879) vector using Lipofectamine 3000 (Thermo Fisher Scientific) according to manufacturer's instructions. Two days after transfection, cells were plated at clonal density and subjected to puromycin selection (1 µg/mL) for 5–6 days. To ensure

comparable levels of *Dppa3* induction, cells were first treated for 18 h with doxycycline (1 μg/mL) and then sorted with FACS based on thresholded levels of DsRed or mScarlet expression, the fluorescent readouts of successful induction. Post sorting, cells were plated back into media without doxycycline for 7 days before commencing experiments.

To generate stable doxycycline-inducible *Dppa3* hESC lines, hES cells were first transfected with equimolar amounts of the pSBtet-3xFLAG-Dppa3-IRES-DsRed-PuroR and Sleeping Beauty transposase pCMV(CAT)T7-SB100[175] (Addgene plasmid #34879) vector using using the P3 Primary Cell 4D-NucleofectorTM Kit (V4XP-3012 Lonza) and the 4D-Nucleofector™ Platform (Lonza), program CB-156. Two days after nucleofection, cells were subjected to puromycin selection (1 μg/mL) for subsequent two days, followed by an outgrowth phase of 4 days. At this stage, cells were sorted with FACS based on thresholded levels of DsRed expression to obtain two bulk populations of positive stable hESC lines with inducible *Dppa3*.

For the generation of the *Uhrf1*<sup>GFP/GFP</sup> cell line, we used our previously described ESC line with a C-terminal MIN-tag (*Uhrf1*<sup>attP/attP</sup>; Bxb1 *attP* site) and inserted the GFP coding sequence as described previously[169]. Briefly, attB-GFP-Stop-PolyA (Addgene plasmid #65526) was inserted into the C-terminal of the endogenous *Uhrf1*<sup>attP/attP</sup> locus by transfection with equimolar amounts of Bxb1 and attB-GFP-Stop-PolyA construct, followed by collection of GFP-positive cells with FACS after 6 days.

**Cellular fractionation**. Cell fractionation was performed as described previously[175] with minor modifications. Approximately $1 \times 10^7$ ESCs were resuspended in 250 μL of buffer A (10 mM HEPES pH 7.9, 10 mM KCl, 1.5 mM MgCl₂, 0.34 M sucrose, 10% glycerol, 0.1% Triton X-100, 1 mM DTT, 1 mM phenylmethylsulfonyl fluoride (PMSF), 1x mammalian protease inhibitor cocktail (PI; Roche)) and incubated for 5 min on ice. Nuclei were collected by centrifugation (4 min, 1300 × *g*, 4 °C) and the cytoplasmic fraction (supernatant) was cleared again by centrifugation (15 min, 20,000 × *g*, 4 °C). Nuclei were washed once with buffer A, and then lysed in buffer B (3 mM EDTA, 0.2 mM EGTA, 1 mM DTT, 1 mM PMSF, 1× PI). Insoluble chromatin was collected by centrifugation (4 min, 1700 × *g*, 4 °C) and washed once with buffer B. Chromatin fraction was lysed with 1× Laemmli buffer and boiled (10 min, 95 °C).

**Western blot**. Western blots were performed as described previously[82,169]. The following antibodies were used:

Rabbit anti-UHRF1 (polyclonal; 1:250; Cell Signalling, D6G8E), mouse anti-alpha-Tubulin (monoclonal; 1:500; Sigma, T9026), rabbit anti-H3 (polyclonal; 1:1000; Abcam, ab1791), mouse anti-GFP (monoclonal; 1:1000; Roche), mouse anti-FLAG M2 (monoclonal; 1:1000; Sigma, F3165), rabbit anti-xDNMT1 (polyclonal;[82]), rabbit anti-xUHRF1 (polyclonal;[82]), rabbit anti-USP7 (polyclonal; Bethyl Lab., A300-033A), rabbit anti-H3 (polyclonal; Abcam, ab1791), rat anti-TET1 (monoclonal; 1:10;[176]), rat anti-alpha-Tubulin (monoclonal; 1:250; Abcam, ab6160). goat anti-rat HRP (polyclonal; 1:1000; Jackson ImmunoResearch), goat anti-rabbit HRP (polyclonal; 1:1000; BioRad), mouse anti-xCDC45 (monoclonal;[177]), mouse anti-xRPA2 (monoclonal;[178]), and mouse anti-PCNA (monoclonal; Santa Cruz, sc56). Uncropped and unprocessed scans of blots can be found in the Source Data file.

**Quantitative real-time PCR (qRT-PCR) analysis**. Total RNA was isolated using the NucleoSpin Triprep Kit (Macherey-Nagel) according to the manufacturer's instructions. cDNA synthesis was performed with the High-Capacity cDNA Reverse Transcription Kit (with RNase Inhibitor; Applied Biosystems) using 500 ng of total RNA as input. qRT-PCR assays with oligonucleotides listed in Supplementary Data 5 were performed in 8 μL reactions with 1.5 ng of cDNA used as input. FastStart Universal SYBR Green Master Mix (Roche) was used for SYBR green detection. The reactions were run on a LightCycler480 (Roche).

**LC-MS/MS analysis of DNA samples**. Isolation of genomic DNA was performed according to earlier published work[57]. 1.0–5 μg of genomic DNA in 35 μL H₂O were digested as follows: An aqueous solution (7.5 μL) of 480 μM ZnSO₄, containing 18.4 U nuclease S1 (Aspergillus oryzae, Sigma-Aldrich), 5 U Antarctic phosphatase (New England BioLabs) and labeled internal standards were added ([¹⁵N₂]-cadC 0.04301 pmol, [¹⁵N₂,D₂]-hmdC 7.7 pmol, [D₃]-mdC 51.0 pmol, [¹⁵N₅]-8-oxo-dG 0.109 pmol, [¹⁵N₂]-fdC 0.04557 pmol) and the mixture was incubated at 37 °C for 3 h. After addition of 7.5 μl of a 520 μM [Na]₂-EDTA solution, containing 0.2 U snake venom phosphodiesterase I (Crotalus adamanteus, USB corporation), the sample was incubated for 3 h at 37 °C and then stored at −20 °C. Prior to LC-MS/MS analysis, samples were filtered by using an AcroPrep Advance 96 filter plate 0.2 μm Supor (Pall Life Sciences).

Quantitative UHPLC-MS/MS analysis of digested DNA samples was performed using an Agilent 1290 UHPLC system equipped with a UV detector and an Agilent 6490 triple quadrupole mass spectrometer. Natural nucleosides were quantified with the stable isotope dilution technique. An improved method, based on earlier published work[57,179] was developed, which allowed the concurrent analysis of all nucleosides in one single analytical run. The source-dependent parameters were as follows: gas temperature 80 °C, gas flow 15 L/min (N₂), nebulizer 30 psi, sheath gas heater 275 °C, sheath gas flow 15 L/min (N₂), capillary voltage 2,500 V in the

positive ion mode, capillary voltage −2,250 V in the negative ion mode and nozzle voltage 500 V. The fragmentor voltage was 380 V/ 250 V. Delta EMV was set to 500 V for the positive mode. Chromatography was performed by a Poroshell 120 SB-C8 column (Agilent, 2.7 μm, 2.1 mm × 150 mm) at 35 °C using a gradient of water and MeCN, each containing 0.0085% (v/v) formic acid, at a flow rate of 0.35 mL/min: 0 → 4 min; 0 → 3.5% (v/v) MeCN; 4 → 6.9 min; 3.5 → 5% MeCN; 6.9 → 7.2 min; 5 → 80% MeCN; 7.2 → 10.5 min; 80% MeCN; 10.5 → 11.3 min; 80 → 0% MeCN; 11.3 → 14 min; 0% MeCN. The effluent up to 1.5 min and after 9 min was diverted to waste by a Valco valve. The autosampler was cooled to 4 °C. The injection volume amounted to 39 μL. Data were processed according to earlier published work[57].

**RNA-seq library preparation**. Digital gene expression libraries for RNA-seq were prepared using the single-cell RNA barcoding sequencing (SCRB-seq) method as described previously[180–182], with minor modifications to accommodate bulk cell populations. In brief, RNA was extracted and purified from ~1 × 10⁶ cells using the NucleoSpin Triprep Kit (Machery-Nagel) according to the manufacturer's instructions. In the initial cDNA synthesis step, purified, bulk RNA (70 ng) from individual samples were subjected to reverse transcription in 10 μL reactions containing 25 units of Maxima H Minus reverse transcriptase (ThemoFisher Scientific), 1× Maxima RT Buffer (ThemoFisher Scientific), 1 mM dNTPs (Thermo-Fisher Scientific), 1 μM oligo-dT primer with a sample-specific barcode (IDT), and 1 μM template-switching oligo (IDT). Reverse transcription reactions were incubated 90 min at 42 °C. Next, the barcoded cDNAs from individual samples were pooled together and then purified using the DNA Clean & Concentrator-5 Kit (Zymo Research) according to the manufacturer's instructions. Purified pooled cDNA was eluted in 18 μL DNase/RNase-Free Distilled Water (Thermo Fisher) and then, to remove residual primers, incubated with 1 μL Exonuclease I Buffer (NEB) and 1 μL Exonuclease I (NEB) (final reaction volume: 20 μL) at 37 °C for 30 min followed by heat-inactivation at 80 °C for 20 min. Full-length cDNA was then amplified via PCR using KAPA HiFi HotStart ReadyMix (KAPA Biosystems) and SINGV6 primer (IDT). The pre-amplification PCR was performed using the following conditions: 3 min at 98 °C for initial denaturation, 10 cycles of 15 s at 98 °C, 30 s at 65 °C, and 6 min at 68 °C, followed by 10 min at 72 °C for final elongation. After purification using CleanPCR SPRI beads (CleanNA), the pre-amplified cDNA pool concentration was quantified using the Quant-iT PicoGreen dsDNA Assay Kit (Thermo Fisher). A Bioanalzyer run using the High-sensitivity DNA Kit (Agilent Technologies) was then performed to confirm the concentration and assess the size distribution of the amplified cDNA pool (Agilent Technologies). Next, 0.8 ng of the pure, amplified cDNA pool was used as input for generating a Nextera XT DNA library (Illumina) following the Manufacturer's instructions with the exception that a custom P5 primer (P5NEXTPT5) (IDT) was used to pre-ferentially enrich for 3′ cDNA ends in the final Nextera XT Indexing PCR[180–182]. After an initial purification step using a 1:1 ratio of CleanPCR SPRI beads (CleanNA), the amplified Nextera XT Library the 300–800 bp range of the library was size-selected using a 2% E-Gel Agarose EX Gels (Life Technologies) and then extracted from the gel using the MinElute Gel Extraction Kit (Qiagen, Cat. No. 28606) according to manufacturer's recommendations. The final concentration, size distribution, and quality of Nextera XT library were assessed with a Bioanalyzer (Agilent Technologies) using a High-sensitivity DNA Kit (Agilent Technologies). The Nextera XT RNA-seq library was paired-end sequenced using a high output flow cell on an Illumina HiSeq 1500.

**Reduced representation bisulfite sequencing (RRBS) library preparation**. For RRBS library preparation, genomic DNA was isolated using the QIAamp DNA Mini Kit (QIAGEN), after an overnight lysis and proteinase K treatment. RRBS library preparation was performed as described previously[183], with slight modifications. In brief, once purified, genomic DNA (100 ng) from each sample was used as starting material and first digested with 60 units of MspI (New England Biolabs) in a 30 μl reaction volume at 37 °C overnight. Digested DNA ends were then repaired and A-tailed by adding a 2 μl of a mixture containing 10 mM dATP, 1 mM dCTP, 1 mM dGTP and Klenow fragment (3′→5′ exo-) (New England Biolabs) to the unpurified digestion reaction and incubated at 30 °C for 20 min followed by 37 °C for 20 min. Individual end-repaired and A-tailed DNA samples were purified using a 2:1 ratio of CleanPCR SPRI beads (CleanNA) and eluted in 20 μl elution buffer (10 mM Tris-HCl, pH 8.5). Next, barcoded adapters were ligated to the eluted DNA fragments in a 30 μl reaction containing 1× T4 Ligase Buffer (New England Biolabs), 2000 units of T4 Ligase (New England Biolabs), and 0.8 μM sample-specific TruSeq adapters (Illuminas) and incubated at 16 °C overnight. After adapter ligation, individual samples were first pooled before being purified with a 2:1 ratio of CleanPCR SPRI beads (CleanNA) and then eluted using 4 μl elution buffer times the number of samples in the pool. Pooled samples were then bisulfite converted using the EZ DNA Methylation-Gold™ Kit (Zymo Research) according to the manufacturer's instructions with the exception that libraries were eluted 2 × 20 μL M-elution buffer (Zymo Research). After bisulfite conversion, libraries were amplified in a 200 μl large-scale PCR reaction containing, 1x PfuTurbo Cx Reaction Buffer (Agilent Technologies), 10 units of PfuTurbo Cx Hotstart DNA Polymerase (Agilent Technologies), 1 mM dNTPs (New England Biolabs), 0.3 μM TruSeq Primers (Illumina), and 20 μl of pooled, bisulfite-converted DNA samples. After dividing the reaction into 4 wells of a 96-well plate

(each containing 50 μl), the PCR was performed using the following cycling conditions: 2 min at 95 °C for initial denaturation and Polymerase activation, 16 cycles of 30 s at 95 °C, 30 s at 65 °C, and 45 s at 72 °C, followed by 7 min at 72 °C for final elongation. After amplification, the samples are pooled together again, subjected to a final round of purification using a 1.2:1 ratio of CleanPCR SPRI beads (CleanNA), and eluted in 40 μl of elution buffer. For an initial assessment of quality and yield, purified RRBS libraries were first analyzed on 2% E-Gel Agarose EX Gels (Life Technologies) and the concentrations then measured using the Quant-iT™ PicoGreen™ dsDNA Assay-Kit (ThermoFisher). The final concentration, size distribution, and quality of each RRBS library was then assessed with a Bioanalyzer (Agilent Technologies) using a High-sensitivity DNA Kit (Agilent Technologies). RRBS libraries were then sequenced on an Illumina HiSeq 1500.

**Targeted bisulfite amplicon (TaBA) sequencing.** Genomic DNA was isolated from $10^6$ cells using the PureLink Genomic DNA Mini Kit (Thermo Fisher Scientific) according to the manufacturer's instructions. The EZ DNA Methylation-Gold Kit (Zymo Research) was used for bisulfite conversion according to the manufacturer's instructions but with the following alterations: 500 ng of genomic DNA was used as input and bisulfite converted DNA was eluted in $2 \times 20$ μL Elution Buffer (10 mM Tris-HCl, pH 8.5).

TaBA-seq library preparation entailed two sequential PCRs to first amplify a specific locus and then index sample-specific amplicons. For the first PCR, the locus specific primers were designed with Illumina TruSeq and Nextera compatible overhangs (Supplementary Data 5). The amplification of bisulfite converted DNA was performed in 25 μL PCR reaction volumes containing 0.4 μM each of forward and reverse primers, 2 mM Betaiinitialne (Sigma-Aldrich, B0300-1VL), 10 mM Tetramethylammonium chloride solution (Sigma-Aldrich T3411-500ML), 1x MyTaq Reaction Buffer, 0.5 units of MyTaq HS (Bioline, BIO-21112), and 1 μL of the eluted bisulfite converted DNA (~12.5 ng). The following cycling parameters were used: 5 min for 95 °C for initial denaturation and activation of the polymerase, 40 cycles (95 °C for 20 s, 58 °C for 30 s, 72 °C for 25 s) and a final elongation at 72 °C for 3 min. Agarose gel electrophoresis was used to determine the quality and yield of the PCR. For purifying amplicon DNA, PCR reactions were incubated with 1.8× volume of CleanPCR beads (CleanNA, CPCR-0005) for 10 min. Beads were immobilized on a DynaMag™-96 Side Magnet (Thermo Fisher, 12331D) for 5 min, the supernatant was removed, and the beads washed 2× with 150 μL 70% ethanol. After air drying the beads for 5 min, DNA was eluted in 15 μL of 10 mM Tris-HCl pH 8.0. Amplicon DNA concentration was determined using the Quant-iT™ PicoGreen™ dsDNA Assay Kit (Thermo Fisher, P7589) and then diluted to 0.7 ng/μL.

Thereafter, indexing PCRs were performed in 25 μL PCR reaction volumes containing 0.08 μM (1 μL of a 2 μM stock) each of i5 and i7 Indexing Primers (Supplementary Data 5), 1x MyTaq Reaction Buffer, 0.5 units of MyTaq HS (Bioline, BIO-21112), and 1 μL of the purified PCR product from the previous step. The following cycling parameters were used: 5 min for 95 °C for initial denaturation and activation of the polymerase, 40 cycles (95 °C for 10 s, 55 °C for 30 s, 72 °C for 40 s) and a final elongation at 72 °C for 5 min. Agarose gel electrophoresis was used to determine the quality and yield of the PCR. An aliquot from each indexing reaction (5 μL of each reaction) was then pooled and purified with CleanPCR magnetic beads as described above and eluted in 1 μL × Number of pooled reactions. Concentration of the final library was determined using PicoGreen and the quality and size distribution of the library was assessed with a Bioanalyzer. Dual indexed TaBA-seq libraries were sequenced on an Illumina MiSeq in 2 × 300 bp output mode.

**RNA-seq processing and analysis.** RNA-seq libraries were processed and mapped to the mouse genome (mm10) using the zUMIs pipeline[184]. UMI count tables were filtered for low counts using HTSFilter[185]. Differential expression analysis was performed in R using DESeq2[186] and genes with an adjusted $P < 0.05$ were considered to be differentially expressed. Hierarchical clustering was performed on genes differentially expressed in TET mutant ESCs respectively, using k-means clustering ($k = 4$) in combination with the ComplexHeatmap (v 1.17.1) R-package[187]. Principal component analysis was restricted to genes differentially expressed during wild-type differentiation and performed using all replicates of wild-type, TET mutant, and Dppa3KO ESCs.

**RRBS alignment and analysis.** Raw RRBS reads were first trimmed using Trim Galore (v.0.3.1) with the "-rrbs" parameter. Alignments were carried out to the mouse genome (mm10) using bsmap (v.2.90) using the parameters "-s 12 -v 10 -r 2 -I 1". Summary statistics of the RRBS results are provided in Supplementary Data 6 and sample reproducibility information is shown in Supplementary Fig. 9. CpG-methylation calls were extracted from the mapping output using bsmaps methratio.py. Analysis was restricted to CpG with a coverage >10. methylKit[188] was used to identify differentially methylated regions between the respective contrasts for the following genomic features: (1) all 1-kb tiles (containing a minimum of three CpGs) detected by RRBS; (2) Repeats (defined by Repbase); (3) gene promoters (defined as gene start sites −2 kb/+2 kb); and (4) gene bodies (defined as longest isoform per gene) and CpG islands (as defined by Ilingworth et al.[189]). Differentially methylated regions were identified as regions with $P < 0.05$ and a difference in

methylation means between two groups greater than 20%. Principal component analysis of global DNA methylation profiles was performed on single CpGs using all replicates of wild-type, T1KO and T1CM ESCs and EpiLCs.

**Chromatin immunoprecipitation (ChIP) and Hydroxymethylated-DNA immunoprecipitation (hMeDIP) alignment and analysis.** ChIP-seq reads for TET1 binding in ESCs and EpiLCs were downloaded from GSE57700[67] and PRJEB19897[66], respectively. hMeDIP reads for wild-type ESCs and T1KO ESCs were download from PRJEB13096[66]. Reads were aligned to the mouse genome (mm10) with Bowtie (v.1.2.2) with parameters "-a -m 3 -n 3 -best -strata". Subsequent ChIP-seq analysis was carried out on data of merged replicates. Peak calling and signal pileup was performed using MACS2 callpeak[190] with the parameters "-extsize 150" for ChIP, "-extsize 220" for hMeDIP, and "-nomodel -B -nolambda" for all samples. Tag densities for promoters and 1 kb Tiles were calculated using the deepTools2 computeMatrix module[191]. TET1 bound genes were defined by harboring a TET1 peak in the promoter region (defined as gene start sites −2 kb/+2 kb).

**Immunofluorescence staining.** For immunostaining, naïve ESCs were grown on coverslips coated with Geltrex (Life Technologies) diluted 1:100 in DMEM/F12 (Life Technologies), thereby allowing better visualization of the cytoplasm during microscopic analysis. All steps during immunostaining were performed at room temperature. Coverslips were rinsed two times with PBS (pH 7.4; 140 mM NaCl, 2.7 mM KCl, 6.5 mM $Na_2HPO_4$, 1.5 mM $KH_2PO_4$) prewarmed to 37 °C, cells fixed for 10 min with 4% paraformaldehyde (pH 7.0; prepared from paraformaldehyde powder (Merck) by heating in PBS up to 60 °C; store at −20 °C), washed three times for 10 min with PBST (PBS, 0.01% Tween20), permeabilized for 5 min in PBS supplemented with 0.5% Triton X-100, and washed two times for 10 min with PBS. Primary and secondary antibodies were diluted in blocking solution (PBST, 4% BSA). Coverslips were incubated with primary and secondary antibody solutions in dark humid chambers for 1 h and washed three times for 10 min with PBST after primary and secondary antibodies. For DNA counterstaining, coverslips were incubated 6 min in PBST containing a final concentration of 2 μg/mL DAPI (Sigma-Aldrich) and washed three times for 10 min with PBST. Coverslips were mounted in antifade medium (Vectashield, Vector Laboratories) and sealed with colorless nail polish.

The following antibodies were used: rabbit anti-DPPA3 (polyclonal; 1:200; Abcam, ab19878), mouse anti-UHRF1 (monoclonal; 1:250; Santa Cruz, sc373750), goat anti-mouse A488 (polyclonal; 1:500; used in IF; Invitrogen, A11029), donkey anti-rabbit Dylight594 (polyclonal; 1:500; Dianova, 711-516-152), anti-GFP-Booster ATTO488 (1:200; Chromotek), mouse anti-5mC (monoclonal; 1:200; Active Motif, 39649), donkey anti-anti-rabbit A555 (polyclonal; 1:500; Invitrogen, A31572), and donkey anti-anti-rabbit A488 (polyclonal; 1:500; Dianova, 711-547-003).

**Immunofluorescence and Live-cell imaging.** For immunofluorescence, stacks of optical sections were collected on a Nikon TiE microscope equipped with a Yokogawa CSU-W1 spinning-disk confocal unit (50 μm pinhole size), an Andor Borealis illumination unit, Andor ALC600 laser beam combiner (405 nm/488 nm/561 nm/640 nm), Andor IXON 888 Ultra EMCCD camera, and a Nikon 100×/1.45 NA oil immersion objective. The microscope was controlled by software from Nikon (NIS Elements, ver. 5.02.00). DAPI or fluorophores were excited with 405 nm, 488 nm, or 561 nm laser lines and bright-field images acquired using Nikon differential interference contrast optics. Confocal image z-stacks were recorded with a step size of 200 nm, 16-bit image depth, a frame size of 1024 × 1024 pixels, and a pixel size of 130 nm. Within each experiment, cells were imaged using the same settings on the microscope (camera exposure time, laser power, and gain) to compare signal intensities between cell lines.

For live-cell imaging, cells were plated on Geltrex-coated glass bottom 2-well imaging slides (Ibidi). Both still and timelapse images were acquired on the Nikon spinning-disk system described above equipped with an environmental chamber maintained at 37 °C with 5% $CO_2$ (Oko Labs), using a Nikon 100x/1.45 NA oil immersion objective and a Perfect Focus System (Nikon). Images were acquired with the 488, 561, and 640 nm laser lines, full-frame (1024 × 1024) with 1 × 1 binning, and a pixel size of 130 nm. Transfection of a RFP-PCNA vector[192] was used to identify cells in S-phase. For DNA staining in live cells, cells were exposed to media containing 200 nM SiR-DNA (Spirochrome) for at least 1 h before imaging. For imaging endogenous DPPA3-HALO in live cells, cells were treated with media containing 50 nM HaloTag-TMR fluorescent ligand (Promega) for 1 h. After incubation, cells were washed 3× with PBS before adding back normal media. Nuclear export inhibition was carried out using media containing 20 nM leptomycin-B (Sigma-Aldrich). Live-cell imaging data was acquired with NIS Elements ver. 4.5 (Nikon). NIS Elements ver. 5.02.00 (Nikon) and Volocity (PerkinElmer) were used for acquiring FRAP data. RICS measurements were acquired using FABSurf (v 1.0).

**Image analysis.** For immunofluorescence images, Fiji software (ImageJ 1.51j)[193],[194] was used to analyze images and create RGB stacks. For analysis of live-cell imaging data, CellProfiler Software (version 3.0)[195] was used to quantify fluorescence intensity

in cells stained with SiR-DNA. CellProfiler pipelines used in this study are available upon request. In brief, the SiR-DNA signal was used to segment ESC nuclei. Mean fluorescence intensity of GFP was measured both inside the segmented area (nucleus) and in the area extending 4–5 pixels beyond the segmented nucleus (cytoplasm). GFP fluorescence intensity was normalized by subtracting the experimentally-determined mean background intensity and background-subtracted GFP intensities were then used for all subsequent quantifications shown in Fig. 4 and Supplementary Figs. 4h, 5h, and 6b, c.

**Fluorescence recovery after photobleaching (FRAP).** For FRAP assays, cells cultivated on Geltrex-coated glass bottom 2-well imaging slides (Ibidi) were imaged in an environmental chamber maintained at 37 °C with 5% CO$_2$ either using the Nikon system mentioned above equipped with a FRAPPA photobleaching module (Andor) or on an Ultraview-Vox spinning-disk system (Perkin-Elmer) including a FRAP Photokinesis device mounted to an inverted Axio Observer D1 microscope (Zeiss) equipped with an EMCCD camera (Hamamatsu) and a 63x/1.4 NA oil immersion objective, as well as 405, 488 and 561 nm laser lines.

For endogenous UHRF1-GFP FRAP, eight pre-bleach images were acquired with the 488 nm laser, after which an area of 4 × 4 pixels was irradiated for a total of 16 ms with a focused 488 nm laser (leading to a bleached spot of ~1 μm) to bleach a fraction of GFP-tagged molecules within cells, and then recovery images were acquired every 250 ms for 1-2 min. Recovery analysis was performed in Fiji. Briefly, fluorescence intensity at the bleached spot was measured in background-subtracted images, then normalized to pre-bleach intensity of the bleached spot, and normalized again to the total nuclear intensity in order to account for acquisition photobleaching. Images of cells with visible drift were discarded.

***Xenopus* egg extracts**. The interphase extracts (low-speed supernatants (LSS)) were prepared as described previously[82]. After thawing, LSS were supplemented with an energy regeneration system (5 μg/ml creatine kinase, 20 mM creatine phosphate, 2 mM ATP) and incubated with sperm nuclei at 3000–4000 nuclei per μl. Extracts were diluted 5-fold with ice-cold CPB (50 mM KCl, 2.5 mM MgCl2, 20 mM HEPES-KOH, pH 7.7) containing 2% sucrose, 0.1% NP-40 and 2 mM NEM, overlaid onto a 30% sucrose/CPB cushion, and centrifuged at 15,000 g for 10 min. The chromatin pellet was resuspended in SDS sample buffer and analyzed by SDS-PAGE. GST-mDPPA3 was added to egg extracts at 50 ng/μl at final concentration.

**Monitoring DNA methylation in *Xenopus* egg extracts**. DNA methylation was monitored by the incorporation of S-[methyl-$^3$H]-adenosyl-L-methionine, incubated at room temperature, and the reaction was stopped by the addition of CPB containing 2% sucrose up to 300 μl. Genomic DNA was purified using a Wizard Genomic DNA purification kit (Promega) according to the manufacturer's instructions. Incorporation of radioactivity was quantified by liquid synchillation counter.

**Plasmid construction for recombinant mDPPA3**. To generate GST-tagged mDPPA3 expression plasmids, mDPPA3 fragment corresponding to full-length protein was amplified by PCR using mouse DPPA3 cDNA and specific primers (Supplementary Data 5). The resulting DNA fragment was cloned into pGEX4T-3 vector digested with EcoRI and SalI using an In-Fusion HD Cloning Kit.

**Protein expression and purification**. For protein expression in *Escherichia coli* (BL21-CodonPlus), the mDPPA3 genes were transferred to pGEX4T-3 vector as described above. Protein expression was induced by the addition of 0.1 mM Isopropyl β–D-1-thiogalactopyranoside (IPTG) to media followed by incubation for 12 h at 20 ˚C. For purification of Glutathione S transferase (GST) tagged proteins, cells were collected and resuspended in Lysis buffer (20 mM HEPES-KOH (pH 7.6), 0.5 M NaCl, 0.5 mM EDTA, 10% glycerol, 1 mM DTT) supplemented with 0.5% NP40 and protease inhibitors, and were disrupted by sonication on ice. After centrifugation, the supernatant was applied to Glutathione Sepharose (GSH) beads (GE Healthcare) and rotated at 4 ˚C for 2 h. Beads were then washed three times with Wash buffer 1 (20 mM Tris-HCl (pH 8.0), 150 mM NaCl, 1% TritionX-100, 1 mM DTT) three times and with Wash buffer 2 (100 mM Tris-HCl (pH 7.5), 100 mM NaCl) once. Bound proteins were eluted in Elution buffer (100 mM Tris-HCl (pH 7.5), 100 mM NaCl, 5% glycerol, 1 mM DTT) containing 42 mM reduced Glutathione and purified protein was loaded on PD10 desalting column equilibrated with EB buffer (10 mM HEPES/KOH at pH 7.7, 100 mM KCl, 0.1 mM CaCl$_2$, 1 mM MgCl$_2$) containing 1 mM DTT, and then concentrated by Vivaspin (Millipore).

**Data collection for the presence of TET1, UHRF1, DNMT1, and DPPA3 throughout metazoa**. Reference protein sequences of TET1 (Human Q8NFU7, Mouse Q3URK3, *Naegleria gruberi* D2W6T1), DNMT1 (Rat Q9Z330, Human P26358, Mouse P13864, Chicken Q92072, Cow Q92072), UHRF1 (Mouse Q8VDF2, Rat Q7TPK1, Zebra fish E7EZF3, Human Q96T88, Cow A7E320, Xenopus laevis F6UA42) and DPPA3 (Mouse Q8QZY3, Human Q6W0C5, Cow A9Q1J7) were downloaded from the Universal Protein Resource (UniProt). Orthologous were identified with *hmmsearch* of the HMMER (http://hmmer.org/)

toolkit using default parameters. Presence of the proteins throughout metazoa was visualized using iTOL[196].

**Chromatin immunoprecipitation coupled to Mass Spectrometry and Proteomics data analysis**. For Chromatin immunoprecipitation coupled to Mass Spectrometry (ChIP-MS), whole cell lysates of the doxycycline-inducible *Dppa3*-FLAG mES cells were used by performing three separate immunoprecipitations with an anti-FLAG antibody and three samples with a control IgG. Trypsinized cells were washed twice by PBS and subsequently diluted to 15*10$^6$ cells per 10 mL PBS. Paraformaldehyde (PFA) was added to a final concentration of 1% and crosslinking was performed at room temperature on an orbital shaker for 10 min. Free PFA was quenched by 125 mM Glycine for 5 min and crosslinked cells were washed twice by ice-cold PBS before cell lysis. Proteins were digested on the beads after the pulldown and desalted subsequently on StageTips with three layers of C18[197]. Here, peptides were separated by liquid chromatography on an Easy-nLC 1200 (Thermo Fisher Scientific) on in-house packed 50 cm columns of ReproSil-Pur C18-AQ 1.9-μm resin (Dr. Maisch GmbH). Peptides were then eluted successively in an ACN gradient for 120 min at a flow rate of around 300 nL/min and injected through a nanoelectrospray source into a Q Exactive HF-X Hybrid Quadrupole-Orbitrap Mass Spectrometer (Thermo Fisher Scientific). After measuring triplicates of a certain condition, an additional washing step was scheduled. During the measurements, the column temperature was constantly kept at 60 °C while after each measurement, the column was washed with 95% buffer B and subsequently with buffer A. Real time monitoring of the operational parameters was established by SprayQc[198] software. Data acquisition was based on a top10 shotgun proteomics method and data-dependent MS/MS scans. Within a range of 400-1650 m/z and a max. injection time of 20 ms, the target value for the full scan MS spectra was $3 \times 10^6$ and the resolution at 60,000.

The raw MS data was then analyzed with the MaxQuant software package (version 1.6.0.7)[199]. The underlying FASTA files for peak list searches were derived from Uniprot (UP000000589_10090.fasta and UP000000589_10090 additional. fasta, version June 2015) and an additional modified FASTA file for the FLAG-tagged *Dppa3* in combination with a contaminants database provided by the Andromeda search engine[200] with 245 entries. During the MaxQuant-based analysis the "Match between runs" option was enabled and the false discovery rate was set to 1% for both peptides (minimum length of 7 amino acids) and proteins. Relative protein amounts were determined by the MaxLFQ algorithm[201], with a minimum ratio count of two peptides.

For the downstream analysis of the MaxQuant output, the software Perseus[202] (version 1.6.0.9) was used to perform two-sided Student's *t*-test with a permutation-based FDR of 0.05 and an additional constant S0 = 1 in order to calculate fold enrichments of proteins between triplicate chromatin immunoprecipitations of anti-FLAG antibody and control IgG. The result was visualized in a scatter plot. The complete catalog of proteins interacting with FLAG-DPPA3 in ESCs including statistics can be found in Supplementary Data 3.

For GO analysis of biological processes the Panther classification system was used[203]. For the analysis, 131 interactors of DPPA3 were considered after filtering the whole amount of 303 significant interactors for a *p*-value of at least 0.0015 and 3 or more identified peptides. The resulting GO groups (determined by a two-sided Fisher's exact test) were additionally filtered for a fold enrichment of observed over expected amounts of proteins of at least 4 and a *p*-value of 5.30 E−08. The result can be found in Supplementary Data 4.

***Dppa3* overexpression in medaka embryos and immunostaining**. Medaka d-rR strain was used. Medaka fish were maintained and raised according to standard protocols. Developmental stages were determined based on a previous study[204]. *Dppa3* and mutant *Dppa3* (R107E) mRNA were synthesized using HiScribe T7 ARCA mRNA kit (NEB, E2060S), and purified using RNeasy mini kit (QIAGEN, 74104). *Dppa3* or mutant *Dppa3* (R107E) mRNA was injected into the one-cell stage medaka embryos. After 7 h of incubation at 28 ˚C, the late blastula (stage 11) embryos were fixed with 4% PFA in PBS for 2 h at room temperature, and then at 4 ˚C overnight. Embryos were dechorionated, washed with PBS, and permeabilized with 0.5% Triton X-100 in PBS for 30 min at room temperature. DNA was denatured in 4 M HCl for 15 min at room temperature, followed by neutralization in 100 mM Tris-HCl (pH 8.0) for 20 min. After washing with PBS, embryos were blocked in blocking solution (2% BSA, 1%DMSO, 0.2% Triton X-100 in PBS) for 1 h at room temperature, and then incubated with 5-methylcytosine antibody (1:200; Active Motif #39649) at 4 °C overnight. The embryos were washed with PBSDT (1% DMSO, 0.1% Triton X-100 in PBS), blocked in blocking solution for 1 h at room temperature, and incubated with Alexa Fluor 555 goat anti-mouse 2nd antibody (1:500; ThermoFisher Scientific #A21422) at 4 °C overnight. After washing with PBSDT, cells were mounted on slides and examined under a fluorescence microscope.

**Fluorescence three hybrid (F3H) assay**. The F3H assay was performed as described previously[96]. In brief, BHK cells containing multiple lac operator repeats were transiently transfected with the respective GFP- and mScarlet-constructs on coverslips using PEI and fixed with 3.0% formaldehyde 24 h after transfection. For DNA counterstaining, coverslips were incubated in a solution of DAPI (200 ng/ml)

in PBS-T and mounted in Vectashield. Images were collected using a Leica TCS SP5 confocal microscope. To quantify the interactions within the lac spot, the following intensity ratio was calculated for each cell in order to account for different expression levels: mScarlet$_{spot}$ − mScarlet$_{background}$)/(GFP$_{spot}$ − GFP$_{background}$).

**Microscale thermophoresis (MST)**. For MST measurements, mUHRF1 C-terminally tagged with GFP- and 6xHis-tag was expressed in HEK 293 T cells and then purified using Qiagen Ni-NTA beads (Qiagen #30230). Recombinant mDPPA3 WT and 1-60 were purified as described above. Purified UHRF1 (200 nM) was mixed with different concentrations of purified DPPA3 (0.15 nM to 5 µM) followed by a 30 min incubation on ice. The samples were then aspirated into NT.115 Standard Treated Capillaries (NanoTemper Technologies) and placed into the Monolith NT.115 instrument (NanoTemper Technologies). Experiments were conducted with 80% LED and 80% MST power. Obtained fluorescence signals were normalized ($F_{norm}$) and the change in $F_{norm}$ was plotted as a function of the concentration of the titrated binding partner using the MO. Affinity Analysis software version 2.1 (NanoTemper Technologies). For fluorescence normalization ($F_{norm} = F_{hot}/F_{cold}$), the manual analysis mode was selected and cursors were set as follows: $F_{cold}$ = −1 to 0 s, $F_{hot}$ = 10 to 15 s. The Kd was obtained by fitting the mean $F_{norm}$ of eight data points (four independent replicates, each measured as a technical duplicate).

**RICS**. Data for Raster Image Correlation Spectroscopy (RICS) was acquired on a home-built laser scanning confocal setup equipped with a 100x NA 1.49 NA objective (Nikon) pulsed interleaved excitation (PIE) as used elsewhere[205]. Samples were excited using pulsed lasers at 470 (Picoquant) and 561 nm (Toptica Photonics), synchronized to a master clock, and then delayed ~20 ns relative to one another to achieve PIE. Laser excitation was separated from descanned fluorescence emission by a Di01-R405/488/561/635 polychroic mirror (Semrock, AHF Analysentechnik) and eGFP and mScarlet fluorescence emission was separated by a 565 DCXR dichroic mirror (AHF Analysentechnik) and collected on avalanche photodiodes, a Count Blue (Laser Components) and a SPCM-AQR-14 (Perkin-Elmer) with 520/40 and a 630/75 emission filters (Chroma, AHF Analysentechnik). Detected photons were recorded by time-correlated single-photon counting.

The alignment of the system was verified prior to each measurement session by performing FCS with PIE on a mixture of Atto-488 and Atto565 dyes excited with pulsed 470 and 561 nm lasers set to 10 µW (measured in the collimated space before entering the galvo-scanning mirror system), 1 µm above the surface of the coverslip[206]. Cells were plated on Ibidi two-well glass bottom slides, and induced with doxycycline overnight prior to measurements. Scanning was performed in cells maintained at 37 °C using a stage top incubator, with a total field-of-view of 12 µm × 12 µm, composed of 300 pixels × 300 lines (corresponding to a pixel size of 40 nm), a pixel dwell time of 11 µs, a line time of 3.33 ms, at one frame per second, for 100–200 s. Pulsed 470 and 561 nm lasers were adjusted to 4 and 5 µW, respectively.

Image analysis was done using the Pulsed Interleaved Excitation Analysis with Matlab (PAM) software[207]. Briefly, time gating of the raw photon stream was performed by selecting only photons collected on the appropriate detector after the corresponding pulsed excitation, thereby allowing cross-talk free imaging for each channel. Then, using the Microtime Image Analysis (MIA) analysis program, slow fluctuations were removed by subtracting a moving average of 3 frames and a region of interest corresponding to the nucleus was selected, excluding nucleoli and dense aggregates. The spatial autocorrelation and cross-correlation functions (SACF and SCCF) were calculated as done previously[208] using arbitrary region RICS:

$$G(\xi, \psi) = \frac{\langle I_{RICS,1}(x,y)I_{RICS,2}(x+\xi, y+\psi)\rangle_{XY}}{\langle I_{RICS,1}\rangle_{XY}\langle I_{RICS,2}\rangle_{XY}} \quad (1)$$

where ξ and ψ are the correlation lags in pixel units along the x- and y-axis scan directions. The correlation function was then fitted to a two-component model (one mobile and one immobile component) in MIAfit:

$$G_{fit}(\xi, \psi) = A_{mob} G_{fit, mob}(\xi, \psi) + A_{imm} \exp\left(-\delta r^2 \omega_{imm}^{-2}\left(\xi^2 + \psi^2\right)\right) + y_0, \quad (2)$$

where:

$$G_{fit, mob}(\xi, \psi) = \left(1 + \frac{4D(\tau_p\xi + \tau_l\psi)}{\omega_r^2}\right)^{-1}\left(1 + \frac{4D(\tau_p\xi + \tau_l\psi)}{\omega_z^2}\right)^{-1/2}$$
$$\cdot \exp\left(-\frac{\delta r^2(\xi^2 + \psi^2)}{\omega_r^2 + 4D(\tau_p\xi + \tau_l\psi)}\right) \quad (3)$$

which yields parameters such as the diffusion coefficient (D) and the amplitudes of the mobile and immobile fractions ($A_{mob}$ and $A_{imm}$). The average number of mobile molecules per excitation volume on the RICS timescale was determined by

$$N_{mob} = \left(\frac{\gamma}{A_{mob}}\right)\left(\frac{2\Delta F}{2\Delta F + 1}\right), \quad (4)$$

where γ is a factor pertaining to the 3D Gaussian shape of the PSF, and $2\Delta F/(2\Delta F + 1)$ is a correction factor when using a moving average subtraction prior to calculating the SACF. The immobilized molecules (i.e. bound fraction) is the contribution of particles that remain visible without significant motion during the acquisition of 5–10 lines of

the raster scan, corresponding to ~30 ms. The cross-correlation model was fitted to the cross-correlation function and the extent of cross-correlation was calculated from the amplitude of the mobile fraction of the cross-correlation fit divided by the amplitude of the mobile fraction of the autocorrelation fit of DPPA3-mScarlet.

**Statistics and reproducibility**. No statistical methods were used to predetermine sample size, the experiments were not randomized, and the investigators were not blinded to allocation during experiments and outcome assessment. Blinding was not implemented in this study as analysis was inherently objective in the overwhelming majority of experiments. For microscopy analysis, where possible, experimenter bias was avoided by selecting fields of view (or individual cells) for acquisition of UHRF1-GFP or DNMT1-GFP signal using the DNA stain (or another marker not being directly measured in the experiment e.g. DsRed/mScarlet as a readout of Dppa3 induction or RFP-PCNA). To further reduce bias, imaging analysis was subsequently performed indiscriminately on all acquired images using semi-automated analysis pipelines (either with CellProfiler or Fiji scripts). All the experimental findings were reliably reproduced in independent experiments as indicated in the Figure legends. In general, all micrographs from immunofluorescence and live cell imaging, immunoblots, and DNA gel images depicted in this study are representative of $n \geq 2$ independent experiments. The number of replicates used in each experiment are described in the figure legends and/or in the Methods section, as are the Statistical tests used. P values or adjusted P values are given where possible. Unless otherwise indicated, all statistical calculations were performed using R Studio 1.2.1335. Next-generation sequencing experiments include at least two independent biological replicates. RNA-seq experiments include $n = 4$ biological replicates comprised of $n = 2$ independently cultured samples from two clones (for T1CM, T2CM, T12CM ESCs and EpiLCs) or four independently cultured samples (for wild-type ESCs and EpiLCs). For RRBS experiments, data are derived from $n = 2$ biological replicates. For bisulfite sequencing of LINE-1 elements $n = 2$ biological replicates were analyzed from two independent clones for T1CM, T2CM, T12CM, and Dppa3KO ESCs or two independent cultures for wt ESCs. LC-MS/MS quantification was performed on at least four biological replicates comprising at least two independently cultured samples (usually even more) from $n = 2$ independent clones (T1CM, T2CM, T12CM, and Dppa3KO ESCs) or four independently cultured samples (wild-type ESCs and cell lines shown in Fig. 5d).

**Reporting summary**. Further information on research design is available in the Nature Research Reporting Summary linked to this article.

## Data availability
Sequencing data reported in this paper are available at ArrayExpress (EMBL-EBI) under accessions "E-MTAB-6785" (wild-type and Tet catalytic mutants RRBS), "E-MTAB-6797" (RNA-seq), "E-MTAB-6800" (Dppa3KO RRBS), "E-MTAB-9654" (TaBA-seq of Tet catalytic mutants during Dppa3 induction) and "E-MTAB-9653" (TaBA-seq of Dppa3KO cells expressing *Dppa3* mutant constructs). The raw mass spectrometry proteomics data from the FLAG-DPPA3 pulldown have been deposited at the ProteomeXchange Consortium via the PRIDE partner repository with the dataset identifier "PXD019794". Publically available data sets used in this study can be found here: "GSE77420" (RRBS of TET triple knockout ESCs), "GSE42616" (PRDM14 ChIP-seq), "GSE46111" (5caC-DIP in TDK knockout ESCs), "GSE57700" (TET1 and TET2 ChIP-seq).

Supplementary Data 1 contains the entire list of differentially methylated promoters classified as either "TET-specific", "DPPA3-specific" or "common", which are summarized in Supplementary Fig. 3i. Supplementary Data 2 contains the extended gene ontology analysis of TET-specific promoters with the five most significant terms displayed in Fig. 3e. Supplementary Data 3 contains the complete catalog of proteins interacting with FLAG-DPPA3 in ESCs, which are plotted in Fig. 4b. Supplementary Data 4 contains the full gene ontology analysis of significant DPPA3 interactors. Source data are provided with this paper.

## Code availability
The PAM and MIA software is available as source code, requiring MATLAB, or as a precompiled, standalone distribution for Windows or MacOS at http://www.cup.uni-muenchen.de/ pc/lamb/software/pam.html or hosted in Git repositories under http://www.gitlab.com/PAM-PIE/PAM and http://www.gitlab.com/PAM-PIE/PAMcompiled.

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

## Acknowledgements

The authors would like to thank Dr. H. Blum, Dr. S. Krebs (Laboratory for Functional Genome Analysis, LMU Munich), and Dr. A. Brachmann (Faculty of Biology, Department of Genetics, LMU Munich) for next-generation sequencing services. We further thank Dr. Ralf Heermann (Bioanalytics core facility, LMU Munich) for help with the MST measurements. We thank V. Laban, C. Kirschner, T.J. Fischer, D. Nixdorf, and V.B. Ozan for help with experiments, J. Koch for technical assistance, Dr. M. Mann and Dr. I. Paron for mass spectrometry, and Dr. I. Hellmann for advice on data analysis and providing computational infrastructure. We would like to thank Dr. S. Stricker and Dr. I. Solovei for helpful discussions and constructive criticism on the manuscript, E. Ntouliou for critical reading of the manuscript, and G. Mahler for inspiration. We thank Dr. Feng Zhang for providing the pSpCas9(BB)-2A-Puro (PX459) and SpCas9-T2A-GFP (PX458) (Addgene plasmids #62988 and #48138) plasmids, Dr. Eric Kowarz for the gift of the pSB-tetPur (Addgene plasmid #60507) construct, and Dr. Zsuzsanna Izsvak for the gift of the pCMV(CAT)T7-SB100 (Addgene plasmid #34879) construct. J.R., C.T., E.U., M.M., and M.D.B. are fellows of the International Max Planck Research School for Molecular Life Sciences (IMPRS-LS). C.B.M. gratefully acknowledges the support of the Fulbright Commission and the late Dr. Glenn Cuomo. F.R.T. thanks the Boehringer Ingelheim Fonds and J.R. the Fonds de Recherche du Québec en Santé for a PhD fellowship. This work was funded by the Deutsche Forschungsgemeinschaft (DFG, German Research Foundation) – Project-ID 213249687 – SFB 1064 to S.B. (A22) and H.L. (A17), by SFB1243/A01 to H.L., and SFB1243/A14 to W.E.

## Author contributions

C.B.M. and S.B. designed and conceived the study. S.B. and H.L. supervised the study. C.B.M., S.B., and H.L. prepared the manuscript with the help of M.D.B. C.B.M. performed cellular and molecular experiments. C.B.M. generated cell lines with help from M.Y., C.B.M. performed RRBS and RNA-Seq with help and supervision from C.Z., S.B., and W.E., J.R. and C.B.M. performed live-cell microscopy and photobleaching analyses. I.G., J.R., and C.B.M. performed RICS experiments under the supervision of D.C.L., C.T. performed MST and F3H assays. A.N. performed Xenopus experiments under the supervision of M.N., R.N. performed the experiments in medaka embryos under the supervision of H.T. M.D.B. and P.S. helped with cell line validation and performed fluorescence microscopy analysis. W.Q. performed the biochemical analyses with assistance from A.A., M.M. performed hESC experiments. E.U. conducted proteomics experiments and analyses under the guidance of M.W. F.R.T. and E.P. quantified modified cytosines by LC-MS/MS with the supervision by T.C. S.B. performed data analysis. All authors read, discussed, and approved the manuscript.

## Funding

## Competing interests

The authors declare no competing interests.
