## [Peer Review File · Nature Communications]

REVIEWER COMMENTS

Reviewer #1 (Remarks to the Author):

In the manuscript titled "Recent evolution of a TET-controlled and DPPA3/STELLA-driven pathway of passive demethylation in mammals", Mulholland et al. show that TET1 and 2 facilitate demethylation of the Dppa3 promoter in naive ES cells and thereby mediate upregulation of Dppa3 expression. They also find that ablation of Dppa3 causes widespread gain of DNA methylation, demonstrating a role of DPPA3 for protecting the genome from aberrant DNA methylation. Consistent with this observation, ectopic expression of Dppa3 induces global hypomethylation, even in the absence of Tet1 and 2 (Tet1,2-dKO). Mulholland et al. further reveal that DPPA3 associates with UHRF1, and inhibits UHRF1-mediated maintenance methylation. As DPPA3 protein is found in mammals, but not in amphibians or fish, the authors propose a model by which the evolutionary gain of DPPA3 could facilitate widespread demethylation observed in post-fertilization embryos, and in primordial germ cells, in mammals.

I have the following comments to the authors.

Major comments:

1) In the past few years, mechanistic dissection of the role of DPPA3/STELLA in mammalian cells has been done intensively. Indeed, it appears that several key findings of this paper have already been reported in previously. For example, hypomethylation of the Dppa3 promoter in naive ESCs, and hypermethylation in primed ESCs is now well-known (PMCID: PMC3863718; PMCID: PMC3847852; and others). DNA methylation dependent repression of the Dppa3 promoter, by DNMT1 (PMCID: PMC3461056), or DNMT3A/3B (PMCID: PMC4414868; PMCID: PMC6734493), has also been shown. Importantly, a role for DPPA3 for suppression of DNA methylation via interactions with UHRF1 (PMID: 25280994), or DNMT1 (PMID: 30487604) has been reported. Consistent with this notion, a physiological role of DPPA3 to protect the genome from DNA methylation has been demonstrated in post-fertilization embryos (PMID: 30487604), and PGCs (PMID: 23595900). Given these previous observations on DPPA3, the authors should clearly show what are the novel findings revealed by their study, and how their findings extend previous knowledge linking DPPA3 and DNA demethylation.

2) By ectopically expressing DPPA3, the authors show that even in the absence of TET1 and 2 (Tet1,2-dKO), DPPA3 alone is sufficient to drive demethylation at the LINE repeats (Fig 3g). This is an interesting observation, and the authors may like to expand on this finding, by discussing more how DPPA3 overexpression in Tet1,2-dKO cells promotes demethylation is specific genomic compartments (or repeat elements), and how this phenomenon is mechanistically regulated.

3) The interaction between DPPA3 and UHRF1 in the protein level that contributes to passive demethylation via suppression of UHRF1-mediated DNA methylation, is a key finding of this paper. However, it should be noted that molecular interactions between these two proteins have been shown by other groups too, such as by Du et al. (PMCID: PMC6552428). Furthermore, previous papers have already shown that UHRF1 binds DPPA3, by mass-spec analysis (PMCID: PMC5696369).

4) The proposed model linking DPPA3 and mammalian evolution is certainly interesting. Indeed, depletion of xUHRF1 and xDNMT1 from the chromatin, under the presence of mDDPA3, are in line with the authors' findings in mouse cells that DPPA3 prevents chromatin loading of UHRF1. In the same way, downregulation of 5mC levels in mid-gastrula stage fish embryos upon injection of mDppa3 reconfirms the role of DPPA3 to promote demethylation. However, the experiments with Xenopus egg extracts and medaka embryos, do seem a bit out of context, when compared to the rest of the paper. Also, the proposed model that evolutionary gain of DPPA3 drive demethylation in the early mammalian embryo appears to be a bit of stretch, given the limited scope of the amphibian egg extract and fish embryo experiments performed in this study.

Minor comments:

5) I agree that the Tet1,2-dependent mechanism is a key mediator for Dppa3 expression. However, it is worth noting that Dppa3 is regulated by a number of other pathways, such as inhibition of DNMT3A/3B-dependent de novo DNA methylation by LIN28 (PMCID: PMC6734493), suppression of H3K9me2 by MAD2L2 (PMCID: PMC5390107), and regulation by enhancer RNA (eRNA) (PMCID: PMC5111363). Consistent with this notion, Dppa3 is upregulated by approximately 150-fold in PGCs, compared to epiblasts (PMCID: PMC3461056). Knocking down Dnmt1 in ES cells leads to only 3-fold upregulation of Dppa3 (PMCID: PMC3461056), indicating that the maintenance methylation pathway is one of the mechanisms to silence Dppa3 expression. Indeed, in Fig 2d, the authors show that knockout of Tet1,2 leads to about 4-fold (log2, 2-fold) suppression of Dppa3; which is reminiscent of the 3-fold upregulation of Dppa3 observed in Dnmt1-KD ESCs (PMCID: PMC3461056).

6. The authors show severe hypermethylation of the genome in Tet1,2-dKO cells (Fig 1a, 1b). In these experiments, the authors may include previously published Tet-KO DNA methylation results (such as in PMCID: PMC4035811), as controls.

7. In page 5, line 115, the authors state that the Tet1,2-cKO cells gain a “more differentiated DNA methylation signature”, which is a bit confusing. Simply saying that these cells show “a methylation signature that is closer to differentiated cells”, might suffice.

8. Absence of clear TET-enrichment peaks in the regions that gain DNA methylation in the Tet1,2-dKO, forms the basis of the “indirect DNA demethylation model by TET proteins”, proposed by the authors in this manuscript. However, this interpretation could be contested. Indeed, it is plausible that TET proteins bind the above-mentioned regions only transiently, and are then evicted by binding of downstream factors such as TDG. In such cases, TET peaks will not be detected by ChIP-seq. However, methods suitable for detection of transient protein-DNA interactions (such as DamID) may show association of TET proteins to these loci.

9. In Fig 4a, the authors report that DPPA3 is localized in the cytoplasm. This is hardly surprising, as several other studies have already reported that TET protein localization in the cytoplasm (for example, see PMCID: PMC5510028).

Taken together, I would suggest that the authors focus on what is novel in their study, and streamline the manuscript in a manner that strengthen these new findings on DPPA3/UHRF1 interactions.

Reviewer #2 (Remarks to the Author):

Genome-wide methylation reprogramming is a remarkable feature of mammalian development, however the mechanisms driving global hypomethylation in naïve embryonic cells are not elucidated.

Here the authors provide evidence that TET enzymes indirectly drive hypomethylation by activating the expression of DPPA3, and that DPPA3 inhibits global methylation by displacing UHRF1 from chromatin. They propose the very interesting model that Dppa3 has specifically evolved to drive global methylation reprogramming in mammals.

It was previously suggested that DPPA3 impairs global DNA methylation by inhibiting UHRF1 function in somatic cell lines and some of the results on the UHRF1-DPPA3 interaction confirm a 2019 publication, thus some aspects of the manuscript are not entirely novel. Nevertheless, this does not alter the general quality of the work and I am very enthusiastic about this manuscript, which provides important novel mechanistic insights into genome DNA hypomethylation associated

with naïve pluripotency in mammalian embryonic cells. The manuscript contains an impressive amount of genomic, biochemical and imaging experiments, which were carefully conducted with many appropriate controls. The manuscript is well written and the results are presented in a clear way.

I have specific concerns that need to be addressed in a revised version:

-The finding of a role of TET1/2 in hypomethylation of naïve cells is in contradiction with a previous article by Von Meyenn et al. *Mol Cell* (2016), which reported no global hypermethylation in TET1/2 DKO naïve ES cells. How can the findings presented here be reconciled with these previous results? This should be discussed.

-I would like to see more quality validation of the RRBS data: % bisulfite conversion, correlation between replicates etc... Furthermore in Figure 1a, do both replicates of TET mutants show similar hypermethylation?

-Figure 2a: it should also be verified by western blot if there are no changes in protein abundance for DNMTs or UHRF1.

-Page 9 line 174 on the comparison between TET KO and DPPA3 KO ESCs: the statement that "the majority (~90%) of common targets reside within repetitive elements (Fig. 3d, Supplementary Fig. 3e,f)" deserves clarification. If I am not mistaken, Fig. 3d shows that 90% of hypermethylation events in repeats are common, which is very different than saying that 90% of common hypermethylation events are in repeats. In addition I do not understand what is represented in Supplementary Fig. 3e. Can this point be clarified?

-To demonstrate causality between TET mediated demethylation and Dppa3 expression, can the authors show that Dppa3 is transcriptionally upregulated upon DNA methylation inhibition in mouse ESCs?

-It is surprising that DPPA3 localization is entirely cytoplasmic in naïve mouse ESCs given that it is supposed to shuttle to the nucleus to impair UHRF1 binding to chromatin. If the authors have an antibody, could they probe DPPA3 by western blot in their nuclear/cytoplasmic fractionation assay?

-The findings reported here are in apparent contradiction with the documented role of DPPA3 in the maintenance of DNA methylation at imprinted genes in early embryos (Nakamura et al., *Nature Cell Biol* 2007). It would be interesting to look at methylation of imprinting control regions in the mutant ESCs.

-The experiment introducing DPPA3 in fish embryos is very interesting. Nevertheless, one should be cautious when measuring global DNA methylation levels by 5mC immunofluorescence. Would it be possible to perform RRBS on pooled embryos?

Minor points:

-The manuscript's initial postulate is that global demethylation in early embryogenesis is mammal-specific whereas TET protein are conserved among vertebrates. Can the authors include specific references in the introduction (line 82) to support their claim that global methylation reprogramming is mammal specific?

-It should be made explicit in the main text how many independent Crispr-Cas9 clones were generated and studied for each condition.

-The panels of Figure 2 are not presented in order in the text.

Reviewer #3 (Remarks to the Author):

In early mammalian development, the genome undergoes a dramatic wave of global DNA methylation erasure followed by reestablishment. The mechanisms at play in this process, as well as the evolutionary underpinnings for its existence, are active areas of research in the field of epigenetics. The emergence of the TET family of enzymes, which can drive DNA demethylation via iterative oxidation of methyl-cytosines, led to a flood of research papers trying to determine how these proteins could contribute to embryonic DNA methylation reprogramming. Here, Mulholland et al use a mouse embryonic stem cell (ESC) model to assess the role of TET enzymes in global DNA methylation. They made the remarkable observation that TET-dependent demethylation and activation of a single gene—*Stella/Dppa3*—is required to maintain low DNA methylation in serum-free culture conditions. In this sense, TET enzymes are required for global demethylation, just in a counterintuitive manner. I believe this curiosity per se will be of wide interest to the field of mammalian epigenetics. The authors went on to show that DPPA3 inhibits the DNA methylation maintenance factor UHRF1 via two mechanisms: sequestering to the cytoplasm and, more importantly, preventing UHRF1 from accessing chromatin. These collective findings are reminiscent of two recent studies: Li et al 2018 (PMID: 30487604) showed that DPPA3 maintains hypomethylation oocytes by UHRF1 cytoplasmic sequestration, and von Meyenn et al 2016 (PMID: 27315559) showed that ESCs cultured without serum exhibit impaired UHRF1. What really separates this study from the others is the rigorous dissection of the mechanism of action. I am not surprised, as the labs that contributed to this study have been leaders in our understanding DNA methylation mechanism, and this particular paper will be an important addition their respective oeuvres. I was particularly impressed by their sophisticated use of microscopy in order to assay live chromatin dynamics. The question becomes, is their study consistent with the previous publications? I think the case can be made that it is: the authors show that UHRF1 is both sequestered (as shown in Li et al), and reduced (as shown in von Meyenn et al) because of DPPA3 action. Also, it should be noted that in the Li et al paper, the cell type being assessed was oocytes, which may be subject to different control of DNA methylation pathways. Finally, the authors pushed beyond the other studies by trying to understand DPPA3 action through an evolutionary lens. DPPA3 is not even present in certain mammalian lineages, such as monotremes, which should be noted, are egg-laying. This implies that DNA methylation reprogramming may be restricted to placental mammals (it has already been shown not to occur in other vertebrates, such as amphibians and fish). As such, DPPA3 may have evolved as the key factor to allow the dramatic reshaping of the epigenome during development. Why this is the case is an interesting matter of debate, that the authors touch on in the discussion. Finally, and strikingly, DPPA3 can induce demethylation when introduced into xenopus (amphibians) or medaka (fish).

All told, I strongly recommend this paper for publication. It is an important paper for the DNA methylation field both in terms of mechanism and evolution. It will be exciting to eventually look in the pre-implantation embryo in order to confirm this mechanism in vivo. I have some improvements that I outline below. We all know these are particular times for reviewing papers, as performing experiments will be difficult. Therefore, I recommend only two small experiments that I would expect in non-lockdown conditions, but would be perfectly willing to overlook given the circumstances:

“Normal times” recommendations:

- Great dissection of mechanism in figure 5, but why not use global quantification of DNA methylation from 0-100%? A LUMA is a quick way to get this result.

- What happens if you express DPPA3 in serum-grown ESCs (which are normally highly methylated). Is this enough to get methylation close to 2i (ie, serum-free) levels?

"Covid-19 times" recommendations

- Figure 2 panels are not described in proper order in text
- Supp figure 2: maybe make the color legend squares, bc it's confusing as ESCs are also circles
- Personal taste, but I prefer UpSet plots to the complicated Venn diagrams like in figure 3c
- In the discussion, there is no mention of imprinting, which is also a phenomenon restricted to eutherian mammals. Is there a link between DPPA3 and imprinting/genetic conflict? For example, the gametes arrive asymmetrically methylated, which may confer parent specific control of in utero growth. DPPA3 evolved to "level" the differences, but imprints can bypass this control. This is a worthwhile matter of discussion.

POINT BY POINT ANSWER

Reviewer #1 (Remarks to the Author):

In the manuscript titled "Recent evolution of a TET-controlled and DPPA3/STELLA-driven pathway of passive demethylation in mammals", Mulholland et al. show that TET1 and 2 facilitate demethylation of the Dppa3 promoter in naïve ES cells and thereby mediate upregulation of Dppa3 expression. They also find that ablation of Dppa3 causes widespread gain of DNA methylation, demonstrating a role of DPPA3 for protecting the genome from aberrant DNA methylation. Consistent with this observation, ectopic expression of Dppa3 induces global hypomethylation, even in the absence of Tet1 and 2 (Tet1,2-dKO). Mulholland et al. further reveal that DPPA3 associates with UHRF1, and inhibits UHRF1-mediated maintenance methylation. As DPPA3 protein is found in mammals, but not in amphibians or fish, the authors propose a model by which the evolutionary gain of DPPA3 could facilitate widespread demethylation observed in post-fertilization embryos, and in primordial germ cells, in mammals.

I have the following comments to the authors.

Major comments:

1) In the past few years, mechanistic dissection of the role of DPPA3/STELLA in mammalian cells has been done intensively. Indeed, it appears that several key findings of this paper have already been reported in previously. For example, hypomethylation of the Dppa3 promoter in naïve ESCs, and hypermethylation in primed ESCs is now well-known (PMCID: PMC3863718; PMCID: PMC3847852; and others). DNA methylation dependent repression of the Dppa3 promoter, by DNMT1 (PMCID: PMC3461056), or DNMT3A/3B (PMCID: PMC4414868; PMCID: PMC6734493), has also been shown. Importantly, a role for DPPA3 for suppression of DNA methylation via interactions with UHRF1 (PMID: 25280994), or DNMT1 (PMID: 30487604) has been reported. Consistent with this notion, a physiological role of DPPA3 to protect the genome from DNA methylation has been demonstrated in post-fertilization embryos (PMID: 30487604), and PGCs (PMID: 23595900). Given these previous observations on DPPA3, the authors should clearly show what are the novel findings revealed by their study, and how their findings extend previous knowledge linking DPPA3 and DNA demethylation.

We thank the reviewer for the comment. We provide a more in-depth discussion of previous literature in the revised manuscript.

2) By ectopically expressing DPPA3, the authors show that even in the absence of TET1 and 2 (Tet1,2-dKO), DPPA3 alone is sufficient to drive demethylation at the LINE repeats (Fig 3g). This is an interesting observation, and the authors may like to expand on this finding, by discussing more how DPPA3 overexpression in Tet1,2-dKO cells promotes demethylation is specific genomic compartments (or repeat elements), and how this phenomenon is mechanistically regulated.

Loss of DNA methylation maintenance has been documented previously to result in drastic loss of DNA methylation at repetitive sequences including LINE-1 elements in mouse ESC (Li et al. 2018). This is in part due to the high abundance of these elements in the genome harbouring a significant portion of the global methylated CpG pool. We agree that it will be revealing to compare the demethylation at different sequences and in different compartments but that would require a clean dissection of the contributions of de novo and maintenance DNA methylation as well as active and passive demethylation mechanisms, which we have planned but will go well beyond the scope of this manuscript.

3) The interaction between DPPA3 and UHRF1 in the protein level that contributes to passive demethylation via suppression of UHRF1-mediated DNA methylation, is a key finding of this paper.

However, it should be noted that molecular interactions between these two proteins have been shown by other groups too, such as by Du et al. (PMCID: PMC6552428). Furthermore, previous papers have already shown that UHRF1 binds DPPA3, by mass-spec analysis (PMCID: PMC5696369).

We were also aware that the UHRF1-DPPA3 interaction has been reported by other groups previously, and referred to these works in our original manuscript. Given the substantial contradictions in the literature with regards to DPPA3's role in regulating DNA methylation in oocytes and zygotes (see Discussion) and the lack of studies exploring DPPA3 function in pluripotent cell types, we felt that drawing any conclusion about DPPA3's ability to modulate DNA methylation in ESCs would be dubious in the absence of a thorough elucidation of the operative mechanism. As such, we present the first unbiased, deep, and systematic interactome of DPPA3 in mESCs. This comprehensive data set fits and complements our other studies in this experimental system. This allows us and the reader also to view the interaction with UHRF1 in context of the other interactions, like e.g. with elements of import and export machinery. Perhaps most importantly, our use of RICS on live endogenously-tagged UHRF1-GFP ESCs allowed us to definitively demonstrate the existence of a direct UHRF1-DPPA3 interaction in living cells for the first time. We also provide confirmatory biochemical evidence for the interaction of UHRF1-DPPA3 in ESCs, which confirms previous *in vitro* work (e.g. Du et al. 2019). Rather than diminish our findings, we believe that it is of high value for the scientific discourse to obtain independent confirmation of a given interaction in other experimental systems. In any case, we have made sure to give due credit to these important previous studies.

4) The proposed model linking DPPA3 and mammalian evolution is certainly interesting. Indeed, depletion of xUHRF1 and xDNMT1 from the chromatin, under the presence of mDPPA3, are in line with the authors' findings in mouse cells that DPPA3 prevents chromatin loading of UHRF1. In the same way, downregulation of 5mC levels in mid-gastrula stage fish embryos upon injection of mDppa3 reconfirms the role of DPPA3 to promote demethylation.

However, the experiments with *Xenopus* egg extracts and medaka embryos, do seem a bit out of context, when compared to the rest of the paper.

Also, the proposed model that evolutionary gain of DPPA3 drive demethylation in the early mammalian embryo appears to be a bit of stretch, given the limited scope of the amphibian egg extract and fish embryo experiments performed in this study.

We believe that the *Xenopus* and Medaka experiments offer important insights for understanding how the regulation of vertebrate DNA methylation can vary so drastically among lineages despite the deep conservation of the DNA (de)methylation machinery. We demonstrate that DPPA3 is capable of inhibiting the maintenance of DNA methylation in both mammals (mouse and human) as well as in distantly related non-mammals that naturally lack both DPPA3 and global DNA methylation reprogramming.

While we agree with the reviewer that the data gleaned from our Medaka and *Xenopus* experiments do not allow us to conclude that the evolutionary gain of DPPA3 was the sole impetus in the emergence of embryonic demethylation in mammals, our data strongly argue that DPPA3 likely contributed to the development of this uniquely mammalian phenomenon. With this important distinction in mind, we have modified the text where appropriate to remove any instances where we may have overstated this point.

Minor comments:

5) I agree that the Tet1,2-dependent mechanism is a key mediator for Dppa3 expression. However, it is worth noting that Dppa3 is regulated by a number of other pathways, such as inhibition of DNMT3A/3B-dependent *de novo* DNA methylation by LIN28 (PMCID: PMC6734493), suppression of H3K9me2 by MAD2L2 (PMCID: PMC5390107), and regulation by enhancer RNA (eRNA) (PMCID: PMC5111363). Consistent with this notion, Dppa3 is upregulated by approximately 150-fold in PGCs, compared to epiblasts (PMCID: PMC3461056). Knocking down Dnmt1 in ES cells leads to only 3-fold upregulation of Dppa3 (PMCID: PMC3461056), indicating

that the maintenance methylation pathway is one of the mechanisms to silence *Dppa3* expression. Indeed, in Fig 2d, the authors show that knockout of Tet1,2 leads to about 4-fold (log2, 2-fold) suppression of *Dppa3*; which is reminiscent of the 3-fold upregulation of *Dppa3* observed in *Dnmt1*-KD ESCs (PMCID: PMC3461056).

We thank the reviewer for drawing our attention to several studies which we had not previously referenced. We appreciate the reviewer's astute observation that DNMT1 depletion and TET inactivation have an almost reciprocal effect on *Dppa3* expression in ESCs. As this previous work provides support for both the importance and extent of DNA methylation in the regulating *Dppa3* expression, we have now included this information in the revised manuscript. We completely agree that *Dppa3* expression is regulated by a complex network of pathways which includes but is not limited to the TET-dependent demethylation of the *Dppa3* promoter. In addition to describing the role of TET and PRDM14 in activating *Dppa3*, we expound on additional regulatory pathways (e.g. LIN28, MAD2L2, and components of the pluripotency factor network) involved in promoting *Dppa3* transcription in the revised manuscript.

6. The authors show severe hypermethylation of the genome in Tet1,2-dKO cells (Fig 1a, 1b). In these experiments, the authors may include previously published Tet-KO DNA methylation results (such as in PMCID: PMC4035811), as controls.

We agree with the reviewer that these seemingly contradicting results require further discussion. We now included a comparison of our data to von Meyenn et al. 2016 in Suppl. Fig. 2f and have added a detailed discussion of this topic to the revised manuscript.

7. In page 5, line 115, the authors state that the Tet1,2-cKO cells gain a "more differentiated DNA methylation signature", which is a bit confusing. Simply saying that these cells show "a methylation signature that is closer to differentiated cells", might suffice.

We apologize for the confusion and have changed the text as suggested by the reviewer.

8. Absence of clear TET-enrichment peaks in the regions that gain DNA methylation in the Tet1,2-dKO, forms the basis of the "indirect DNA demethylation model by TET proteins", proposed by the authors in this manuscript. However, this interpretation could be contested. Indeed, it is plausible that TET proteins bind the above-mentioned regions only transiently, and are then evicted by binding of downstream factors such as TDG. In such cases, TET peaks will not be detected by ChIP-seq. However, methods suitable for detection of transient protein-DNA interactions (such as DamID) may show association of TET proteins to these loci.

We agree with the reviewer that lack of binding does not necessarily exclude catalytic activity at a given site. To investigate this, we analysed TET1/2 occupancy at regions enriched for DNA demethylation intermediates (5caC) and can show that these sites are enriched for TET1 and TET2 (Suppl Figure 2g). Hypermethylated sites in Tet CM ESCs are enriched for neither active demethylation intermediates nor TET protein, providing an additional indication that global hypermethylation is an indirect effect of the loss of catalytic activity. This is strongly supported by our data from *Dppa3*KO ESCs, which exhibit a nearly identical pattern of hypermethylation as Tet CM ESCs despite TET proteins being active.

9. In Fig 4a, the authors report that DPPA3 is localized in the cytoplasm. This is hardly surprising, as several other studies have already reported that TET protein localization in the cytoplasm (for example, see PMCID: PMC5510028).

In ESCs DPPA3 appears to localize predominantly in the cytoplasm which had not yet been reported. This is crucial information given the variability of DPPA3 localization during development (Wolfe et al. 2017) as well as the finding that it inhibits UHRF1 in the nucleus.

Taken together, I would suggest that the authors focus on what is novel in their study, and streamline the manuscript in a manner that strengthens these new findings on DPPA3/UHRF1 interactions.

We thank the reviewer for the comment and have followed the suggestion in the revised manuscript.

Reviewer #2 (Remarks to the Author):

Genome-wide methylation reprogramming is a remarkable feature of mammalian development, however the mechanisms driving global hypomethylation in naïve embryonic cells are not elucidated.

Here the authors provide evidence that TET enzymes indirectly drive hypomethylation by activating the expression of DPPA3, and that DPPA3 inhibits global methylation by displacing UHRF1 from chromatin. They propose the very interesting model that Dppa3 has specifically evolved to drive global methylation reprogramming in mammals.

It was previously suggested that DPPA3 impairs global DNA methylation by inhibiting UHRF1 function in somatic cell lines and some of the results on the UHRF1-DPPA3 interaction confirm a 2019 publication, thus some aspects of the manuscript are not entirely novel. Nevertheless, this does not alter the general quality of the work and I am very enthusiastic about this manuscript, which provides important novel mechanistic insights into genome DNA hypomethylation associated with naïve pluripotency in mammalian embryonic cells. The manuscript contains an impressive amount of genomic, biochemical and imaging experiments, which were carefully conducted with many appropriate controls. The manuscript is well written and the results are presented in a clear way.

I have specific concerns that need to be addressed in a revised version:

-The finding of a role of TET1/2 in hypomethylation of naïve cells is in contradiction with a previous article by Von Meyenn et al. Mol Cell (2016), which reported no global hypermethylation in TET1/2 DKO naïve ES cells. How can the findings presented here be reconciled with these previous results? This should be discussed.

We agree with the reviewer that these seemingly contradicting results require further discussion. We now included a comparison of our data to von Meyenn et al. 2016 in Suppl. Fig. 2f and have added a more detailed discussion of this topic to the revised manuscript.

-I would like to see more quality validation of the RRBS data: % bisulfite conversion, correlation between replicates etc... Furthermore in Figure 1a, do both replicates of TET mutants show similar hypermethylation?

We have added quality validation of the RRBS data to the revised manuscript (Suppl. Table 6 and Suppl Fig. 9)

-Figure 2a: it should also be verified by western blot if there are no changes in protein abundance for DNMTs or UHRF1.

We thank the reviewer for the suggestion. We have added the requested western blots to Suppl Figure 2h. Inactivation or loss of TET proteins did not affect protein abundance of the key maintenance DNA methylation regulator, UHRF1.

-Page 9 line 174 on the comparison between TET KO and DPPA3 KO ESCs: the statement that "the majority (~90%) of common targets reside within repetitive elements (Fig. 3d, Supplementary Fig. 3e,f)" deserves clarification. If I am not mistaken, Fig. 3d shows that 90% of hypermethylation events in repeats are common, which is very different than saying that 90% of common hypermethylation events are in repeats. In addition I do not understand what is represented in Supplementary Fig. 3e. Can this point be clarified?

We agree with the reviewer that this passage was not phrased correctly and have changed it in the revised manuscript. Former Supplementary Fig 3e (now 3h) represents the fraction of commonly hypermethylated (dark blue) elements to T1CM, T2CM, and T12CM ESCs and Dppa3KO ESCs. We have added more detailed information to the figure legend.

-To demonstrate causality between TET mediated demethylation and Dppa3 expression, can the authors show that Dppa3 is transcriptionally upregulated upon DNA methylation inhibition in mouse ESCs?

Other studies have previously shown that DPPA3 expression is upregulated upon loss of DNA methylation (Mochizuki et al. 2012). In fact, Dppa3 is among the few genes which have been shown to respond directly to promoter DNA methylation (Auclair et al., 2014; Hackett et al., 2013; Hayashi et al., 2008; Kalkan et al., 2017). We have added this information as well as a plot depicting Dppa3 expression values in wt and DNMT1KO cells (data from: Sharif et al. 2016) to the revised manuscript.

-It is surprising that DPPA3 localization is entirely cytoplasmic in naive mouse ESCs given that it is supposed to shuttle to the nucleus to impair UHRF1 binding to chromatin. If the authors have an antibody, could they probe DPPA3 by western blot in their nuclear/cytoplasmic fractionation assay?

We apologize to the reviewer if the phrasing in the original manuscript was confusing. We made sure to clearly state that the localization of the DPPA3 is not entirely cytoplasmic. This is evident in our live cell imaging experiments, especially our RICS analysis (Fig. 6) which focused on the interaction and dynamics of UHRF1 and DPPA3 within the volume of the nucleus. Nevertheless, to further clarify this issue we now provide a western blot in which we immunoblotted DPPA3 in nuclear and cytoplasmic fraction clearly showing that DPPA3 is largely but not exclusively localized to the cytoplasm (Suppl. Fig. 4e).

-The findings reported here are in apparent contradiction with the documented role of DPPA3 in the maintenance of DNA methylation at imprinted genes in early embryos (Nakamura et al., Nature Cell Biol 2007). It would be interesting to look at methylation of imprinting control regions in the mutant ESCs.

We thank the reviewer for this suggestion. We now provide an analysis of imprinting control regions (ICRs) of DPPA3 knockout cells in the revised manuscript. In comparison to wild type cells ICRs appear to be hypermethylated in ESCs lacking DPPA3. We discuss the implications of this finding in the revised manuscript.

-The experiment introducing DPPA3 in fish embryos is very interesting. Nevertheless, one should be cautious when measuring global DNA methylation levels by 5mC immunofluorescence. Would it be possible to perform RRBS on pooled embryos?

We thank the reviewer for their suggestion and agree that an additional line of evidence would strengthen our 5mC immunofluorescence (IF) data demonstrating that DPPA3 induces dramatic DNA methylation loss in medaka embryos. To this end, we have performed bisulfite sequencing of two intergenic regions in control and Dppa3 mRNA (500ng/ μ l) injected Medaka embryos at the late

blastula stage (~8 h after fertilization). These new data strongly support the genome-wide DNA demethylation observed via IF and have been incorporated into Figure 7h in the revised manuscript.

Minor points:

-The manuscript's initial postulate is that global demethylation in early embryogenesis is mammal-specific whereas TET protein are conserved among vertebrates. Can the authors include specific references in the introduction (line 82) to support their claim that global methylation reprogramming is mammal specific?

We have added specific references to the literature in the appropriate passage of the discussion.

-It should be made explicit in the main text how many independent Crispr-Cas9 clones were generated and studied for each condition.

We have added this information to the main text. For all experiments two independent mutant clones were used.

-The panels of Figure 2 are not presented in order in the text.

We thank both reviewers 2 and 3 for pointing this out and apologize for any confusion this may have caused. We have changed the order of two sentences in the text such that the two panels in question (Fig. 2c and 2e) are now presented in the correct sequence.

Reviewer #3 (Remarks to the Author):

In early mammalian development, the genome undergoes a dramatic wave of global DNA methylation erasure followed by reestablishment. The mechanisms at play in this process, as well as the evolutionary underpinnings for its existence, are active areas of research in the field of epigenetics. The emergence of the TET family of enzymes, which can drive DNA demethylation via iterative oxidation of methyl-cytosines, led to an flood of research papers trying to determine how these proteins could contribute to embryonic DNA methylation reprogramming. Here, Mulholland et al use a mouse embryonic stem cell (ESC) model to assess the role of TET enzymes in global DNA methylation. They made the remarkable observation that TET-dependent demethylation and activation of a single gene—Stella/Dppa3—is required to maintain low DNA methylation in serum-free culture conditions. In this sense, TET enzymes are required for global demethylation, just in a counterintuitive manner. I believe this curiosity per se will be of wide interest to the field of mammalian epigenetics. The authors went on to show that DPPA3 inhibits the DNA methylation maintenance factor UHRF1 via two mechanisms: sequestering to the cytoplasm and, more importantly, preventing UHRF1 from accessing chromatin. These collective findings are reminiscent of two recent studies: Li et al 2018 (PMID: 30487604) showed that DPPA3 maintains hypomethylation oocytes by UHRF1 cytoplasmic sequestration, and von Meyenn et al 2016 (PMID: 27315559) showed that ESCs cultured without serum exhibit impaired UHRF1. What really separates this study from the others is the rigorous dissection of the mechanism of action. I am not surprised, as the labs that contributed to this study have been leaders in our understanding DNA methylation mechanism, and this particular paper will be an important addition their respective oeuvres. I was particularly impressed by their sophisticated use of microscopy in order to assay live chromatin dynamics. The question becomes, is their study consistent with the previous publications? I think the case can be made that it is: the authors show that UHRF1 is both sequestered (as shown in Li et al), and reduced (as shown in von Meyenn et al) because of DPPA3 action. Also, it should be noted that in the Li et al paper, the cell type being assessed was oocytes, which may be subject to different control of DNA methylation pathways. Finally, the authors pushed beyond the other studies by trying to understand DPPA3 action through an evolutionary

lens. DPPA3 is not even present in certain mammalian lineages, such as monotremes, which should be noted, are egg-laying. This implies that DNA methylation reprogramming may be restricted to placental mammals (it has already been shown not to occur in other vertebrates, such as amphibians and fish). As such, DPPA3 may have evolved as the key factor to allow the dramatic reshaping of the epigenome during development. Why this is the case is an interesting matter of debate, that the authors touch on in the discussion. Finally, and strikingly, DPPA3 can induce demethylation when introduced into xenopus (amphibians) or medaka (fish).

All told, I strongly recommend this paper for publication. It is an important paper for the DNA methylation field both in terms of mechanism and evolution. It will be exciting to eventually look in the pre-implantation embryo in order to confirm this mechanism in vivo. I have some improvements that I outline below. We all know these are particular times for reviewing papers, as performing experiments will be difficult. Therefore, I recommend only two small experiments that I would expect in non-lockdown conditions, but would be perfectly willing to overlook given the circumstances:

We thank the reviewer for the appreciation of the current circumstances and the impact they have on our work. While we agree that these experiments would be very interesting we do not think that they change the core message of our manuscript and given the situation chose to address the "Covid-19 times" recommendations.

"Normal times" recommendations:

- Great dissection of mechanism in figure 5, but why not use global quantification of DNA methylation from 0-100%? A LUMA is a quick way to get this result.
- What happens if you express DPPA3 in serum-grown ESCs (which are normally highly methylated). Is this enough to get methylation close to 2i (ie, serum-free) levels?

"Covid-19 times" recommendations

- Figure 2 panels are not described in proper order in text

We thank both reviewers 2 and 3 for pointing this out and apologize for any confusion this may have caused. We have changed the order of two sentences in the text such that the two panels in question (Fig. 2c and 2e) are now presented in the correct sequence.

- Supp figure 2: maybe make the color legend squares, bc it's confusing as ESCs are also circles

We have adjusted the legend according to the reviewers suggestions.

- Personal taste, but I prefer UpSet plots to the complicated Venn diagrams like in figure 3c

We thank the reviewer for introducing us to UpSet plots. We agree that this depiction is simpler to understand while at the same time communicating the absolute size of intersections. We have added the Venn diagram in Figure 3 as an Upset plot to Suppl Figure 3.

- In the discussion, there is no mention of imprinting, which is also a phenomenon restricted to eutherian mammals. Is there a link between DPPA3 and imprinting/genetic conflict? For example, the gametes arrive asymmetrically methylated, which may confer parent specific control of in utero growth. DPPA3 evolved to "level" the differences, but imprints can bypass this control. This is a worthwhile matter of discussion.

We agree with the reviewer and have added a discussion about the possible link between DPPA3 and imprinting to the revised manuscript.

REVIEWERS' COMMENTS

Reviewer #1 (Remarks to the Author):

The authors have sufficiently addressed most of my comments in the revised manuscript. In the rebuttal letter, the authors mention that "We present the first unbiased, deep, and systematic interactome of DPPA3 in mESCs", and that "This allows us and the reader also to view the interaction with UHRF1 in context of the other interactions, like e.g. with elements of import and export machinery", which I agree with. This paper therefore adds new information on the interactions between DPPA3 and UHRF1, and provides a mechanism for the global hypomethylation that takes place during early mammalian development.

I, however, have some comments on the formatting of the manuscript.

1. I have struggled, and I assume that many other readers will have the same experience, to decode the main message of this paper. A main cause of this problem arises from the way the story is written. The title says "Recent evolution of a TET-controlled and DPPA3/STELLA-driven pathway of passive demethylation in mammals". However, the study is basically about DPPA3 and UHRF1. Also, given that it is mainly a mechanistic study, albeit with some experiments using non-mammalian systems, I wonder whether there is a compelling need to include "evolution" in the title. As a general audience of this paper, I would feel comfortable if the title was simpler and to the point, such as "DPPA3/STELLA promotes passive demethylation by direct interaction and cytoplasmic sequestration of UHRF1". However, this is up to the authors to decide.
2. The text is often unnecessarily complicated. Again, from the viewpoint of the reader, I would prefer simplification of some of the phrasing. For example, in the abstract (line 40), the authors write "suggesting a rather indirect mechanism". Why not simply say "suggesting an indirect mechanism"? In line 276, "causing an even greater reduction", could be simply written as "causing a greater reduction". In addition, hyperboles such as (line 435) "a momentous change" or (line 436) "far-reaching consequences" could be omitted.
3. The section "DPPA3-mediated inhibition of UHRF1 chromatin binding causes hypomethylation and is attenuated by nuclear export" is very hard to follow. In the subtitle, it is not clear what exactly is "attenuated" by nuclear export. There are, again, several instances of "even greater (change to: greater)", "even antagonize (change to: antagonize)" and "rather attenuates (change to: attenuates)". I presume that the authors are trying to say that "DPPA3 binds to UHRF1 and disrupts UHRF1 chromatin binding, leading to widespread hypomethylation". If so, I suggest that the authors re-write this section, taking care to reduce scientific jargons and increase readability.
4. I agree that PRDM14 mediated regulation of DPPA3 (lines 156-163) is an important issue. At the same time, I wonder if including the PRDM14 section in this story dramatically improve the overall message? If not, the authors may consider removing (including some parts of the discussion, such as lines 454-468) the PRDM14 part, to simplify the story.
5. Another key message buried in the story is that the hypomethylation mediated by DPPA3 takes place via the interaction of DPPA3 with UHRF1, and not DNMT1. Although DNMT1 and UHRF1 are both essential components of the maintenance methylation machinery, it has been shown that ablation of Dnmt1 or Uhrf1 have different impacts on de-repression of retrotransposons. In other words, interactions of DPPA3 with UHRF1 should be sufficient to achieve global demethylation, while making sure that there is no dramatic upregulation of retrotransposons.
6. The discussion section now covers a broad range of topics, but is also very long. Would it be possible to simplify the text and compress it to about half of the present length?
7. Please re-write the section "Our results show that TET proteins are responsible for active and - indirectly via DPPA3 also for -passive DNA demethylation" (lines 45-46), as the present text is hard to decipher. It should be helpful to mention that the passive demethylation is mediated by the interaction between DPPA3 and UHRF1.

8. In line 46, what does "the recent emergence" mean? The recent emergence in "mammals"?
9. Line 79, despite the "deep" conservation, could be changed to despite the "high level" of conservation.
10. In line 81, how about changing "with 5mC patterns remaining constant", to "in contrast, 5mC patterns remains constant"?
11. How about re-writing lines (82-84) "This discrepancy implies the existence of so far unknown mammalian-specific pathways and factors controlling the establishment and maintenance of genomic hypomethylation", as "This discrepancy implies the existence mammalian-specific pathways that were not known before, to regulate establishment and maintenance of genomic hypomethylation"?
12. Lines 133-134, "the DNA methylation machinery being upregulated" is confusing. Could be re-written as "the upregulation of the proteins involved in the DNA methylation machinery".
13. Lines 171-172, "Repetitive sequences and TEs, in particular, were severely hypermethylated including 98% of L1 elements", could be changed as "Repetitive sequences and TEs were severely hypermethylated. In particular, 98% of L1 elements showed decrease of DNA methylation."
14. Line 237, change "as evidenced" to "as demonstrated".
15. I don't understand what the authors are trying to say by "Additionally, these data demonstrated increased UHRF1 chromatin binding to underlie the more heterogenous nuclear UHRF1 distributions in Dppa3KO and T12CM ESCs" (lines 238-240).
16. Lines 273-274, Fig. 6b), "DPPA3 requires both the capacity to interact with UHRF1 as well as a functional nuclear export signal to promote nucleocytoplasmic shuttling of UHRF1 in naïve ESCs", could be re-written as "interaction of DPPA3 with UHRF1, and nuclear export of DPPA3, are both required for the shuttling of UHRF1 from the nucleus to the cytoplasm in naïve ESCs".

Taken together, I am convinced that this is a great story, and that the experiments are also state-of-the-art. If the text is simplified, and the scientific jargons are removed as much as possible, it should be much easier for the reader to follow the results and grasp the final message.

Reviewer #2 (Remarks to the Author):

The authors have performed experiments and text edits to address all my comments in a satisfactory manner.

Two last details:

- The title of the new Supp Fig.9 should be corrected because it is a repetition of the title of Supp Fig. 8.
- Please add numbers on the y-axis of the boxplot in the new supp Fig. 3f

Reviewer #3 (Remarks to the Author):

The authors have addressed all my concerns. I maintain my strong enthusiasm for publication of this study.

POINT BY POINT RESPONSE

REVIEWERS' COMMENTS

Reviewer #1 (Remarks to the Author):

The authors have sufficiently addressed most of my comments in the revised manuscript. In the rebuttal letter, the authors mention that “We present the first unbiased, deep, and systematic interactome of DPPA3 in mESCs”, and that “This allows us and the reader also to view the interaction with UHRF1 in context of the other interactions, like e.g. with elements of import and export machinery”, which I agree with. This paper therefore adds new information on the interactions between DPPA3 and UHRF1, and provides a mechanism for the global hypomethylation that takes place during early mammalian development.

I, however, have some comments on the formatting of the manuscript.

1. I have struggled, and I assume that many other readers will have the same experience, to decode the main message of this paper. A main cause of this problem arises from the way the story is written. The title says “Recent evolution of a TET-controlled and DPPA3/STELLA-driven pathway of passive demethylation in mammals”. However, the study is basically about DPPA3 and UHRF1. Also, given that it is mainly a mechanistic study, albeit with some experiments using non-mammalian systems, I wonder whether there is a compelling need to include “evolution” in the title. As a general audience of this paper, I would feel comfortable if the title was simpler and to the point, such as “DPPA3/STELLA promotes passive demethylation by direct interaction and cytoplasmic sequestration of UHRF1”. However, this is up to the authors to decide.

We understand the reviewer’s point but we strongly believe that the current title succinctly highlights the most important and novel insights offered by our study. As such, we would like to respectfully decline the suggestion to change the title.

2. The text is often unnecessarily complicated. Again, from the viewpoint of the reader, I would prefer simplification of some of the phrasing. For example, in the abstract (line 40), the authors write “suggesting a rather indirect mechanism”. Why not simply say “suggesting an indirect mechanism”? In line 276, “causing an even greater reduction”, could be simply written as “causing a greater reduction”. In addition, hyperboles such as (line 435) “a momentous change” or (line 436) “far-reaching consequences” could be omitted.

In an effort to address the reviewer’s comments, we have removed the words “rather” and “even” from lines 40 and 276, respectively, as requested.

3. The section “DPPA3-mediated inhibition of UHRF1 chromatin binding causes hypomethylation and is attenuated by nuclear export” is very hard to follow. In the subtitle, it is not clear what exactly is “attenuated” by nuclear export. There are, again, several instances of “even greater (change to: greater)”, “even antagonize (change to: antagonize)” and “rather attenuates (change to: attenuates)”. I presume that the authors are trying to say that “DPPA3 binds to UHRF1 and disrupts UHRF1 chromatin binding, leading to widespread hypomethylation”. If so, I suggest that the authors re-write this section, taking care to reduce scientific jargons and increase readability.

We agree that this section was difficult to understand. We have removed several superlatives as requested and have followed the reviewer’s suggestion to rewrite this section to improve clarity.

4. I agree that PRDM14 mediated regulation of DPPA3 (lines 156-163) is an important issue. At the same time, I wonder if including the PRDM14 section in this story dramatically improve the overall message? If not, the authors may consider removing (including some parts of the discussion, such as lines 454-468) the PRDM14 part, to simply the story.

PRDM14 was the first factor identified to drive the global hypomethylation associated with naive pluripotency and confirmed in several subsequent studies (Leitch et al. 2013, Hackett et al. 2013, Yamaji et al. 2013, Okashita et al. 2014). As PRDM14 transcriptionally represses Dnmt3 expression, PRDM14 was proposed to drive global hypomethylation by suppressing de novo DNA methylation. However, a systematic investigation of the determinants of naive hypomethylation demonstrated that the loss of de novo DNMT3 activity was not sufficient to drive global demethylation, which was instead shown to primarily occur as a result of impaired maintenance DNA methylation (von Meyenn et al. 2016). This raises the question of how PRDM14 could drive demethylation? Although PRDM14 is not the main focus of our story, our data offer important insights that can serve to resolve the uncertainty in the field caused by seemingly conflicting reports. We believe that the TET-controlled and DPPA3-mediated pathway we describe here truly offers the missing link that mechanistically connects the well-documented demethylating factor PRDM14 to the inhibition of maintenance DNA methylation responsible for naive hypomethylation.

5. Another key message buried in the story is that the hypomethylation mediated by DPPA3 takes place via the interaction of DPPA3 with UHRF1, and not DNMT1. Although DNMT1 and UHRF1 are both essential components of the maintenance methylation machinery, it has been shown that ablation of Dnmt1 or Uhrf1 have different impacts on de-repression of retrotransposons. In other words, interactions of DPPA3 with UHRF1 should be sufficient to achieve global demethylation, while making sure that there is no dramatic upregulation of retrotransposons.

We appreciate the reviewer's point. Indeed acute DNMT1 loss leads to a transient upregulation of transposable elements whereas such derepression is far less severe when UHRF1 is knocked out (Sharif et al. 2016). While perhaps not as severe as in DNMT1 KOs, UHRF1 loss results in a substantial derepression of transposons (Sharif et al. 2007; Sharif et al. 2016). In any case, the transcriptional consequences of DPPA3-mediated inhibition of UHRF1 function were not investigated in our study. In the absence of transcriptome data from Dppa3 KO or OE with which to make comparisons with previously published conditional DNMT1 KO and UHRF1 KO expression data, we feel that it would be too speculative and beyond the scope of our current work to make claims about the importance of DPPA3 promoting hypomethylation via an interaction with UHRF1 as opposed to DNMT1.

6. The discussion section now covers a broad range of topics, but is also very long. Would it be possible to simplify the text and compress it to about half of the present length?

We agree with the reviewer that the discussion has indeed become quite long. However, many of the additions covering a seemingly broad range of topics were specifically included to satisfy reviewer requests in the first round of revisions. As such and having apparently satisfactorily addressed said requests, we are reluctant to make significant alterations at this stage.

7. Please re-write the section "Our results show that TET proteins are responsible for active and - indirectly via DPPA3 also for -passive DNA demethylation" (lines 45-46), as the present text is hard to decipher. It should be helpful to mention that the passive demethylation is mediated by the interaction between DPPA3 and UHRF1.

We thank the reviewer for his suggestion and have re-written the abstract to be clearer and easier to understand. We also mention the interaction between DPPA3 and UHRF1.

8. In line 46, what does "the recent emergence" mean? The recent emergence in "mammals"?

We removed this ambiguity in the revised abstract.

9. Line 79, despite the “deep” conservation, could be changed to despite the “high level” of conservation.

We agree that “deep” might be too ambiguous and now describe the conservation as “extensive.”

10. In line 81, how about changing “with 5mC patterns remaining constant”, to “in contrast, 5mC patterns remains constant”?

We agree that this change better emphasizes the difference in developmental DNA methylation regulation between mammals and non-mammalian vertebrates and we changed the text accordingly.

11. How about re-writing lines (82-84) “This discrepancy implies the existence of so far unknown mammalian-specific pathways and factors controlling the establishment and maintenance of genomic hypomethylation”, as “This discrepancy implies the existence mammalian-specific pathways that were not known before, to regulate establishment and maintenance of genomic hypomethylation”?

We have reworded the sentence in question as much as possible while also trying to preserve the original message we intended it to convey.

12. Lines 133-134, “the DNA methylation machinery being upregulated” is confusing. Could be re-written as “the upregulation of the proteins involved in the DNA methylation machinery”.

We have rewritten the lines in question to reduce confusion.

13. Lines 171-172, “Repetitive sequences and TEs, in particular, were severely hypermethylated including 98% of L1 elements”, could be changed as “Repetitive sequences and TEs were severely hypermethylated. In particular, 98% of L1 elements showed decrease of DNA methylation.”

We have changed this passage according to the reviewers suggestion

14. Line 237, change “as evidenced” to “as demonstrated”.

We have made this change as requested.

15. I don't understand what the authors are trying to say by “Additionally, these data demonstrated increased UHRF1 chromatin binding to underlie the more heterogenous nuclear UHRF1 distributions in Dppa3KO and T12CM ESCs” (lines 238-240).

We wanted to explain that the FRAP data from endogenous UHRF1 in Dppa3KO and T12CM ESCs demonstrated that the more pronounced focal patterning of nuclear UHRF1 initially observed in Dppa3KO and T12CM ESCs was indeed the result of increased UHRF1 chromatin binding. We have rewritten the sentence to make this point more comprehensible.

16. Lines 273-274, Fig. 6b), “DPPA3 requires both the capacity to interact with UHRF1 as well as a functional nuclear export signal to promote nucleocytoplasmic shuttling of UHRF1 in naïve ESCs”, could be re-written as “interaction of DPPA3 with UHRF1, and nuclear export of DPPA3, are both required for the shuttling of UHRF1 from the nucleus to the cytoplasm in naïve ESCs”.

We have changed the text as suggested.

Taken together, I am convinced that this is a great story, and that the experiments are also state-of-the-art. If the text is simplified, and the scientific jargons are removed as much as possible, it should be much easier for the reader to follow the results and grasp the final message.

We greatly appreciate the reviewer's enthusiasm for our story and positive assessment of the quality of our work. We thank the reviewer for their suggestions on improving the clarity of the text, as these have markedly improved the readability of several sections.

Reviewer #2 (Remarks to the Author):

The authors have performed experiments and text edits to address all my comments in a satisfactory manner.

Two last details:

-The title of the new Supp Fig.9 should be corrected because it is a repetition of the title of Supp Fig. 8.

We thank the reviewer for drawing our attention to this oversight. The title of Supplementary Figure 9 has been amended in the most recent revised version of the manuscript.

-Please add numbers on the y-axis of the boxplot in the new supp Fig. 3f

We thank the reviewer for pointing this out and added numbers to the y-axis in Suppl. Fig. 3f.

Reviewer #3 (Remarks to the Author):

The authors have addressed all my concerns. I maintain my strong enthusiasm for publication of this study.